# Practical, Provably-Correct Interactive Learning in the Realizable Setting: The Power of True Believers

**Julian Katz-Samuels**
University of Wisconsin, Madison
katzsamuels@wisc.edu

**Blake Mason**
Rice University
bm63@rice.edu

**Kevin Jamieson**
University of Washington
jamieson@cs.washington.edu

**Robert Nowak**
University of Wisconsin, Madison
rdnowak@wisc.edu

## Abstract

We consider interactive learning in the realizable setting and develop a general framework to handle problems ranging from best arm identification to active classification. We begin our investigation with the observation that agnostic algorithms *cannot* be minimax-optimal in the realizable setting. Hence, we design novel computationally efficient algorithms for the realizable setting that match the minimax lower bound up to logarithmic factors and are general-purpose, accommodating a wide variety of function classes including kernel methods, Hölder smooth functions, and convex functions. The sample complexities of our algorithms can be quantified in terms of well-known quantities like the extended teaching dimension and haystack dimension. However, unlike algorithms based directly on those combinatorial quantities, our algorithms are computationally efficient. To achieve computational efficiency, our algorithms sample from the version space using Monte Carlo "hit-and-run" algorithms instead of maintaining the version space explicitly. Our approach has two key strengths. First, it is simple, consisting of two unifying, greedy algorithms. Second, our algorithms have the capability to seamlessly leverage prior knowledge that is often available and useful in practice. In addition to our new theoretical results, we demonstrate empirically that our algorithms are competitive with Gaussian process UCB methods.

## 1 Introduction

In this paper, we study interactive learning where an algorithm makes a decision and observes feedback that it then uses to update its behavior. Interactive learning problems are becoming increasingly widespread in the information era. Examples include A/B/n testing where technology companies perform large-scale experiments to adaptively collect data to optimize their products on platforms like websites or smart phone applications [1]; active classification where learning algorithms adaptively collect data with the hope of learning high-quality predictive models using a much smaller number of labels than is typically required in supervised learning [2]; and environmental monitoring using sensor networks [3].

At a high-level, there are two main algorithmic paradigms for interactive learning: *agnostic* algorithms and *realizability-based* algorithms. Agnostic algorithms may use a model class $\mathcal{F}$ to guide learning, but do not assume that the true data-generating process is well-modeled by $\mathcal{F}$. Because of this, agnostic algorithms tend to have the advantages of being robust to model misspecification and noise. Due to these virtues, agnostic algorithms have received much attention in the literature on interactive learning, e.g., in active classification [4, 5, 6]. By contrast, realizability-based algorithms assume

35th Conference on Neural Information Processing Systems (NeurIPS 2021).

that the model class $\mathcal{F}$ accurately models the real-world and leverages the structure in $\mathcal{F}$ to guide and potentially accelerate learning. Computationally efficient realizability-based algorithms have only been developed for *specific* model classes for problems like best arm identification [7, 8, 9] and regret minimization [10], and the literature lacks a general framework for developing computationally efficient minimax optimal algorithms for generic function classes in the realizable setting.

The starting point of this paper is the basic question: in the realizable setting, can agnostic algorithms compete with realizability-based algorithms? In this paper, we begin by giving a series of negative results that demonstrate that agnostic algorithms pay a significant cost for their robustness to model misspecification. As an example, we show that *any agnostic active classification algorithm* is minimax suboptimal for a class of realizable instances and thus has no hope of competing with realizability-based algorithms in the realizable setting. These results motivate us to develop *a general framework for computationally efficient and sample-efficient algorithms for generic function classes in the realizable setting.* In doing so, we solve an open problem dating back to the work of [11] on the Haystack Dimension, developing the first computationally efficient algorithm for best arm identification with generic function classes that matches the minimax lower bound up to logarithmic factors. Finally, we empirically demonstrate the generality and practicality of our new approach, GRAILS, for function classes ranging from vanilla kernel methods to kernel methods with side information and to the class of convex functions.

## 2 Problem Setup

Let $\mathcal{X}$ denote the input space and $\mathcal{Y} \subset \mathbb{R}$ the output space. We assume that $|\mathcal{Y}| < \infty$, but will relax this assumption later. Let $x_1, \ldots, x_n \in \mathcal{X}$ be a fixed pool of $n$ measurements (or arms) with associated scores $y_1, \ldots, y_n \in \mathcal{Y}$. At each round $t$, the learner selects (or queries) $I_t \in [n]$ and observes $y_{I_t}$. We assume that the learner is given a function class $\mathcal{F} \subset \mathcal{Y}^{\mathcal{X}}$ where $\mathcal{Y}^{\mathcal{X}}$ denotes the set of all functions mapping $\mathcal{X}$ to $\mathcal{Y}$. We say that *realizability* holds if there exists $f^* \in \mathcal{F}$ such that $f^*(x_i) = y_i$ for all $i \in [n]$. An algorithm is *realizability-based* if it assumes realizability. We focus on the noiseless setting here, but it is straightforward to extend our algorithms to handle the case where $y_{I_t}$ is perturbed by independent, additive noise by simple repeated sampling.

We consider the following three objectives:

- **Best arm identification:** The goal is to identify an element of $\operatorname{argmin}_{i \in [n]} y_i$ using as few queries as possible.

- **Cumulative Loss Minimization:** The goal is to identify an element of $\operatorname{argmin}_{i \in [n]} y_i$ while minimizing the loss $\sum_{t=1}^{T} y_{I_t}$ incurred where $T$ is the round that the agent identifies an element of $\operatorname{argmin}_{i \in [n]} f^*(x_i)$.

- **Active Classification:** The goal is to identify an element of $\operatorname{argmin}_{f \in \mathcal{F}} \sum_{i=1}^{n} \mathbb{1}\{y_i \neq f(x_i)\}$ using as few queries as possible.

Best-arm identification is a well-studied problem with applications ranging from clinical trials to A/B/n testing. Cumulative loss minimization is a new problem. Applications include running a clinical trial to determine which of a collection of drugs is most effective while, due to ethical concerns about giving patients ineffective drugs, minimizing the number of participants with bad outcomes. It is closely related to regret minimization where the goal is instead to minimize $\sum_{t=1}^{T_0} y_{I_t} - \min_{i \in [n]} y_i$ for a fixed time horizon $T_0$. Finally, active classification is a mature field whose goal is to minimize the number of labels required to learn a high-quality classifier.

Our main focus in this work is on minimax optimality for a fixed $\mathcal{F}$ in the realizable setting:

- **Best arm identification:** $\Lambda_{best}(\mathcal{F})$ is the smallest integer $q$ such that there exists some algorithm $\mathcal{A}$ such that for every $f^* \in \mathcal{F}$, $\mathcal{A}$ outputs an element of $\operatorname{argmin}_{i \in [n]} f^*(x_i)$ after at most $q$ queries.

- **Cumulative Loss Minimization:** $\Lambda_{loss}(\mathcal{F})$ is the smallest real number $q$ such that there exists some algorithm $\mathcal{A}$ such that for every $f^* \in \mathcal{F}$, $\mathcal{A}$ outputs an element of $\operatorname{argmin}_{i \in [n]} f^*(x_i)$ after incurring a loss of at most $q$.

- **Active Classification:** $\Lambda_{class}(\mathcal{F})$ is the smallest integer $q$ such that there exists some algorithm $\mathcal{A}$ such that for every $f^* \in \mathcal{F}$, $\mathcal{A}$ outputs $f^*$ after at most $q$ queries.

We emphasize that the above notion of minimax optimality is with respect to the class $\mathcal{F}$. Next, we briefly summarize our contributions:

- **Best arm identification:** Assuming we can sample efficiently from a distribution $\pi$ with support $\mathcal{F}$, we give a greedy and computationally efficient algorithm that obtains a sample complexity of $O(\log(n) \log(\frac{1}{\mathbb{P}_\pi(S_{f^*})}) \upsilon^*_{best})$ where $\mathbb{P}_\pi(S_{f^*})$ is the probability of sampling $f^*$ and $\upsilon^*_{best}$ is a combinatorial quantity related to the extended teaching dimension and is a minimax lower bound. This is the first computationally efficient algorithm for best arm identification with generic function classes that matches the minimax lower bound up to logarithmic factors.

- **Loss minimization:** We propose a new algorithm that achieves a loss of $O(\upsilon^*_{loss} \log(|\mathcal{F}|))$ in the worst case where $\upsilon^*_{loss}$ is the minimax lower bound. We show that when applied to the regret minimization setting with general function classes, our algorithm achieves a state-of-the-art regret bound that is always better than the prior state-of-the-art regret bound in [11], can be arbitrarily better, and for a large set of function classes matches the minimax lower bound up to logarithmic factors. Furthermore, using our techniques from our algorithm for best arm identification, we make this algorithm computationally efficient by leveraging a sampling oracle.

- **Active Classification:** We show that there exists a class of realizable instances such that any agnostic algorithm must query $\tilde{\Omega}(\sqrt{n})$ arms to identify the true classifier $f^*$, but there is a realizability-based algorithm that requires $O(\log(n))$ queries. This demonstrates an exponential gap between agnostic and realizability-based algorithms for active classification.

# 3 Related Work

Best arm identification has received much attention in the literature but predominantly for special classes of functions (e.g., linear) [12, 13, 7, 8, 9]. By contrast, our work concerns best arm identification with *general function classes*, which has received much less attention, and is most closely related to [11]. [11] introduce a combinatorial quantity $\mathrm{HD}(\mathcal{F})$ known as the *haystack dimension*, and propose a greedy algorithm that achieves a sample complexity of $\mathrm{HD}(\mathcal{F}) \log(|\mathcal{F}|)$, but is not computationally efficient. We build on this work by designing a greedy algorithm that achieves computational efficiency by appealing to [14] to sample functions from the function class $\mathcal{F}$. A key technical challenge is that the version space under a greedy algorithm is not convex and therefore standard sampling algorithms like "hit and run" cannot be directly applied to it.

[15] gives a general computationally efficient algorithm that solves many pure exploration problems such as active classification and clustering. Their algorithm requires specifying a distance $d$ between functions in the function class $\mathcal{F}$ and proceeds by shrinking the average diameter of $\mathcal{F}$ as measured by $d$. Unfortunately, for problems such as best arm or $\epsilon$-good arm identification, it is unclear how to construct the appropriate distance function to achieve optimal performance or even if such a function exists for a given application.

**Active Classification:** There have been many works on active classification [6, 2], both in the realizable setting (e.g., [16, 17, 18]) and on agnostic algorithms (e.g., [5, 19, 4, 20]). Proposition 1 shows that agnostic algorithms are minimax suboptimal in the realizable setting, suggesting that they may never be able to match the performance of realizability-based algorithms. Our work is closely related to the problem of *Exact learning* where a learning algorithm is required to identify the true $f^*$ with probability 1 [21, 22], in contrast to the PAC requirement where a failure probability of $\delta$ is permitted. While this requirement is certainly strong, this framework enables the design of practical algorithms, as suggested by our experiments.

**Bayesian Optimization:** Our work is also related to the vast field of Bayesian optimization. We review a few relevant theoretical results and refer readers to [3] for a thorough survey. [23] propose and analyze GP-UCB. Although GP-UCB is applied often to best arm identification problems, [23] only give a regret bound and optimality is unclear. Other notable works include [24] whose results are asymptotic and the recent paper [25], which gives a new algorithm for best arm identification for kernel bandits based on experimental design, but its optimality is unclear.

## 4 No Free Lunch for Agnostic Algorithms

Many agnostic algorithms have been proposed in the active classification literature (e.g. [5, 19, 4]). We say an algorithm $\mathcal{A}$ is $\delta$-*agnostic* if for any labeling of the data $y \in \mathcal{Y}^n$, $\mathcal{A}$ finds the best classifier, $\text{argmin}_{f \in \mathcal{F}} \sum_{i=1}^{n} \mathbb{1}\{y \neq f(x_i)\}$, with probability at least $1 - \delta$. Despite considering a much larger class of possible labelings, agnostic algorithms have been shown to achieve the lower bound in the *realizable setting* for the well-studied problem of thresholds where $\mathcal{F} = \{f_1, \ldots, f_n\}$ and $f_i(x_j) = 1$ if $j \leq i$ and $f_i(x_j) = 0$ otherwise [26]. This raises the question: *can agnostic algorithms be optimal over classes of realizable instances in general*? The following result shows that the sample complexity of agnostic algorithms is necessarily worse than the minimax lower bound by an exponential factor.

**Proposition 1.** *Let $\delta \in (0, \min(\frac{1}{20}, \frac{1}{n}))$. Consider the active classification setting. There exists $x_1, \ldots, x_n$ and $\mathcal{F}$ forming a class of instances $\mathcal{I} = \{(x_i, f(x_i))_{i=1}^{n} : f \in \mathcal{F}\}$ such that*

- *the expected number of samples of any $\delta$-agnostic algorithm is $\Omega\left(\log(1/\delta)\frac{\sqrt{n}}{\log(n/\delta)}\right)$ on one of the instances in $\mathcal{I}$, and*

- *there exists a realizability-based algorithm that solves each instance in $\mathcal{I}$ in $O(\log(n))$ queries.*

It turns out agnostic algorithms are also minimax suboptimal for regret minimization, a problem closely related to loss minimization as we will discuss in more detail in Section 6. Here, we say that a regret minimization algorithm $\mathcal{A}$ is $\epsilon$-*agnostic* if for any $y \in \mathcal{Y}^n$ such that $\min_{j \in [n] \setminus \{i_*\}} y_j - y_{i_*} \geq \epsilon$ where $i_* \in \text{argmin}_{i \in [n]} y_i$, $\mathcal{A}$ suffers at most bounded regret independent of the time horizon.

**Proposition 2.** *There exists $\mathcal{F}$ such that any $1$-agnostic algorithm suffers regret at least $O(|\mathcal{F}|)$ for some instance in $\mathcal{F}$ while there exists a realizability-based algorithm suffering regret at most $O(\log(|\mathcal{F}|))$.*

For best-arm identification, a fully agnostic algorithm must consider any $y \in \mathcal{Y}^n$ and it is therefore trivial that it would need to query every $x_i$. Therefore, we consider a weaker notion of agnostic algorithm. For $k \in \mathbb{N}$ and $\delta \in (0, 1)$, we say an algorithm $\mathcal{A}$ is $(\delta, k)$-*agnostic* if for any $y \in \mathcal{Y}^n$ such that $\min_{f \in \mathcal{F}} \sum_{i=1}^{n} \mathbb{1}\{f(x_i) \neq y_i\} \leq k$, $\mathcal{A}$ identifies $\text{argmin}_{i \in [n]} y_i$ with probability at least $1 - \delta$. Despite only allowing for small amounts of mispecification, there is still an exponential gap between the performance of agnostic algorithms and the minimax lower bound.

**Proposition 3.** *Let $\delta \in (0, \min(\frac{1}{40}, \frac{1}{n}))$. There exists $\mathcal{F}$ such that for any algorithm $\mathcal{A}$ that is $(\delta, 1)$-agnostic with respect to $\mathcal{F}$, $\mathcal{A}$ takes $\Omega\left(\frac{n}{\log(n/\delta)}\right)$ queries in expectation on some instance in $\mathcal{F}$, while there exists a realizability-based algorithm requiring $O(1)$ samples on all instances in $\mathcal{F}$.*

**Techniques:** The above results rely on a novel approach for constructing instance-dependent lower bounds for the noiseless setting. The key idea is a reduction of the noiseless setting to a setting where observations are corrupted with a Gaussian random variable, that is, when the arm $i \in [n]$ is queried, the agent observes $y_i + \eta$ where $\eta \sim N(0, 1)$, instead of $y_i$. This reduction enables the application of the transportation Lemma from the multi-armed bandit literature [27] to the noiseless setting and thereby to construct instances where agnostic algorithms *necessarily* perform poorly.

## 5 Best Arm Identification

Given the limitations of agnostic algorithms for interactive learning in the realizable setting established in the prior section, we now turn to developing *realizability-based* algorithms that are computationally efficient and match the minimax lower bound up to logarithmic factors. In this section, we examine the best arm identification problem in which the goal is to identify an $i_* \in \text{argmin}_{i \in [n]} y_i$ using as few queries as possible. Practitioners may be willing to sacrifice a bit of optimality if that makes it easier to solve a problem and thus here we state some of our results and our algorithm in terms of $\epsilon$-good arm identification, a strict generalization of best arm identification. In this problem, we are given $\epsilon \geq 0$ and the goal is to identify an $\epsilon$-good arm, that is, an $i_* \in [n]$ such that $y_{i_*} \leq \min_{i \in [n]} y_i + \epsilon$. When $\epsilon = 0$, this reduces to best arm identification.

We begin by introducing a new quantity, inspired by the extended teaching dimension, for quantifying the difficulty of identifying an $\epsilon$-good arm in a worst-case sense.

$$v_{best,\epsilon}^*(\mathcal{F}) = \max_{g:\mathcal{X} \mapsto \mathcal{Y}} \min_{I \subset [n]} |I|$$

$$\text{s.t. } \exists j \in [n] : \{f \in \mathcal{F} : f(x_i) = g(x_i) \, \forall i \in I\} \subset \left\{ f \in \mathcal{F} : f(x_j) \leq \min_{l \in [n]} f(x_l) + \epsilon \right\}.$$

We occasionally write $v_{best,\epsilon}^*$ instead of $v_{best,\epsilon}^*(\mathcal{F})$ when the context leaves no ambiguity. When $\epsilon = 0$, we simply write $v_{best}^*(\mathcal{F})$ instead of $v_{best,\epsilon}^*(\mathcal{F})$. In words, $v_{best}^*$ is the minimum number of samples required so that for any function $g : \mathcal{X} \mapsto \mathcal{Y}$, there is a subset of queries of size $v_{best}^*$ that can make the best arm $i_* \in [n]$ unambiguous by eliminating all $f \in \mathcal{F}$ that do not put $i_*$ as the best. The following theorem establishes $v_{best}^*$ as a lower bound to the optimal minimax sample complexity $\Lambda_{best}$ for best arm identification.

**Theorem 1.** *For any $\mathcal{F} \subset \mathcal{Y}^{\mathcal{X}}$, $v_{best}^*(\mathcal{F}) \leq \Lambda_{best}(\mathcal{F})$.*

The setting of best arm identification in general function classes was previously studied in [11] in which the sample complexity results were quantified in terms of the Haystack dimension. Specifically, letting $\mathcal{F}' \subset \mathcal{F}$, define $\mathcal{F}'((x_i, y)) = \{f \in \mathcal{F} : f(x_i) \neq y\}$, the subset of functions in $\mathcal{F}'$ that disagree with the label $y$ on $x_i$, and $\mathcal{F}'(x_i) = \{f \in \mathcal{F} : i \in \arg\min_{j \in [n]} f(x_j)\}$, the set of functions in $\mathcal{F}'$ that are minimized at $x_i$. Define $\gamma(\mathcal{F}') := \sup_{i \in [n]} \inf_{y \in \mathcal{Y}} \frac{|\mathcal{F}'(x_i) \cup \mathcal{F}'((x_i, y))|}{|\mathcal{F}'|}$. The Haystack dimension is defined as:

$$\text{HD}(\mathcal{F}) := \frac{1}{\inf_{\mathcal{F}' \subset \mathcal{F}} \gamma(\mathcal{F}')}.$$

The following Proposition shows that $v_{best}^*(\mathcal{F})$ is never significantly less than the Haystack dimension $\text{HD}(\mathcal{F})$ and is never greater than $\text{HD}(\mathcal{F})$ by more than a $O(\log(|\mathcal{F}|))$ factor.

**Proposition 4.** *For any $\mathcal{F} \subset \mathcal{Y}^{\mathcal{X}}$, $HD(\mathcal{F}) - 1 \leq v_{best}^*(\mathcal{F}) \leq cHD(\mathcal{F}) \log(|\mathcal{F}|)$, where $c$ is a positive universal constant.*

## 5.1 Sampling Oracles for Efficient Realizable Active Learning

In this section we introduce the concept of a *sampling oracle*, a key tool for achieving computational efficiency. In contrast to prior active methods that enumerate an intractably large version space (e.g., [11, 21]), we instead place a measure over the version space to track its size without explicitly storing it and use sampling to approximate this measure. Let $\mathcal{R} \subset \mathbb{R}^{\mathcal{X}}$ be a set of regression functions where $\mathbb{R}^{\mathcal{X}}$ denotes the set of all functions $r : \mathcal{X} \mapsto \mathbb{R}$. Given $\mathcal{R} \subset \mathbb{R}^{\mathcal{X}}$, we say that we have access to a *sampling oracle* for $\mathcal{R}$ if there exists a distribution $\pi$ on $\mathcal{R}$ such that

- we can draw $r \sim \pi$, and
- for any $\widetilde{\mathcal{R}} = \{r \in \mathcal{R} : (r(x_1), \ldots, r(x_n))^\top \in \mathcal{H}\}$ where $\mathcal{H}$ is the intersection of $\text{poly}(n, d)$ halfspaces of the form $\{x \in \mathbb{R}^n : w^\top x \leq b\}$, we can sample $r \sim \pi_{\widetilde{\mathcal{R}}}$ where for any measurable $A \subset \widetilde{\mathcal{R}} \, \pi_{\widetilde{\mathcal{R}}}(A) = \frac{\pi(A)}{\pi(\widetilde{\mathcal{R}})}$.

By representing constraints on the version space as a set $\mathcal{H}$ in the second bullet, we can leverage sampling techniques to estimate the measure over the version space efficiently. We show in Appendix B that non-trivial sampling oracles are available for many useful function classes such as linear models, kernel methods, and convex functions. To keep the presentation simple, we assume here that we can compute probabilities exactly and show that approximation suffices in the appendix.

For $z \in \mathbb{R}$, let $\mathcal{D}[z]$ denote the $y \in \mathcal{Y}$ that is closest to $z$, tiebreaking by rounding down. The set of regressors $\mathcal{R}$, together with $D[\cdot]$, induces a set of discretized functions mapping $\mathcal{X}$ to $\mathcal{Y}$

$$\mathcal{F}_{\mathcal{R}} = \{\mathcal{D}[r] : r \in \mathcal{R}\}.$$

Our algorithmic approach is to use $\mathcal{F}_{\mathcal{R}}$ as our predictors and to leverage the structure of $\mathcal{R}$ to achieve computational efficiency. Our algorithm relies on *greedy volume reduction*, which requires that the $f \in \mathcal{F}$ have nonzero measure and the discretized setting ensures this. In the appendix, we extend our algorithms to the continuous setting but at the cost of a potentially suboptimal sample complexity. We make the realizability assumption that there exists $r^* \in \mathcal{R}$ such that $\mathcal{D}[r^*(x_i)] = y_i$ for all $i \in [n]$.

## 5.2 The Algorithm

In this section, we present the algorithm for $\epsilon$-good arm identification, a strict generalization of best arm identification. GRAILS (see Algorithm 1) implicitly maintains a version space $\mathcal{R}_t$ at each round over $\mathcal{R}$ and thus we define

$$\mathcal{R}_t(x_i, y) = \{r \in \mathcal{R}_t : \mathcal{D}[r(x_i)] \neq y\}, \ \mathcal{R}_{t,\epsilon}(x_i) = \{r \in \mathcal{R}_t : \mathcal{D}[r(x_i)] \leq \min_{j \in [n]} \mathcal{D}[r(x_j)] + \epsilon\},$$

defined analogously to the set $\mathcal{F}'((x_i, y))$ and $\mathcal{F}'(x_i)$ above. $\mathcal{R}_t(x_i, y)$ consists of functions in the version space that are inconsistent with the observation $(x_i, y)$ and $\mathcal{R}_{t,\epsilon}(x_i)$ is the set of functions in the version space for which $x_i$ is an $\epsilon$-good arm and therefore no longer need to be considered. When $\epsilon = 0$, we write $\mathcal{R}_t(x_i)$ instead of $\mathcal{R}_{t,\epsilon}(x_i)$.

Algorithm 1 takes the greedy approach. At each round $t$, our algorithm queries a point $I_t \in [n]$ that maximizes the measure of the functions removed from the version space under a distribution $P_k$, namely $\mathbb{P}_{r \sim P_k}(r \in \mathcal{R}_t(x_i) \cup \mathcal{R}_{t,\epsilon}(x_i, y))$, for the worst case observation $y \in \mathcal{Y}$. Once the algorithm queries $I_t$, it adds $I_t$ to $O_t$, the set of observations up to time $t$.

Now, we describe our sampling approach. To keep the presentation simple and to capture the main ideas, we assume here $\mathcal{R}$ is convex. Define $\texttt{better}_\epsilon = \max_{y < \min_{i \in O_t} y_i - \epsilon} y$, the largest value that is smaller than any observation by more than $\epsilon$, which is well-defined since $\mathcal{Y}$ is discrete. Note that $r \in \mathcal{R}_t$ only if there exists $l \in [n] \setminus O_t$ such that $\mathcal{D}[r(x_l)] \leq \texttt{better}_\epsilon$. We may decompose the version space $\mathcal{R}_t$ as the union of $O(n)$ sets:

$$\mathcal{R}_t = \cup_{l \in [n] \setminus O_t} \{r \in \mathcal{R} : \mathcal{D}[r(x_i)] = y_i \, \forall i \in O_t, \text{ and } \mathcal{D}[r(x_l)] \leq \texttt{better}_\epsilon\} = \cup_{l \in [n] \setminus O_t} C_l(O_t)$$

where

$$C_l(O_t) := \{r \in \mathcal{R} : \mathcal{D}[r(x_i)] = y_i \, \forall i \in O_t, \text{ and } \mathcal{D}[r(x_l)] \leq \texttt{better}_\epsilon\}.$$

Each set $C_l(O_t)$ is an intersection of $\mathcal{R}$ with $O(n)$ halfspaces and is a convex set. Unfortunately, a union of convex sets need not be convex so one cannot hope to directly apply algorithms like hit-and-run to efficiently sample from $\mathcal{R}_t$. To overcome this, the algorithm samples from a mixture with each component supported on $C_l(O_t)$, a convex set. Operating in stages, in stage $k$, it puts the measure $P_k$ over the remaining functions in the version spaces where

$$P_k := \frac{1}{n - |O_t|} \sum_{l \in [n] \setminus O_t} \pi_{C_l(O_t)}.$$

Sampling from the mixture $P_k$ is a key algorithmic innovation in this work and enables efficient sampling from a non-convex version space.

We make two final remarks. First, STOP($\mathcal{F}_\mathcal{R}, O_t$) is a subroutine for terminating the algorithm. It essentially checks whether the version space $\mathcal{R}_t$ is empty by checking whether each of the $C_l(O_t)$ sets is feasible. For many function classes of interest such as linear models, kernel methods, and the class of convex functions, this can be formulated as a convex feasibility problem (see the Appendix for a concrete instance). Second, in practice, one may not know the true model class precisely apriori, but it is straightforward to do model selection through the standard doubling technique. For example, for the class of Lipschitz functions, one may not know the true Lipschitz constant. In these situations, one can apply a standard doubling trick on the Lipschitz constant.

## 5.3 The Upper Bound

We now introduce some notation necessary to state our upper bound. Define for every $f \in \mathcal{F}_\mathcal{R}$ the set

$$S_f = \{r \in \mathcal{R} : \mathcal{D}[r(x_i)] = f(x_i) \, \forall i \in [n]\}.$$

The sets $S_f$ induce a partition of $\mathcal{R}$.

**Theorem 2.** *Fix $\mathcal{R}$. Let $\epsilon \geq 0$. There exists a universal constant $c > 0$ such that if $t$ is greater than*

$$c v^*_{best,\epsilon}(\mathcal{F}_\mathcal{R}) \log(\frac{1}{\mathbb{P}_\pi(S_{f^*})}) \log(n),$$

*then Algorithm 1 has pulled an arm $I_s$ at some $s \leq t$ such that $y_{I_s} \leq \min_{i \in [n]} y_i + \epsilon$.*

```
P_1 ⟵ π, R_1 ⟵ R, k ⟵ 1, t_1 = 1, O_1 ⟵ ∅;
for t = 1, 2, ... do
    Let I_t ∈ arg max_{i∈[n]\O_t} min_{y∈Y} ℙ_{r∼P_k}(r ∈ R_{t,ε}(x_i) ∪ R_t(x_i, y));
    Query x_{I_t} and observe y_{I_t} and set O_{t+1} ⟵ O_t ∪ {I_t} ;
    Let R_{t+1} ⟵ R_t \ (R_{t,ε}(x_{I_t}) ∪ R_t(x_{I_t}, y_{I_t}));
    if ℙ_{r∼P_k}(r ∈ R_{t+1}) ≤ 1/{2n} then
        k ⟵ k + 1;
        t_k ⟵ t + 1, P_{k+1} ⟵ 1/{n−|O_{t_k}|} ∑_{l∈[n]\O_{t_k}} π_{C_l(O_{t_k})};
    if STOP(F_R, O_t) then
        return arg min_{i∈O_t} y_i
```
**Algorithm 1:** GRAILS (GReedy Algorithm for Interactive Learning using Sampling)

When $\epsilon = 0$, Algorithm 1 and Theorem 2 together solve the open problem from [11] of developing a computationally efficient optimal algorithm for best arm identification for generic function classes that matches the minimax lower bound up to logarithmic factors.

**Comparison to prior work:** [11] proposes a *computationally inefficient* algorithm for best arm identification that obtains a sample complexity of $O(\mathrm{HD}(F_R)\log(|F_R|))$. By Proposition 4, $v_{best}^*(F_R)$ is upper bounded by $\mathrm{HD}(F_R)\log(|F_R|)$, and thus our sample complexity is loose by a factor of $\log(\frac{1}{\mathbb{P}_\pi(S_{f^*})})\log(n)$. Although our bound is indeed looser than the bound in [11], we note that computationally efficient algorithms for other active learning problems that match the minimax lower bound up to logarithmic factors have a similar logarithmic dependence on the inverse probability of sampling the true function [18, 15]. Thus, it is an important open question whether it is possible to develop computationally efficient and nearly minimax optimal algorithms for active learning that weaken or remove the dependence on $\log(\frac{1}{\mathbb{P}_\pi(S_{f^*})})$ in their sample complexity.

We close this section with a simple instance of best arm identification in a linear function class that provides an easy instantiation of our upper bound.

**Proposition 5.** *Let* $x_1, \ldots, x_n \in \mathbb{R}^{n+1}$ *such that* $x_i = \frac{i}{n}e_1 + 10 \cdot e_{i+1}$. *Let* $R_i = \{r(v) = v_1 - b + v_{i+1} : b \in \frac{1}{n}[i, i+1]\}$ *where* $v_i$ *denotes the ith entry of* $v$ *and* $R = \cup_{i=1}^n R_i$. *Define* $Y = \{0, 1, 10\}$. *Let* $\pi$ *be a uniform distribution over* $[0, 1]$. *Fix* $f^* \in F_R$. *Then,* GRAILS *returns the best arm in* $v_{best}^*(F_R)\log(\frac{1}{\mathbb{P}_\pi(S_{f^*})})\log(n) \leq O(\log(n)^2)$ *samples.*

By contrast, any non-adaptive algorithm would require $\Omega(n)$ samples for the above function class. Furthermore, we suspect that the additional logarithmic factor stemming from Theorem 2 is an artifact of the analysis. We discuss other instances in the appendix.

## 6 Cumulative Loss Minimization

In this Section, we consider the task of loss minimization. To interpret each $y_i$ as a loss, we assume $\min_{y\in Y} y \geq 0$. Recall that in this setting when the learner identifies an element $i^* \in \arg\min_{i\in[n]} f^*(x_i)$, she declares that the game is over. We stress that the learner can identify an element $i^* \in \arg\min_{i\in[n]} f^*(x_i)$ by eliminating all $f \in F$ such that $i^* \notin \arg\min_{i\in[n]} f(x_i)$.

To begin, we introduce the following novel quantity for quantifying the worst-case difficulty of cumulative loss minimization, inspired by the extended teaching dimension:

$$v_{loss}^*(F) := \max_{g:X\mapsto Y} \min_{I\subset[n]} \min_{\widetilde{I}\subset I:|\widetilde{I}|=|I|-1} \sum_{i\in\widetilde{I}} g(x_i)$$
$$\text{s.t. } \exists j \in [n] : \{f \in F : f(x_i) = g(x_i) \,\forall i \in [I]\} \subset \{f \in F : j \in \arg\min_{i\in[n]} f(x_i)\}.$$

We occasionally write $v_{loss}^*$ instead of $v_{loss}^*(F)$ when the context leaves no ambiguity. In words, $v_{loss}^*$ is the loss that must be incurred (up to the penultimate round) for some scoring function $g : X \mapsto Y$ in order to identify the best arm. Next, we show that $v_{loss}^*$ is a minimax lower bound.

**Theorem 3.** *For all* $F \subset Y^X$, $v_{loss}^*(F) \leq \Lambda_{loss}(F)$.

The following Proposition gives an upper bound of $v_{loss}^*$ in terms of $v_{best}^*$.

```
P_1 ⟵ π, R_1 ⟵ R, k ⟵ 1, t_1 = 1, O_1 ⟵ ∅;
for t = 1, 2, . . . do
    Let I_t ∈ argmin_{i∈[n]\O_t} max_{y∈Y} y / (P_{r∼P_k}(r ∈ R_t(x_i) ∪ R_t((x_i,y))));
    Query x_{I_t} and observe y_{I_t} and set O_{t+1} ⟵ O_t ∪ {I_t} ;
    Let R_{t+1} ⟵ R_t \ (R_t(x_{I_t}) ∪ R_t((x_{I_t}, y_{I_t})));
    if P_{r∼P_k}(r ∈ R_{t+1}) ≤ 1/(2n) then
        k ⟵ k + 1;
        t_k ⟵ t + 1, P_{k+1} ⟵ 1/(n−|O_{t_k}|) Σ_{l∈[n]\O_{t_k}} π_{C_l(O_{t_k})};
    if STOP(F_R, O_t) then
        return argmin_{i∈O_t} y_i
```

**Algorithm 2:** `GRAILS` for Loss Minimization

**Proposition 6.** *For all $F \subset Y^X$, $v^*_{loss}(F) \le v^*_{best}(F) max_{y\in Y} y$.*

In words, this Proposition reflects that one strategy for cumulative loss minimization is to minimize the number of queries to identify the best arm, ignoring the losses.

### 6.1 The Algorithm and Upper Bound

Algorithm 2 is similar to Algorithm 1: it is greedy and operates in phases, sampling from the mixture $P_k$ in the $k$th phase due to the nonconvexity of the version space. The main difference is the objective for selecting $I_t$: it queries the arm $I_t$ that for the worst case $y \in Y$ minimizes the ratio of the loss incurred in round $t$ and the volume of the functions removed under the measure $P_k$. This objective is inspired by the greedy algorithm for weighted set cover [28].

Define $S_{\min} = argmin_{S_f : f\in F_R, P_\pi(S_f)>0} P_\pi(S_f)$. $S_{\min}$ is the subset of $F_R$ in the partition that has the least nonzero probability under $\pi$. Recall $R$ is a set of regression functions.

**Theorem 4.** *Fix $R$. Algorithm 2 identifies $argmin_{i\in[n]} f^*(x_i)$ after incurring a loss of at most*

$$\widetilde{O}(\Lambda_{loss}(F_R)) = 2(v^*_{loss}(F_R) + max_{i\in[n], f\in F_R} f(x_i)) \log(n) \log\left(\frac{1}{P(S_{\min})}\right).$$

The above guarantee is optimal up to a multiplicative factor of $\log(n)\log(\frac{1}{P(S_{\min})})$ and an additive factor of $max_{i\in[n], f\in F} f(x_i)$, which is typically of lower order. We also give an algorithm that enumerates the function class and therefore is not efficient when $|F_R|$ is exponential in problem-dependent parameters, but has a stronger guarantee. Due to space constraints and its similarity to Algorithm 2, we defer its presentation to the supplementary material and provide the result here.

**Theorem 5.** *Fix $F$. Algorithm 2 identifies $argmin_{i\in[n]} f^*(x_i)$ after incurring a loss of at most*

$$\widetilde{O}(\Lambda_{loss}(F)) = 2(v^*_{loss}(F) + max_{i\in[n], f\in F} f(x_i)) \log(|F|).$$

Theorems 4 and 5 are the first algorithms for cumulative loss minimization that match the minimax lower bound up to logarithmic factors. As a corollary, we obtain results for *regret minimization*, in which the goal is to identify $argmin_{i\in[n]} f^*(x_i)$ while minimizing the regret $\sum_{t=1}^T f^*(x_{I_t}) - \min_{i\in[n]} f^*(x_i)$ incurred where $T$ is the round that the agent identifies $argmin_{i\in[n]} f^*(x_i)$.

**Remark 1.** *If $\min_j f(x_j) = 0$ for all $f \in F$, then loss minimization is equivalent to regret minimization. In this case, Theorem 5 gives nearly optimal minimax bounds for the regret minimization metric. Furthermore, if $\min_j f(x_j) = \min_j f'(x_j) =: opt$ for all $f, f' \in F$, then by subtracting opt from each $f$, we reduce to the setting where $\min_j f(x_j) = 0$ for all $f \in F$ and obtain a nearly optimal minimax bound.*

To the best of our knowledge, *these are the first regret bounds that match the minimax lower bound up to logarithmic factors for a large and general class of function classes.*

**Comparison to [11].** Next, we compare our algorithm in Theorem 5 to the *computationally inefficient* algorithm from [11] for regret minimization in the noiseless setting. Their regret scales as

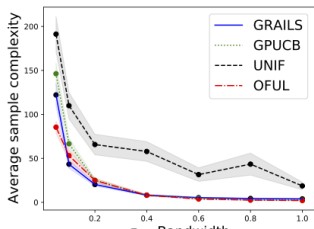
Figure 1: Varying $\sigma$

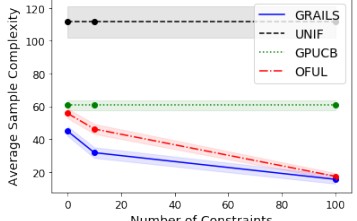
Figure 2: Prior Knowledge

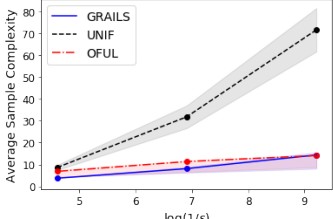
Figure 3: Convex Functions

$O(\Delta_{\max}\mathrm{HD}(\mathcal{F})\ln(|\mathcal{F}|))$, ignoring lower order terms. The following Proposition shows that our cumulative regret bound is never worse than theirs by more than a polylogarithmic factor. Let $y_* = \mathrm{argmin}_{y\in\mathcal{Y}}y$ and $\Delta_{\max} = \max_{y\in\mathcal{Y}}y - y_*$ and $\Delta_{\min} = \min_{y\in\mathcal{Y}:y\neq y_*}y - y_*$.

**Proposition 7.** *Let $\mathcal{F}$ such that $\upsilon_{best}^*(\mathcal{F}) \geq 1$. There exists a universal constant $c > 0$ such that for large enough $T_0$, if Algorithm 3 is run for $T_0$ rounds, the regret of Algorithm 3 is bounded above by*

$$c\upsilon_{best}^*(\mathcal{F})\ln(|\mathcal{F}|)\Delta_{max} \leq cHD(\mathcal{F})\ln(|\mathcal{F}|)^2\Delta_{max}.$$

The above Proposition says that if the horizon is long enough, then the cumulative regret in Theorem 5 is never that much worse than the cumulative regret of the algorithm in [11].

On the other hand, there exists an instance where the regret of our Algorithm in Theorem 5 is significantly better than the regret of the algorithm in [11].

**Proposition 8.** *For any $\xi > 0$, there exists an instance with $\Delta_{max} \geq \Delta_{\min} + \xi$ where (ignoring logarithmic factors) the regret minimization algorithm from [11] obtains a regret of at least $\Delta_{max}n/2$ while the guarantee in Theorem 5 is $n\Delta_{\min} + \Delta_{max}$.*

Thus, the regret of [11] scales as $\Delta_{\max}n$, while the regret of Algorithm 3 scales as $\Delta_{\min}n$, and the gap between $\Delta_{\max}$ and $\Delta_{\min}$ can be made aribtrarily large. *The key advantage of our algorithm over the work of [11] is that their algorithm is an explore-and-then-commit algorithm, ignoring the cost of information gain, whereas our algorithm Algorithm 3 (see Appendix) is cost-aware weighing the tradeoff between information gain and loss incurred.*

**Relation to Regret Minimization.** We conclude this section by discussing the relationship between loss minimization and regret minimization. Our first observation is that any minimax-optimal regret-minimizing algorithm is also a minimax-optimal loss-minimizing algorithm, as shown by the following Proposition.

**Proposition 9.** *Fix $T_0 \in \mathbb{N}$. Let $loss(\mathcal{A}; f; T_0) = \sum_{t=1}^{T_0} f(x_{I_t})$ and $regret(\mathcal{A}; f; T_0) = \sum_{t=1}^{T_0} f(x_{I_t}) - \min_{j\in[n]} f(x_j)$ denote the loss and regret incurred by an algorithm $\mathcal{A}$ over $T_0$ rounds. Let $\bar{\mathcal{A}}$ be an algorithm such that $\max_{f\in\mathcal{F}}regret(\bar{\mathcal{A}}; f; T_0) \leq c\min_{\mathcal{A}}\max_{f\in\mathcal{F}}regret(\mathcal{A}; f; T_0)$. Then,*

$$max_{f\in\mathcal{F}}loss(\bar{\mathcal{A}}; f; T_0) \leq (c+1)\min_{\mathcal{A}}max_{f\in\mathcal{F}}loss(\mathcal{A}; f; T_0).$$

On the other hand, a minimax optimal algorithm for loss minimization can have regret that is arbtrarily worse than the minimax lower bound, implying that regret minimization subsumes loss minimization.

**Proposition 10.** *For any $\xi > 0$, there exists $\mathcal{F}$ such that the minimax regret is 1, but a loss minimizing algorithm obtains a regret of $\xi$.*

As discussed above, *we provide the first upper bounds for loss minimization that match the minimax lower bound up to logarithmic factors, as well as state-of-the-art results for regret minimization with general function classes.*

## 7   Experiments

In this Section, we present experiments comparing `GRAILS` to uniform sampling (`UNIF`), `GP-UCB` [23], and `OFUL` [29]. We consider a version of `GRAILS` that is designed for continuous output spaces, and

which we present and analyze in the appendix. For `OFUL`, we construct the lower confidence bound by solving a constrained optimization problem based on the prior feedback and the function class structure. Our algorithm outperforms the other algorithms and demonstrates the ability to seamlessly incorporate prior knowledge to accelerate learning, and works for function classes ranging from kernel methods to convex functions. While `OFUL` tends to perform only slightly worse than `GRAILS`, unlike `GRAILS`, *it does not have general theoretical guarantees* that can handle the incorporation of constraints given by prior knowledge or specially structured spaces like the set of convex functions, and therefore we consider it a strong, if heuristic, baseline. *Indeed, in the appendix, we show that `OFUL` is minimax suboptimal for some function classes.* Due to space constraints, we provide additional implementation details in the appendix.

**Varying $\sigma$.** In Figure 1, we plot the average number of samples necessary to achieve a simple regret of less than $0.01$ with the shaded region showing 1 standard error taken over 36 independent trials. We generated random 2d functions living in a RBF Reproducing Kernel Hilbert Space (RKHS) with parameter $\sigma$ varying. We evaluate the function at $400$ points taken in a grid of $[0,1]^2$. As $\sigma$ decreases and the effective dimension increases, the performance of all methods becomes similar to uniform sampling. Conversely, when $\sigma$ is large and the dimension decreases, the active methods improve markedly upon uniform sampling. In all cases, `GRAILS` matches or slightly exceeds the performance of the baseline methods.

**Kernel Methods with Prior Knowledge:** In many practical settings, prior knowledge is available to practitioners, e.g., of the form $f^*(x) \geq f^*(x')$ for some pairs $x, x' \in X$. It is straightforward to incorporate into our algorithm any prior knowledge that can be expressed in terms of a polyhedron in the output space $\mathbb{R}^n$. By contrast, while there are a variety of approaches to incorporate constraints into GP regression methods [30], it is not immediately clear how to use constrained GP models in adaptive sampling methods. In this experiment, we use $x_1, \ldots, x_n \in [0,6]$ equally spaced with $n = 250$. We use the Gaussian RBF kernel with $\sigma = 0.075$. For each of the 60 trials, we draw a random function from the RKHS. We vary the number of random pairwise constraints given to `GRAILS` and `OFUL`. Figure 2 depicts the average number of samples required to achieve a simple regret of $0.005$ and shows that as `GRAILS` and `OFUL` obtain more prior knowledge, their performance improves.

**Convex Functions:** In this experiment, the algorithms use the function class $\mathcal{F}$ consisting of all convex functions. We generate 300 points in an equally spaced grid on the interval $[0,1]$. $f^*(x) = 5(x - x_{\min})^2$ where for each of the 30 trials $x_{\min}$ is drawn uniformly at random from $[0,1]$. We only consider `GRAILS`, `OFUL`, and `UNIF`. Figure 3 depicts the average number of samples that each algorithm uses to obtain an $\epsilon$-accurate solution, showing that `GRAILS` does slightly better than `OFUL`, while, as expected, the sample complexity of `UNIF` blows up as $\epsilon$ decreases.

# 8  Conclusion

Our work leaves many open questions. First, while `GRAILS` is computationally efficient as evidenced by our experiments, it becomes computationally burdensome as $n$ grows very large. Further work is required to scale `GRAILS` to the regime where $n$ is extremely large (e.g., $n \approx 10,000$). Second, we have designed computationally efficient algorithms that match the minimax lower bound up to logarithmic factors in the *noiseless setting* and it remains an open question how to achieve near minimax optimality in the noisy setting. Finally, it is an important future direction to develop computationally efficient and near minimax optimal algorithms in the continuous output setting.

## Acknowledgments and Disclosure of Funding

Kevin Jamieson was supported in part by the NSF TRIPODS II grant DMS 2023166. This work was partially supported by AFOSR grant FA9550-18-1-0166.

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
