$$\boxed{\begin{array}{l}
P_1 \longleftarrow \pi, \mathcal{R}_1 \longleftarrow \mathcal{R}, k \longleftarrow 1, t_1 = 1, O_1 \longleftarrow \emptyset; \\
\textbf{for } t = 1, 2, \ldots \textbf{ do} \\
\quad \text{Let } I_t \in \arg\max_{i \in [n] \setminus O_t} \min_{y \in \mathcal{Y}} \mathbb{P}_{r \sim P_k}(r \in \mathcal{R}_{t,\epsilon}(x_i) \cup \mathcal{R}_t(x_i, y)); \\
\quad \text{Query } x_{I_t} \text{ and observe } y_{I_t} \text{ and set } O_{t+1} \longleftarrow O_t \cup \{I_t\} ; \\
\quad \text{Let } \mathcal{R}_{t+1} \longleftarrow \mathcal{R}_t \setminus (\mathcal{R}_{t,\epsilon}(x_{I_t}) \cup \mathcal{R}_t(x_{I_t}, y_{I_t})); \\
\quad \textbf{if } \mathbb{P}_{r \sim P_k}(r \in \mathcal{R}_{t+1}) \leq \frac{1}{2n} \textbf{ then} \\
\quad\quad k \longleftarrow k + 1; \\
\quad\quad t_k \longleftarrow t + 1, P_{k+1} \longleftarrow \frac{1}{n - |O_{t_k}|} \sum_{l \in [n] \setminus O_{t_k}} \pi_{C_l(O_{t_k})}; \\
\quad \textbf{if } STOP(\mathcal{F}_\mathcal{R}, O_t) \textbf{ then} \\
\quad\quad \textbf{return } \arg\min_{i \in O_t} y_i
\end{array}}$$

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

$$\text{s.t. } \exists j \in [n] : \{f \in \mathcal{F} : f(x_i) = g(x_i) \, \forall i \in [I]\} \subset \{f \in \mathcal{F} : j \in \mathrm{argmin}_{i \in [n]} f(x_i)\}.$$

We occasionally write $v_{loss}^*$ instead of $v_{loss}^*(\mathcal{F})$ when the context leaves no ambiguity. In words, $v_{loss}^*$ is the loss that must be incurred (up to the penultimate round) for some scoring function $g : \mathcal{X} \mapsto \mathcal{Y}$ in order to identify the best arm. Next, we show that $v_{loss}^*$ is a minimax lower bound.

**Theorem 3.** *For all $\mathcal{F} \subset \mathcal{Y}^\mathcal{X}$, $v_{loss}^*(\mathcal{F}) \leq \Lambda_{loss}(\mathcal{F})$.*

The following Proposition gives an upper bound of $v_{loss}^*$ in terms of $v_{best}^*$.

**Proposition 6.** *For all $\mathcal{F} \subset \mathcal{Y}^\mathcal{X}$, $v_{loss}^*(\mathcal{F}) \leq v_{best}^*(\mathcal{F}) \max_{y \in \mathcal{Y}} y$.*

In words, this Proposition reflects that one strategy for cumulative loss minimization is to minimize the number of queries to identify the best arm, ignoring the losses.

```
P_1 ⟵ π, R_1 ⟵ R, k ⟵ 1, t_1 = 1, O_1 ⟵ ∅;
for t = 1, 2, . . . do
    Let I_t ∈ argmin_{i∈[n]\O_t} max_{y∈Y} y / P_{r∼P_k}(r∈R_t(x_i)∪R_t((x_i,y)));
    Query x_{I_t} and observe y_{I_t} and set O_{t+1} ⟵ O_t ∪ {I_t} ;
    Let R_{t+1} ⟵ R_t \ (R_t(x_{I_t}) ∪ R_t((x_{I_t}, y_{I_t})));
    if P_{r∼P_k}(r ∈ R_{t+1}) ≤ 1/2n then
        k ⟵ k + 1;
        t_k ⟵ t + 1, P_{k+1} ⟵ 1/(n−|O_{t_k}|) Σ_{l∈[n]\O_{t_k}} π_{C_l(O_{t_k})};
    if STOP(F_R, O_t) then
        return argmin_{i∈O_t} y_i
```

**Algorithm 2: GRAILS for Loss Minimization**

## 6.1 The Algorithm and Upper Bound

Algorithm 2 is similar to Algorithm 1: it is greedy and operates in phases, sampling from the mixture $P_k$ in the $k$th phase due to the nonconvexity of the version space. The main difference is the objective for selecting $I_t$: it queries the arm $I_t$ that for the worst case $y \in \mathcal{Y}$ minimizes the ratio of the loss incurred in round $t$ and the volume of the functions removed under the measure $P_k$. This objective is inspired by the greedy algorithm for weighted set cover [28].

Define $S_{\min} = \text{argmin}_{S_f: f \in \mathcal{F}_\mathcal{R}, \mathbb{P}_\pi(S_f) > 0} \mathbb{P}_\pi(S_f)$. $S_{\min}$ is the subset of $\mathcal{F}_\mathcal{R}$ in the partition that has the least nonzero probability under $\pi$. Recall $\mathcal{R}$ is a set of regression functions.

**Theorem 4.** *Fix $\mathcal{R}$. Algorithm 2 identifies $\text{argmin}_{i\in[n]} f^*(x_i)$ after incurring a loss of at most*

$$\widetilde{O}(\Lambda_{loss}(\mathcal{F}_\mathcal{R})) = 2(v^*_{loss}(\mathcal{F}_\mathcal{R}) + max_{i\in[n], f\in\mathcal{F}_\mathcal{R}} f(x_i)) \log(n) \log\left(\frac{1}{\mathbb{P}(S_{min})}\right).$$

The above guarantee is optimal up to a multiplicative factor of $\log(n)\log(\frac{1}{\mathbb{P}(S_{\min})})$ and an additive factor of $max_{i\in[n], f\in\mathcal{F}} f(x_i)$, which is typically of lower order. We also give an algorithm that enumerates the function class and therefore is not efficient when $|\mathcal{F}_\mathcal{R}|$ is exponential in problem-dependent parameters, but has a stronger guarantee. Due to space constraints and its similarity to Algorithm 2, we defer its presentation to the supplementary material and provide the result here.

**Theorem 5.** *Fix $\mathcal{F}$. Algorithm 2 identifies $\text{argmin}_{i\in[n]} f^*(x_i)$ after incurring a loss of at most*

$$\widetilde{O}(\Lambda_{loss}(\mathcal{F})) = 2(v^*_{loss}(\mathcal{F}) + max_{i\in[n], f\in\mathcal{F}} f(x_i)) \log(|\mathcal{F}|).$$

Theorems 4 and 5 are the first algorithms for cumulative loss minimization that match the minimax lower bound up to logarithmic factors. As a corollary, we obtain results for *regret minimization*, in which the goal is to identify $\text{argmin}_{i\in[n]} f^*(x_i)$ while minimizing the regret $\sum_{t=1}^T f^*(x_{I_t}) - \min_{i\in[n]} f^*(x_i)$ incurred where $T$ is the round that the agent identifies $\text{argmin}_{i\in[n]} f^*(x_i)$.

**Remark 1.** *If $\min_j f(x_j) = 0$ for all $f \in \mathcal{F}$, then loss minimization is equivalent to regret minimization. In this case, Theorem 5 gives nearly optimal minimax bounds for the regret minimization metric. Furthermore, if $\min_j f(x_j) = \min_j f'(x_j) =: opt$ for all $f, f' \in \mathcal{F}$, then by subtracting opt from each $f$, we reduce to the setting where $\min_j f(x_j) = 0$ for all $f \in \mathcal{F}$ and obtain a nearly optimal minimax bound.*

To the best of our knowledge, *these are the first regret bounds that match the minimax lower bound up to logarithmic factors for a large and general class of function classes.*

**Comparison to [11].** Next, we compare our algorithm in Theorem 5 to the *computationally inefficient* algorithm from [11] for regret minimization in the noiseless setting. Their regret scales as $O(\Delta_{\max}\text{HD}(\mathcal{F})\ln(|\mathcal{F}|))$, ignoring lower order terms. The following Proposition shows that our cumulative regret bound is never worse than theirs by more than a polylogarithmic factor. Let $y_* = \text{argmin}_{y\in\mathcal{Y}} y$ and $\Delta_{\max} = \max_{y\in\mathcal{Y}} y - y_*$ and $\Delta_{\min} = \min_{y\in\mathcal{Y}:y\neq y_*} y - y_*$.

**Proposition 7.** *Let $\mathcal{F}$ such that $v^*_{best}(\mathcal{F}) \geq 1$. There exists a universal constant $c > 0$ such that for large enough $T_0$, if Algorithm 3 is run for $T_0$ rounds, the regret of Algorithm 3 is bounded above by*

$$cv^*_{best}(\mathcal{F})\ln(|\mathcal{F}|)\Delta_{max} \leq c\text{HD}(\mathcal{F})\ln(|\mathcal{F}|)^2\Delta_{max}.$$

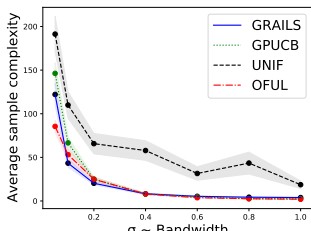
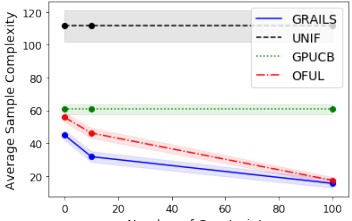
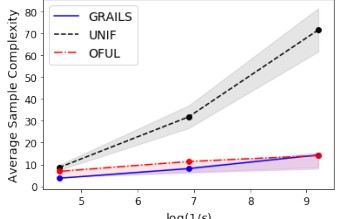

Figure 1: Varying $\sigma$  Figure 2: Prior Knowledge  Figure 3: Convex Functions

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

# Contents

## A Suboptimality of Agnostic Algorithms Proofs

### A.1 Active Classification

*Proof of Proposition 1.* Let $m \in \mathbb{N}$ and $\ell = m + 1$. Note $n = m\ell + m$. Define for all $i \in [m]$

$$f_i(x_j) = \begin{cases} 1 & j \in \{\ell(i-1) + 1, \ldots, i\ell\} \cup \{m\ell + 1, \ldots, m\ell + i\} \\ -1 & \text{o/w} \end{cases}$$

Further, define $f_0(x_i) = -1$ for all $i \in [n]$. Let $\mathcal{F} = \{f_0, f_1, \ldots, f_m\}$. This instance essentially couples disjoint sets and thresholds.

Suppose that $y_i = f_0(x_i)$ for all $i$, so that $f_0 = f^*$ (and note that the realizability assumption holds).

Note that if an algorithm assumes realizability, then it suffices to perform binary search on $x_{m\ell+1}, \ldots, x_{m(l+1)}$ (which is a thresholds instance) and this procedure terminates and outputs $f_0$ after $O(\log(m))$ queries.

Next, we provide a lower bound for any $\delta$-agnostic algorithm wrt active classification on this instance. By corollary 1, any $\delta$-agnostic algorithm wrt active classification requires

$$\frac{\log(1/\delta)}{\log(n/\delta)} \phi^*$$

samples in expectation, so it suffices to lower bound $\phi^*$. Define the alternative instances for all $i \in [m]$.

$$\bar{y}_j^{(i)} = \begin{cases} 1 & \text{if } j \in \{\ell(i-1) + 1, \ldots, i\ell\} \\ y_j & \text{if otherwise} \end{cases}$$

Note that since $\ell = m + 1$, we have that

$$\sum_{i=1}^n \mathbb{1}\{f_j(x_i) \neq \bar{y}_i^{(j)}\} = j < m + 1 = \sum_{i=1}^n \mathbb{1}\{f_0(x_i) \neq \bar{y}_i^{(j)}\}$$

and thus $\bar{y}^{(j)}$ is an alternative instance. Therefore, we have that

$$\phi^* = \min_\lambda \max_{\tilde{y} \in \{-1,1\}^n : \text{argmin}_{f \in \mathcal{F}} \sum_{i=1}^n \mathbb{1}\{\tilde{y}_i \neq f(x_i)\}} \frac{1}{\sum_{i=1}^n \lambda_i \mathbb{1}\{y_i \neq \tilde{y}_i\}}$$

$$\geq \min_\lambda \max_{i \in [m]} \frac{1}{\sum_{j \in \{\ell(i-1)+1, \ldots, i\ell\}} \lambda_j}$$

The minimizing $\lambda$ puts $\frac{1}{\sum_{j \in \{\ell(i-1)+1, \ldots, i\ell\}} \lambda_j} = C$ for all $i \in [m]$ for some constant $C \in \mathbb{R}$. We also have that

$$1 = \sum_{i=1}^{m\ell} \lambda_i = \sum_{i=1}^m \sum_{j \in \{\ell(i-1)+1, \ldots, i\ell\}} \lambda_j = \sum_{i=1}^m \frac{1}{C} = \frac{m}{C}$$

Therefore, we have shown that $\phi^* \geq m = \Omega(\sqrt{n})$. This completes the proof.

$\square$

Recall that we say an algorithm $\mathcal{A}$ is $\delta$-agnostic wrt $\mathcal{F}$ if for any $y \in \mathcal{Y}^n$, $\mathcal{A}$ outputs $\text{argmin}_{f \in \mathcal{F}} \sum_{i=1}^n \mathbb{1}\{y_i \neq f(x_i)\}$ with probability at least $1 - \delta$.

We define the *Gaussian Active Classification problem* as follows: when arm $i$ is pulled, $y_i + \eta$ where $\eta \sim N(0, 1)$ is observed. The goal is to identify $\text{argmin}_{f \in \mathcal{F}} \sum_{i=1}^n \mathbb{1}\{y_i \neq f(x_i)\}$ We call a pair $(y, \mathcal{F})$ where $y \in \{-1, 1\}^n$ an *instance*.

**Proposition 11.** *Let $\delta \in (0, \frac{1}{n})$. Fix $\mathcal{Y} = \{-1, 1\}$, $\mathcal{F} \subset \mathcal{Y}^{\mathcal{X}}$, and $y \in \{-1, 1\}$. If there exists a $\delta$-agnostic algorithm wrt $\mathcal{F}$ for noiseless active classification that achieves an expected sample complexity of $\tau$ on the instance $(y, \mathcal{F})$, then there exists a $2\delta$-agnostic algorithm wrt $\mathcal{F}$ for Gaussian active classification that achieves an expected sample complexity of $c \log(n/\delta)\tau$ samples on the instance $(y, \mathcal{F})$, where $c > 0$ is a universal constant.*

*Proof.* We perform a reduction by constructing a new algorithm $\mathcal{A}'$ for the Gaussian active classification problem instantiated by $\mathcal{F}$. Define $\mathcal{A}'$ as follows: at round $t$, pull the arm $I_t$ that $\mathcal{A}$ would choose to pull in the noiseless active classification problem $O(\log(n/\delta))$ times. If the resulting mean is positive, then tell $\mathcal{A}$ that $\widehat{y}_{I_t} = 1$ otherwise tell $\mathcal{A}$ that $\widehat{y}_{I_t} = -1$. If the algorithm $\mathcal{A}$ terminates and outputs an $f \in \mathcal{F}$, then output $f$ and terminate.

Define $f^* := \mathrm{argmin}_{f \in \mathcal{F}} \sum_{i=1}^n \mathbb{1}\{y_i \neq f(x_i)\}$ and let $\widehat{f}$ be the classifier returned by $\mathcal{A}'$. Note that

$$\mathbb{P}(\widehat{f} \neq f^*) \leq \mathbb{P}(\{\exists i \in [n] : \widehat{y}_i \neq y_i\} \cup \widehat{f} \neq f^*)$$
$$\leq 2\delta$$

using standard subGaussian bounds and the assumption that $\mathcal{A}$ is $\delta$-agnostic. Thus, $\mathcal{A}'$ is $2\delta$-agnostic algorithm wrt $\mathcal{F}$ for Gaussian active classification.

Let $T$ denote the (random) number of queries taken by $\mathcal{A}'$ on the instance $(y, \mathcal{F})$. Define the event $\Sigma = \{\widehat{y}_i \neq y_i \forall i \in [n]\}$. We have that

$$\mathbb{E}[T] = \mathbb{E}[T|\Sigma]\mathbb{P}(\Sigma) + \mathbb{E}[T|\Sigma^c]\mathbb{P}(\Sigma^c) \leq \tau \log(n/\delta) + n \log(n/\delta)\frac{1}{n} \leq 2\tau \log(n/\delta)$$

$\square$

**Corollary 1.** *Let $y \in \{-1, 1\}^n$, and $\mathcal{F} \subset \{-1, 1\}^{\mathcal{X}}$, and define $f^* := argmin_{f \in \mathcal{F}} \sum_{i=1}^n \mathbb{1}\{y_i \neq f(x_i)\}$. Let $\mathcal{A}$ be a $\delta$-agnostic algorithm wrt $\mathcal{F}$ on the active classification problem. Suppose $\mathcal{A}$ an expected sample complexity of $\mathbb{E}[\tau_y]$. Then,*

$$\mathbb{E}[\tau_y] \geq c\frac{\log(1/4.8\delta)}{\log(n/\delta)}\phi^*$$

*where*

$$\phi^* = \min_{\lambda} max_{\tilde{y} \in \{-1,1\}^n : argmin_{f \in \mathcal{F}} \sum_{i=1}^n \mathbb{1}\{\tilde{y}_i \neq f(x_i)\} \neq f^*} \frac{1}{\sum_{i=1}^n \lambda_i \mathbb{1}\{y_i \neq \tilde{y}_i\}}$$

*Proof.* Since $\mathcal{A}$ is $\delta$-agnostic wrt $\mathcal{F}$ for active classification, by Proposition 11, we have that there exists a $2\delta$-agnostic algorithm $\mathcal{A}'$ wrt $\mathcal{F}$ for Gaussian active classification and that it has expected sample complexity of at most $c \log(n/\delta)\mathbb{E}[\tau_\theta]$. Define $\nu_{\tilde{y}_i} = N(\tilde{y}_i, 1)$. Using the Transportation Lemma from [13], since $\mathcal{A}'$ is $2\delta$-agnostic wrt $\mathcal{F}$, we have that for any $\tilde{y} \in \{-1, 1\}^n$ such that $argmin_{f \in \mathcal{F}} \sum_{i=1}^n \mathbb{1}\{\tilde{y}_i \neq f(x_i)\} \neq f^*$,

$$\sum_{i=1}^n \mathbb{E}[T_i] \mathbf{KL}(\nu_{y_i}, \nu_{\tilde{y}_i}) \geq \log(\frac{1}{4.8\delta})$$

where $T_i$ denotes the number of times that arm $i$ is pulled. Defining $\lambda_i = \frac{\mathbb{E}[T_i]}{\sum_j \mathbb{E}[T_j]}$ and noting that

$$\mathbf{KL}(\nu_{y_i}, \nu_{\tilde{y}_i}) = \begin{cases} 2 & y_i = y_i \\ 0 & \text{otherwise} \end{cases}$$

this implies that

$$\sum_{i=1}^n \mathbb{E}[T_i] \geq \frac{1}{2}\log(1/4.8\delta)\frac{1}{\sum_{i=1}^n \lambda_i \mathbb{1}\{y_i \neq \tilde{y}_i\}}$$

Maximizing over $\tilde{y} \in \{-1, 1\}^n$ such that $argmin_{f \in \mathcal{F}} \sum_{i=1}^n \mathbb{1}\{\tilde{y}_i \neq f(x_i)\} \neq f^*$ and then minimizing over $\lambda \in \mathbf{\Delta}_n$, we have that

$$c\mathbb{E}[\tau_y] \log(n/\delta)$$
$$\geq \sum_{i=1}^n \mathbb{E}[T_i]$$
$$\geq \frac{1}{2}\log(1/4.8\delta) \min_{\lambda} max_{\tilde{y} \in \{-1,1\}^n : argmin_{f \in \mathcal{F}} \sum_{i=1}^n \mathbb{1}\{\tilde{y}_i \neq f(x_i)\} \neq f^*} \frac{1}{\sum_{i=1}^n \lambda_i \mathbb{1}\{y_i \neq \tilde{y}_i\}}.$$

Rearranging the above bound give the result. $\square$

## A.2 Best Arm Identification

Recall that for $k \in \mathbb{N}$ and $\delta \in (0, 1)$, we say an algorithm $\mathcal{A}$ is $(\delta, k)$-agnostic if for any $y \in \mathcal{Y}^n$ such that $\min_{f \in \mathcal{F}} \sum_{i=1}^{n} \mathbb{1}\{f(x_i) \neq y_i\} \leq k$, $\mathcal{A}$ identifies $\operatorname{argmin}_{i \in [n]} y_i$ with probability at least $1 - \delta$.

We define the *Gaussian Best Arm Identification problem* as follows: when arm $i$ is pulled, $y_i + \eta$ where $\eta \sim N(0, 1)$ is observed. The goal is to identify $\operatorname{argmin}_{i \in [n]} y_i$. We call a pair $(y, \mathcal{F})$ where $y \in \{-1, 1\}^n$ an *instance*.

**Proposition 12.** *Let $\delta \in (0, \frac{1}{n})$. Fix $\mathcal{Y} = \{-1, 1\}$, $\mathcal{F} \subset \mathcal{Y}^{\mathcal{X}}$, and $y \in \{-1, 1\}$. Let $\mathcal{A}$ be a $(\delta, k)$-agnostic algorithm wrt $\mathcal{F}$ for noiseless best arm identification that achieves an expected sample complexity of $\tau$ on the instance $(y, \mathcal{F})$ and has a probability $p_i$ of sampling $i \in [n]$. Then there exists a $(2\delta, k)$-agnostic algorithm wrt $\mathcal{F}$ for Gaussian best arm identification that achieves an expected sample complexity of $c \log(n/\delta)\tau$ samples on the instance $(y, \mathcal{F})$, where $c > 0$ is a universal constant, and has a probability $p_i$ of sampling $i \in [n]$.*

*Proof.* The proof is identical to the proof of Proposition 11. The probability $p_i$ is the same for Gaussian best arm identification setting by the construction of $\mathcal{A}'$ in the proof of Proposition 11. $\square$

*Proof of Proposition 3.* Let $\mathcal{Y} = \{w_1, \ldots, w_L\}$ with $w_1 < w_2 < \ldots < w_L$. Let $\mathcal{F} = \{f^*\}$ with $f^* : \mathcal{X} \mapsto \{w_1, \ldots, w_L\}$ such that for all $i \in [n]$ $f^*(x_i) \in \{w_2, \ldots, w_{L-1}\}$ and $\operatorname{argmin}_{i \in [n]} f^*(x_i) = \{i_*\}$.

**Step 0: there exists an arm that is not queried with constant probability.** Let $\theta \in \{w_1, \ldots, w_L\}^n$ such that $\theta_i = f^*(x_i)$ for all $i \in [n]$. Let $T_i$ denote the number of times that arm $i$ is queried (a random variable). Towards a contradiction, suppose that $\mathbb{E}_\theta[\sum_{i=1}^{n} T_i] < n/8$. Since

$$n/8 > \mathbb{E}_\theta[\sum_{i=1}^{n} T_i] \geq \min_{i \in n} n \mathbb{E}_\theta[T_i].$$

Thus, there exists $i_0 \in [n]$ such that $1/8 > \mathbb{E}_\theta[T_{i_0}] \geq \mathbb{P}_\theta(T_{i_0} > 0)$.

**Step 1: Reduction**

By Proposition 12, since $\delta < 1/n$, there exists a $(2\delta, 1)$-agnostic algorithm $\mathcal{A}'$ wrt $\mathcal{F}$ for Gaussian best arm identification that achieves an expected sample complexity of $c \log(n/\delta)\tau$ samples on the instance $(y, \mathcal{F})$ and the probability of querying $i_0$ is at most $1/8$.

**Step 2.1: Constructing a bad alternative** Let $\widehat{i}$ denote the arm returned at the end of the game. Suppose $i_0 \notin \operatorname{argmin}_{j \in [n]} f^*(x_j)$. Then, since $\delta < \frac{1}{40}$ and $\mathcal{A}'$ is $(2\delta, 1)$-agnostic wrt $\mathcal{F}$ for Gaussian best arm identification

$$\mathbb{P}_\theta(\widehat{i} \neq i_0) \geq 19/20.$$

Define $B = \{T_{i_0} = 0\} \cap \{\widehat{i} \neq i_0\}$. Then,

$$\mathbb{P}_\theta(B^c) \leq \mathbb{P}_\theta(T_{i_0} > 0) + \mathbb{P}_\theta(\widehat{i} = i_0) \leq 1/8 + 1/20 \leq 1/5.$$

Define $\tilde{\theta} \in \{w_1, \ldots, w_L\}^n$ by

$$\tilde{\theta}_j = \begin{cases} f^*(x_j) & j \neq i_0 \\ w_1 & j = i_0 \end{cases}.$$

Note that since by assumption $f^*(x_i) \in \{w_2, \ldots, w_{L-1}\}$ for all $i \in [n]$, under $\tilde{\theta}$, $i_0$ is the unique minimizer.

Define

$$\widehat{\mathrm{kl}}_{i, T_i} = \sum_{s=1}^{T_i} \log(f_\theta(Z_s)/f_{\tilde{\theta}}(Z_s))$$

where $Z_s$ is the observation on the $s$th pull of arm $i$, $f_{\tilde\theta}$ denotes the density of the distribution associated with arm $i$ under $\tilde\theta$, specifically $N(\tilde\theta_i, 1)$. Then, by the change of measure identity (Lemma 18) from [13],

$$
\begin{aligned}
\mathbb{P}_{\tilde\theta}(\widehat{i} \neq i_0) &\geq \mathbb{P}_{\tilde\theta}(B) \\
&= \mathbb{E}_\theta[\mathbb{1}\{B\} \exp(-T_i \widehat{\mathrm{kl}}_{i,T_i})] \\
&= \mathbb{P}_\theta(B) \\
&\geq \frac{4}{5}.
\end{aligned}
$$

where we used the fact that the only difference between problem $\theta$ and problem $\tilde\theta$ is the mean of $i$th arm and on the event $A$, $T_i = 0$. But, this is a contradiction since under $\tilde\theta$, $i_0$ is the minimizer, $\sum_{i=1}^n \mathbb{1}\{\theta_i \neq \tilde\theta_i\} = 1$, and the algorithm $\mathcal{A}$ is $(2\delta, 1)$-agnostic with $\delta \in (0, 1/20)$.

**Step 2.2:**

Suppose that $i_0 = i_*$. Define $\bar\theta \in \{w_1, \ldots, w_L\}^n$ by

$$
\bar\theta_j = \begin{cases} f^*(x_j) & j \neq i_* \\ w_L & j = i_* \end{cases}.
$$

Define the events $\Sigma_1 = \{T_{i_*} = 0\}, \Sigma_2 = \{i_* = \widehat{i}\}, \Sigma = \Sigma_1 \cap \Sigma_2$. Note that

$$
\begin{aligned}
\mathbb{P}_\theta(\Sigma^c) &\leq \mathbb{P}_\theta(\Sigma_1^c) + \mathbb{P}_\theta(\Sigma_2^c) \\
&= \mathbb{P}_\theta(T_{i_*} > 0) + \mathbb{P}_\theta(\widehat{i}_* \neq \widehat{i}) \\
&\leq 1/8 + 2\delta \\
&\leq 1/5
\end{aligned}
$$

This implies by a similar argument to the one made in Step 2.1 that

$$
\begin{aligned}
\mathbb{P}_{\bar\theta}(\Sigma_2) &\geq \mathbb{P}_{\bar\theta}(\Sigma) = \mathbb{E}_\theta[\mathbb{1}\{\Sigma\} \exp(-T_{i_*} \widehat{\mathrm{kl}}_{i,T_{i_*}})] \\
&= \mathbb{P}_\theta(\Sigma) \\
&\geq 4/5.
\end{aligned}
$$

But, this contradicts the assumption that $\mathcal{A}$ is $(2\delta, 1)$-agnostic since $\sum_{i=1}^n \mathbb{1}\{\theta_i \neq \bar\theta_i\} = 1$ and $i_*$ is suboptimal in the instance given by $\bar\theta$ and $\mathcal{A}$ therefore makes a mistake with probability at least $4/5 > \delta$, a contradiction.

$\square$

### A.3  Regret Minimization

*Proof of Proposition 2.* Consider an agnostic algorithm $\mathcal{A}$ for noiseless regret minimization. We assume there exist $n$ arms $x_1, \cdots, x_n =: \mathcal{X}$ and an unknown true function $f^*$ in a class $\mathcal{F} : \mathcal{X} \to \mathbb{R}$. In the noiseless regime, we assume that pulling $x_i$ returns $f^*(x_i)$. In the noisy regime, we assume that pulling $x_i$ returns $f^*(x_i) + \eta$ where $\eta$ is a 1 sub-Gaussian random variable.

In the noiseless regime, we assume that $\mathcal{A}$ achieves a bounded regret $R_\mathcal{A}(f^*, \mathcal{X}) < \infty$ independent of the horizon. This is a mild assumption as noise is not present in the responses. Indeed, the simple algorithm that samples each $x_i$ once and then chooses the $x^*$ achieving minimal loss $f(x)$ thereafter has bounded regret. Additionally, we may assume without loss of generality that $\mathcal{A}$ never re-samples a suboptimal arm as its value is known.

**Step 1**: Reduction from regret minimization in the noiseless regime to the noisy regime.

By [31], an algorithm $\mathcal{A}'$ for the noisy regime is said to be *consistent* if its regret at time $T$, $R_{f^*}^{\mathcal{A}'}(T)$ obeys

$$
\limsup_{T \to \infty} \frac{\log(R_{f^*}^{\mathcal{A}'}(T))}{\log(T)} \leq 0.
$$

We say that $f^*$ is $\epsilon$-separated on $x_1, \cdots, x_n$ if $\min_{i,j:f^*(x_i) \neq f^*(x_j)} |f^*(x_i) - f^*(x_j)| \geq \epsilon$. Next, we show how to use algorithm $\mathcal{A}$ in the noiseless regime to form a consistent algorithm $\mathcal{A}'$ for 1-separated instances. For simplicity, we assume that $f^*$ takes values over the integers. Let $T > 0$ denote the time horizon and assume that $\max|f(x_i) - f(x_j)| \leq B$. At each time $t$, $\mathcal{A}'$

1. queries $\mathcal{A}$ for an arm $x_i$ to sample next

2. samples $x_i$ $c\log(nT)$ times for sufficiently large constant $c$.

3. Average the losses and round the number to the nearest integer.

4. Pass the rounded average to $\mathcal{A}$.

Next, we argue that $\mathcal{A}'$ is consistent. For sufficiently large $c$ and a union bound over the $n$ arms, with probability $1 - 1/T$, we have that $f^*(x_i) = R\left(\frac{1}{c\log(nT)}\sum_{j=1}^{c\log(nT)} f(x_i) + \eta_j\right)$ where $\eta_j$ is the noise on the $j^{\text{th}}$ sample of $x_i$ and $R(\cdot)$ is a rounding function that rounds to the nearest integer value. In this case, since $\mathcal{A}$ has bounded regret and never re-pulls suboptimal arms, the total regret is $cR_A(f^*, \mathcal{X})\log(nT)$. Otherwise, with probability $1/T$, $\mathcal{A}'$ receives regret no worse than $BT$. Hence the total regret is bounded by

$$R_{f^*}^{\mathcal{A}'}(T) \leq cR_A(f^*, \mathcal{X})\log(nT) + B.$$

Therefore,

$$\limsup_{T \to \infty} \frac{\log(R_{f^*}^{\mathcal{A}'}(T))}{\log(T)} = \limsup_{T \to \infty} \frac{\log\left(cR_A(f^*, \mathcal{X})\log(nT) + B\right)}{\log(T)}$$
$$= \limsup_{T \to \infty} \frac{\log\left(cR_A(f^*, \mathcal{X})\log(nT)\right)}{\log(T)}$$
$$= 0$$

which implies that $\mathcal{A}'$ is consistent.

**Step 2**: Instance dependent regret bounds for consistent algorithms.

For any consistent algorithm $\mathcal{A}'$, by Corollary 2 of [31]

$$C(f^*, \mathcal{X}) \leq \limsup_{n \to \infty} \frac{R_{f^*}^{\mathcal{A}'}(T))}{\log(n)} = \limsup_{n \to \infty} \frac{cR_A(f^*, \mathcal{X})\log(nT) + B}{\log(n)} = cR_A(f^*, \mathcal{X})$$

where $C(f^*, \mathcal{X})$ is the optimal solution to

$$\min_{\alpha \in (0,\infty]^{\mathcal{X}}} \sum_{x \in \mathcal{X}^-} \alpha_i \Delta_i \ \text{ s.t } 1/\alpha_i \leq \Delta_i^2/2 \ \forall x_i \in \mathcal{X}^-$$

where $\Delta_i := f^*(x_i) - \min_{x \in \mathcal{X}} f^*(x)$ and $\mathcal{X}^- := \{x \in \mathcal{X} : \Delta_x > 0\}$. Choosing $\alpha_i = 2/\Delta_i^2$ to saturate the bound, we have that

$$C(f^*, \mathcal{X}) = \sum_{x \in \mathcal{X}^-} \frac{2}{\Delta_i}.$$

**Step 3**: Constructing an instance where $R_A(f^*, \mathcal{X})$ is large.

Fix $n \in \mathbb{N}$ and consider a set of points $\mathcal{X} = \{x_1, \cdots, x_{2n}\}$. We consider a family of $n$ functions $f_1, \cdots, f_n$ mapping from $\mathcal{X}$ to $\{-1, 0, 1\}$. By definition each $f_i$ is 1-separated. Define

$$f_i(x_j) = \begin{cases} 1 & \text{if } j = i \\ 0 & \text{if } j \leq n \text{ and } j \neq i \\ -1 & \text{if } j \in [2n - i, 2n] \end{cases}$$

For any $f_i$, the set of suboptimal arms is given by $\mathcal{X}^- = [2n]\backslash\{i\}$. Choose $f^* \in \{f_1, \cdots, f_n\}$ uniformly. By step 2, we have that $R_A(f^*, \mathcal{X}) \gtrsim (n - 1)$ since $\Delta_i = O(1)$ where we have used the fact that $\mathcal{A}$ was an agnostic algorithm with bounded regret.

Conversely, a realizability-based algorithm that knows the version space $\{f_1, \cdots, f_n\}$ could perform binary search over $x_{n+1}, \cdots, x_{2n}$ and identify the maximizer of $f^*$ in $O(\log(n))$ pulls and achieving regret at most $O(\log(n))$. $\qquad \square$

# B    Construction of Sampling Oracles for Efficient Active Learning

We now show that non-trivial sampling oracles are available for many useful function classes.

**Linear Models:** Define $\mathcal{R} = \{\langle a, \cdot \rangle : a \in \mathbb{R}^d, \|a\| \leq B\}$ where $\|\cdot\|$ is some norm and $B \in \mathbb{R}$. Let $\pi$ denote the uniform distribution on $D := \{a \in \mathbb{R}^d : \|a\| \leq B\}$. Since $D$ a compact convex set, there exists a polynomial time algorithm to sample from $D$ [14]. Now, define $\bar{D} = D \cap \cap_{j \in [m]} \{a : \langle a, w_j \rangle \leq b_j\}$ where $w_j n \mathbb{R}^d$ and $b_j \in \mathbb{R}$. Let $\bar{\pi}$ be the uniform distribution on $\bar{D}$. Note that $D$ has a membership oracle: for a given $x \in \mathbb{R}^d$, one can check whether $x \in D$ by computing $\|x\|$. Since $\bar{D}$ is a convex body and has a membership oracle, there exists a polynomial time algorithm to sample from $\bar{D}$. Finally, note that $\mathrm{uniform}(\bar{D})$ is the desired condition distribution:

$$\mathbb{P}_{z \sim \mathrm{uniform}(\bar{D})}(z \in A) = \frac{\mathbb{P}_{z \sim \mathrm{uniform}(D)}(z \in A)}{\mathbb{P}_{z \sim \mathrm{uniform}(D)}(z \in \bar{D})}.$$

Thus, the sample oracle exists for $\mathcal{R}$.

**Kernel Method:** Let $k : \mathcal{X} \times \mathcal{X} \mapsto \mathbb{R}$ be a kernel and let $\mathcal{R}$ be the associated RKHS. Let $\mathcal{R}_B = \{f \in \mathcal{R} : \|f\|_{\mathcal{R}} \leq B\}$. Fix $x_1, \ldots, x_n \in \mathcal{X}$. By standard arguments in kernel methods (e.g., Representer Theorem [32]), for every $f \in \mathcal{R}_B$, there exists $f' \in \mathcal{R}_B$ such that $f'(\cdot) = \sum_{i=1}^n \alpha_i k(\cdot, x_i)$ and $f'(x_i) = f(x_i)$ for all $i \in [n]$. Thus, it suffices to consider the function class

$$\tilde{\mathcal{R}} = \{\sum_{i=1}^n \alpha_i k(\cdot, x_i) : \|\alpha\|_{K^{1/2}} = \left\| \sum_{i=1}^n \alpha_i k(\cdot, x_i) \right\|_{\mathcal{R}} \leq B\}.$$

Thus, it is enough to sample from the uniform distribution on

$$D = \{\alpha \in \mathbb{R}^n : \|\alpha\|_{K^{1/2}} \leq B\}$$

which reduces to the linear case. Intersecting $D$ with halfspaces is also similar to the linear case.

**Convex Functions:** Let $\mathcal{R} = \{f : \mathcal{X} \mapsto [-B_1, B_1], f \text{ convex}, \max_{x \in \mathcal{X}, g \in \partial f(x)} \|g\|_2 \leq B_2\}$ where $\partial f(x)$ denotes the subdifferential of $f$ at $x$. Let $f \in \mathcal{R}$. Then, $f(x_i) \geq f(x_j) + g_j^\top (x_i - x_j)$ for all $i, j \in [n]$ where $g_j \in \partial f(x_j)$ with $\|g_j\|_2 \leq B_2$. Thus, there exists $\widehat{y}_i = f(x_i)$ and $g_i \in \mathbb{R}^d$ with $\|g_i\|_2 \leq R_2$ for $i = 1, \ldots, n$ such that

$$\widehat{y}_i \geq \widehat{y}_j + g_j^\top (x_i - x_j).$$

Note that $\bar{f}(x_i) = \max_{i=1,\ldots,n} \widehat{y}_i + g_i^\top (x - x_i)$ is convex and $\bar{f}(x_i) = f(x_i)$ for all $i$ [33]. This shows that it suffices to sample from the set $D :=$

$$\{(\widehat{y}_i)_{i=1}^n (g_i)_{i=1}^n) \in \mathbb{R}^{n(d+1)} : \widehat{y}_i \geq \widehat{y}_j + g_j^\top (x_i - x_j) \wedge \widehat{y}_i \in [-B_1, B_1], \|g_i\|_2 \leq B_2 \forall i \in [n]\}.$$

Note that $D$ has a membership oracle: one can efficiently check whether $x \in D$ by checking whether it satisfies the inequalities defining $D$. Since $D$ is convex and compact, and has a membership oracle, one can construct a polynomial-time algorithm for sampling from it. Intersecting $D$ with halfspaces is similar to the linear case. Note that one can construct a sampling oracle in a similar way for $\alpha$-**strongly convex functions**: one need only replace the constraint $\widehat{y}_i \geq \widehat{y}_j + g_j^\top (x_i - x_j)$ with $\widehat{y}_i \geq \widehat{y}_j + g_j^\top (x_i - x_j) + \frac{\alpha}{2} \|x_i - x_j\|_2^2$.

# C    Best Arm Identification Proofs

Here we state and prove a strictly more general result than Theorem 1 for $\epsilon$-good arm identification. Theorem 1 follows as a special case. First, we define:

**Definition 1.** $\Lambda_{best,\epsilon}(\mathcal{F})$ *is the smallest integer $q$ such that there exists some algorithm $\mathcal{A}$ such that for every $f^* \in \mathcal{F}$, $\mathcal{A}$ outputs an element of $\{j \in [n] : f^*(x_j) \leq \min_{i \in [n]} f^*(x_i) + \epsilon\}$ after at most $q$ queries.*

**Theorem 6.** *For any $\mathcal{F} \subset \mathcal{Y}^{\mathcal{X}}$, $\upsilon_{best,\epsilon}^*(\mathcal{F}) \leq \Lambda_{best,\epsilon}(\mathcal{F})$.*

*Proof of Theorem 6.* Consider an algorithm $\mathcal{A}$ that for all $f^* \in \mathcal{F}$ identifies $i_0 \in [n]$ such that $f^*(x_{i_0}) \leq \min_{i \in [n]} f^*(x_i) + \epsilon$ using at most $C(\mathcal{F})$ queries. Towards a contradiction, suppose

$C(\mathcal{F}) < v^*_{best,\epsilon}$. There exists some $g : \mathcal{X} \mapsto \mathcal{Y}$ achieving the maximum in $v^*_{best}$. Let $I \subset [n]$ be the subset of arms queried by $\mathcal{A}$. Then, by definition of $v^*_{best}$, $\forall j \in [n]$ there exist $f, f' \in \mathcal{F}$ such that $f(x_i) = f'(x_i) = g(x_i)$ for all $i \in I$ and either $f(x_j) > \min_{i \in [n]} f(x_i) + \epsilon$ or $f(x_j) > \min_{i \in [n]} f'(x_i) + \epsilon$. Thus, $\mathcal{A}$ cannot distinguish between these two functions and determine the $\epsilon$-good arm. Therefore, if $\mathcal{A}$ outputs $j \in [n]$ after $C(\mathcal{F}) < v^*_{best}$ queries, there exists some $f \in \mathcal{F}$ on which $\mathcal{A}$ would make a mistake, leading to a contradiction. $\qquad\square$

*Proof of Proposition 4.* **The Lower Bound.** Let $F \subset \mathcal{F}$. Define $g : \mathcal{X} \mapsto \mathcal{Y}$ by

$$g(x_i) = \operatorname{argmin}_{y \in \mathcal{Y}} |F(x_i) \cup F((x_i, y))|$$

By definition of $v^*_{best}$, there exists $\tilde{I} \subset [n]$ such that $|\tilde{I}| \leq v^*_{best}$ and

$$\{f \in F : f(x_i) = g(x_i) \forall i \in \tilde{I}\} \subset \{f \in F : j_0 \in \operatorname{argmin}_{k \in [n]} f(x_k)\}$$

Then, defining $I = \tilde{I} \cup \{j_0\}$, we have that

$$F = \{f \in F : \exists i \in I \text{ s.t. } f(x_i) \neq g(x_i) \text{ or } i \in \operatorname{argmin}_{k \in [n]} f(x_k)\}.$$

Therefore,

$$
\begin{aligned}
|F| &= |\{f \in F : \exists i \in I \text{ s.t. } f(x_i) \neq g(x_i) \text{ or } i \in \operatorname{argmin}_{k \in [n]} f(x_k)\}| \\
&\leq \sum_{i \in I} |\{f \in F : f(x_i) \neq g(x_i) \text{ or } i \in \operatorname{argmin}_{k \in [n]} f(x_k)\}| \\
&\leq (v^*_{best} + 1) \max_{i \in I} |\{f \in F : f(x_i) \neq g(x_i) \text{ or } i \in \operatorname{argmin}_{k \in [n]} f(x_k)\}| \\
&\leq (v^*_{best} + 1) \max_{i \in [n]} |\{f \in F : f(x_i) \neq g(x_i) \text{ or } i \in \operatorname{argmin}_{k \in [n]} f(x_k)\}| \\
&= (v^*_{best} + 1) \max_{i \in [n]} \operatorname{argmin}_{y \in \mathcal{Y}} |\{f \in F : f(x_i) \neq y \text{ or } i \in \operatorname{argmin}_{k \in [n]} f(x_k)\}| \\
&= (v^*_{best} + 1) \max_{i \in [n]} \operatorname{argmin}_{y \in \mathcal{Y}} |F((x_i, y)) \cup F(x_i)|
\end{aligned}
$$

Rearranging, we have that

$$\frac{|F|}{\max_{i \in [n]} \operatorname{argmin}_{y \in \mathcal{Y}} |F((x_i, y)) \cup F(x_i)|} \leq v^*_{best} + 1.$$

Since the above inequality holds for any $F \subset \mathcal{F}$, we have that

$$\mathrm{HD}(F) = \sup_{F \subset \mathcal{F}} \frac{|F|}{\max_{i \in [n]} \operatorname{argmin}_{y \in \mathcal{Y}} |F((x_i, y)) \cup F(x_i)|} \leq v^*_{best} + 1$$

This completes the lower bound.

**The Upper Bound.** The proof is essentially identical to the proof from [11]. Let $g : \mathcal{X} \mapsto \mathcal{Y}$ be arbitrary. Consider the algorithm from [11] that sets $\mathcal{F}_1 = \mathcal{F}$, $\mathcal{F}_{t+1} = \mathcal{F}_t \backslash \mathcal{F}_t(x_{I_t}) \cup \mathcal{F}_t((x_{I_t}, g(x_{I_t})))$, and at each round $t$ chooses $I_t \in \arg\max_{i \in [n]} \inf_{y \in \mathcal{Y}} \mathcal{F}_t(x_i) \cup \mathcal{F}_t((x_{I_t}, y))$. Consider round $t \leq c\mathrm{HD}(\mathcal{F}) \ln(|\mathcal{F}|)$ for some positive constant to be determine later. If $|\mathcal{F}_t| = 0$, then we are done. Thus, suppose that $|\mathcal{F}_t| > 1$. By definition, we have that

$$\mathrm{HD}(\mathcal{F}) \leq \gamma(\mathcal{F}_t) \leq \frac{|\mathcal{F}_t(x_{I_t}) \cup \mathcal{F}_t((x_{I_t}, f^*(x_{I_t}))|}{|\mathcal{F}_t|}.$$

Therefore, using $|\mathcal{F}_t| = |\mathcal{F}_{t+1}| + |\mathcal{F}_t(x_{I_t}) \cup \mathcal{F}_t((x_{I_t}, f^*(x_{I_t}))|$, we have that

$$|\mathcal{F}_{t+1}| \leq (1 - \mathrm{HD}(\mathcal{F}))|\mathcal{F}_t|.$$

Unrolling the recurrence, we have that $|\mathcal{F}_{t+1}| \leq (1 - \mathrm{HD}(\mathcal{F}))^t |\mathcal{F}|$. Plugging in $t = c\mathrm{HD}(\mathcal{F}) \ln(|\mathcal{F}|)$ for a large enough positive constant $c$ shows that $\mathcal{F}_t = \emptyset$. Therefore there trivially exists $j \in [n]$ such that for any $f \in \mathcal{F}$ such that $f(x_i) = g(x_i)$ for all $i \in \{I_1, \ldots, I_t\}$, $j \in \operatorname{argmin}_{k \in [n]} f(x_k)$. Thus, by definition of $v^*_{best}$, we have that

$$v^*_{best} \leq c\mathrm{HD}(\mathcal{F}) \ln(|\mathcal{F}|).$$

$\qquad\square$

*Proof of Theorem 2.* For the sake of abbreviation, let $\mathbb{P}(\cdot) := \mathbb{P}_{r\sim\pi}(\cdot)$ and $\mathbb{P}_k(\cdot) := \mathbb{P}_{r\sim P_k}(\cdot)$.

**Step 1:** The Algorithm 1 breaks time into phases $[t_1, t_2], [t_2, t_3], \cdots$. In each phase $k$, the algorithm samples functions from a distinct distribution $P_k$. Consider phase $k$, let $t \in [t_k, t_{k+1}]$. Define $\theta \in \mathcal{Y}^n$ by

$$\theta_i = \begin{cases} y_i & i \in O_t \\ \text{argmin}_{y \in \mathcal{Y}} \mathbb{P}_k(r \in \mathcal{R}_t(x_i) \cup \mathcal{R}_{t,\epsilon}((x_i, y))) & \text{o/w} \end{cases}.$$

By definition of $v_{best,\epsilon}^*$, there exists $\tilde{I} = \{i_1, \ldots, i_{v_{best,\epsilon}^*}\}$ such that there exists $j_0 \in [n]$ satisfying

$$\{r \in \mathcal{R}_t : \mathcal{D}[r(x_i)] = \theta_i \; \forall i \in \tilde{I}\} \subset \{r \in \mathcal{R}_t : \mathcal{D}[r(x_{j_0})] \le \min_{l \in [n]} \mathcal{D}[r(x_l)] + \epsilon\}$$

Define $I = \{i_1, \ldots, i_{v_{best}^*}, j_0\}$. Then, we have that

$$\mathcal{R}_t = \{r \in \mathcal{R}_t : \exists i \in I \text{ s.t. } \mathcal{D}[r(x_i)] \le \min_{l \in [n]} \mathcal{D}[r(x_l)] + \epsilon \text{ or } \mathcal{D}[r(x_i)] \ne \theta_i\}.$$

From this, it follows that

$$\mathbb{P}_k(\mathcal{R}_t) = \mathbb{P}_k(\{r \in \mathcal{R}_t : \exists i \in I \text{ s.t. } \mathcal{D}[r(x_i)] \le \min_{l \in [n]} \mathcal{D}[r(x_l)] + \epsilon \text{ or } \mathcal{D}[r(x_i)] \ne \theta_i\})$$

$$\le \sum_{i \in I} \mathbb{P}_k(\{r \in \mathcal{R}_t : \mathcal{D}[r(x_i)] \le \min_{l \in [n]} \mathcal{D}[r(x_l)] + \epsilon \text{ or } \mathcal{D}[r(x_i)] \ne \theta_i\})$$

$$\le (v_{best,\epsilon}^* + 1)\max_{i \in I} \mathbb{P}_k(\{r \in \mathcal{R}_t : \mathcal{D}[r(x_i)] \le \min_{l \in [n]} \mathcal{D}[r(x_l)] + \epsilon \text{ or } \mathcal{D}[r(x_i)] \ne \theta_i\})$$

$$= (v_{best,\epsilon}^* + 1)\mathbb{P}_k(\{r \in \mathcal{R}_t : \mathcal{D}[r(x_{I_t})] \le \min_{l \in [n]} \mathcal{D}[r(x_l)] + \epsilon \text{ or } \mathcal{D}[r(x_i)] \ne \theta_{I_t}\}) \quad (1)$$

$$\le (v_{best,\epsilon}^* + 1)\mathbb{P}_k(\{r \in \mathcal{R}_t : \mathcal{D}[r(x_{I_t})] \le \min_{l \in [n]} \mathcal{D}[r(x_l)] + \epsilon \text{ or } \mathcal{D}[r(x_i)] \ne y_{I_t}\}) \quad (2)$$

$$= (v_{best,\epsilon}^* + 1)\mathbb{P}_k(\mathcal{R}_{t,\epsilon}(x_{I_t}) \cup \mathcal{R}_t((x_{I_t}, y_{I_t})))$$

where line (1) comes from the definition of $I_t$ and the definition of $\theta$, and the line (2) comes from the definition of $\theta$. Noticing that

$$\mathcal{R}_t = (\mathcal{R}_{t,\epsilon}(x_{I_t}) \cup \mathcal{R}_t((x_{I_t}, y_{I_t}))) \cup (\mathcal{R}_t \setminus (\mathcal{R}_{t,\epsilon}(x_{I_t}) \cup \mathcal{R}_t((x_{I_t}, y_{I_t}))))$$
$$= (\mathcal{R}_{t,\epsilon}(x_{I_t}) \cup \mathcal{R}_t((x_{I_t}, y_{I_t}))) \cup \mathcal{R}_{t+1},$$

we have that

$$\mathbb{P}_k(\mathcal{R}_{t+1}) = \mathbb{P}_k(\mathcal{R}_t) - \mathbb{P}_k(\mathcal{R}_{t,\epsilon}(x_{I_t}) \cup \mathcal{R}_t((x_{I_t}, y_{I_t}))) \le (1 - \frac{1}{v_{best}^* + 1})\mathbb{P}_k(\mathcal{R}_t).$$

Unraveling this recursive statement, we have that

$$\mathbb{P}_k(\mathcal{R}_{t+1}) \le (1 - \frac{1}{v_{best,\epsilon}^* + 1})^{t-t_k}.$$

Thus, we see that in phase $k$ at $t = t_k + s_k$ with $s_k := O(\ln(n)v_{best}^*)$, we have that

$$\mathbb{P}_k(\mathcal{R}_{t_k+s_k}) \le \frac{1}{2n}. \quad (3)$$

**Step 2: Reduction in each version space.** Fix the first round $t$ such that $\mathbb{P}_k(r \in \mathcal{R}_t) \leq \frac{1}{2n}$. Note that $t_{k+1} = t$. Fix $l_0 \in [n] \setminus O_{t_k}$. Then, we have that

$$
\begin{aligned}
\frac{1}{2n} &\geq \frac{1}{n - |O_{t_k}|} \sum_{l \in [n] \setminus O_{t_k}} \mathbb{P}_{f \sim \pi_{C_l(O_{t_k})}}(\mathcal{R}_t) \\
&\geq \frac{1}{n} \sum_{l \in [n] \setminus O_{t_k}} \mathbb{P}_{f \sim \pi_{C_l(O_{t_k})}}(\mathcal{R}_t) \\
&\geq \frac{1}{n} \mathbb{P}_{f \sim \pi_{C_{l_0}(O_{t_k})}}(\mathcal{R}_t) \\
&= \frac{1}{n} \frac{\mathbb{P}(\mathcal{R}_t \cap C_{l_0}(O_{t_k}))}{\mathbb{P}(C_{l_0}(O_{t_k}))} & (4) \\
&= \frac{1}{n} \frac{\mathbb{P}(\mathcal{R}_{t_{k+1}} \cap C_{l_0}(O_{t_k}))}{\mathbb{P}(C_{l_0}(O_{t_k}))} \\
&\geq \frac{1}{n} \frac{\mathbb{P}(C_{l_0}(O_{t_{k+1}}))}{\mathbb{P}(C_{l_0}(O_{t_k}))} & (5)
\end{aligned}
$$

where the equality (4) follows by the definition of conditional probability and in the inequality (5) we used that

$$
C_{l_0}(O_{t_{k+1}}) \subset \mathcal{R}_t \cap C_{l_0}(O_{t_k}).
$$

The above display can be seen since by inspection of the definition for any $t_k \leq t_{k+1}$ and $l \in [n]$,

$$
\begin{aligned}
C_l(O_{t_{k+1}}) &= \{r \in \mathcal{R} : \mathcal{D}[r(x_i)] = y_i \forall i \in O_{t_{k+1}}\} \cap \{\mathcal{D}[r(x_l)] + \epsilon < \mathcal{D}[r(x_i)] \forall i \in O_{t_{k+1}}\} \\
&\subset \{r \in \mathcal{R} : \mathcal{D}[r(x_i)] = y_i \forall i \in O_k\} \cap \{\mathcal{D}[r(x_l)] + \epsilon < \mathcal{D}[r(x_i)] \forall i \in O_{t_k}\} \\
&\subset C_l(O_{t_k})
\end{aligned}
$$

and

$$
C_l(O_{t_{k+1}}) \subset \cup_{j \in [n] \setminus O_{t_{k+1}}} C_j(O_{t_{k+1}}) = \mathcal{R}_{t_{k+1}}.
$$

Thus, we have that

$$
\frac{\mathbb{P}(C_{l_0}(O_{t_k}))}{2} \geq \mathbb{P}(C_{l_0}(O_{t_{k+1}})). \tag{6}
$$

**Step 3: Eliminating functions.** Let $f^* \in \mathcal{F}_{\mathcal{R}}$ such that there exists $r^* \in \mathcal{R}$ such that $f^*(x_i) = \mathcal{D}[r^*(x_i)] = y_i$ for all $i \in [n]$. Thus, using (3) and (6) in Steps 1 and 2 respectively, we see that after $\bar{t} := c v_{best}^* \log(\frac{1}{\mathbb{P}(S_{f^*})}) \log(n)$ queries we have that for all $l \in [n]$,

$$
\mathbb{P}(C_l(O_{\bar{t}}))) < \mathbb{P}(S_{f^*}).
$$

This implies that $S_{f^*} \not\subset C_l(O_{t_{\log(\frac{1}{\mathbb{P}(S_{f^*})})}}))$ for all $l$. Therefore,

$$
S_{f^*} \not\subset \cup_{l=1}^n C_l(O_{\bar{t}})) = \mathcal{R}_{\bar{t}}.
$$

Therefore, $f^*$ was kicked out some round prior to $\bar{t}$. Since $\mathcal{D}[r^*(x_i)] = y_i$ for all $i \in [n]$, inspection of Algorithm 1 reveals that the only way that $f^*$ could be removed from the version space is there exists some $s \leq \bar{t} \, I_s$ is queried where $f^*(x_{I_s}) \leq \min_{j \in [n]} f^*(x_j) + \epsilon$. This completes the proof.

$\square$

## D  Cumulative Loss Minimization Proofs

*Proof of Theorem 3.* Consider an algorithm $\mathcal{A}$ that after incurring cost at most $C(\mathcal{F})$ declares that it has identified the minimizer. There exists some $\bar{g} : \mathcal{X} \mapsto \{w_1, \ldots, w_L\}$ achieving the maximum in $v_{loss}^*$. Suppose the algorithm queries $i_1, \ldots, i_{t_0}$ and receives feedback $g(x_{i_j})$ and time $t_0$ is the earliest point at which the algorithm can declare that it can identify $j_0 \in [n]$ such that

$$
\{f \in \mathcal{F} \text{ s.t. } f(x_i) = \bar{g}(x_i) \forall i \in \{i_1, \ldots, i_{t_0}\}\} \subset \{f \in \mathcal{F} : j_0 \in \text{argmin}_{i \in [n]} f(x_i)\}.
$$

$$\mathcal{F}_1 \longleftarrow \mathcal{F}\,;$$
**for** $t = 1, 2, \ldots, T$ **do**

    Let $I_t \in \mathrm{argmin}_{i \in [n] \setminus \{I_1, \ldots, I_{t-1}\}} \max_{\bar{f} \in \mathcal{F}_t} \frac{\bar{f}(x_i)}{|\mathcal{F}_t(x_i) \cup \mathcal{F}((x_i, \bar{f}(x_i))|}$ ;

    Query $x_{I_t}$ and observe $f^*(x_{I_t})$;

    Let $\mathcal{F}_{t+1} \longleftarrow \mathcal{F}_t \setminus \mathcal{F}_t(x_{I_t}) \cup \mathcal{F}((x_{I_t}, f^*(x_{I_t}))$

**Algorithm 3:** `GRAILS` for Loss Minimization via Enumeration

We claim that there exists $\bar{f}$ such that

$$\{f \in \mathcal{F} \text{ s.t. } f(x_i) = \bar{f}(x_i) \forall i \in \{i_1, \ldots, i_{t_0-1}\}\} \neq \emptyset.$$

If not, then the algorithm could terminate at time $t_0 - 1$, contradicting the definition of $t_0$. Thus, on the instance given by $\bar{f}$, the algorithm suffers

$$\sum_{s=1}^{t_0-1} \bar{f}(x_{i_s})$$

$$= \sum_{s=1}^{t_0-1} \bar{g}(x_{i_s})$$

$$\geq \min_{I \subset [n]} \min_{\tilde{I} \subset I : |\tilde{I}| = |I|-1} \sum_{i \in \tilde{I}} \bar{g}(x_i)$$

$$\text{s.t. } \exists j \in [n] \text{ s.t. } \{f \in \mathcal{F} \text{ s.t. } f(x_i) = \bar{g}(x_i) \forall i \in [I]\} \subset \{f \in \mathcal{F} : j \in \mathrm{argmin}_{i \in [n]} f(x_i)\}$$

$$= \max_{g : \mathcal{X} \mapsto \mathcal{Y}} \min_{I \subset [n]} \min_{\tilde{I} \subset I : |\tilde{I}| = |I|-1} \sum_{i \in \tilde{I}} g(x_i)$$

$$\text{s.t. } \exists j \in [n] \text{ s.t. } \{f \in \mathcal{F} \text{ s.t. } f(x_i) = g(x_i) \forall i \in [I]\} \subset \{f \in \mathcal{F} : j \in \mathrm{argmin}_{i \in [n]} f(x_i)\}$$

$$= v_{loss}^*$$

where we used the definition of $\bar{g}$ as attaining the maximum in the definition of $v_{loss}^*$. This completes the proof.

$\square$

Here, we give an inefficient algorithm for cumulative loss minimization, Algorithm 3.

*Proof of Theorem 5.* Consider round $t$. Define the function $g : \{x_1, \ldots, x_n\} \mapsto \mathcal{Y}$ in the following way:

$$g(x_i) = \begin{cases} \tilde{f}(x_i) \text{ s.t. } \tilde{f} = \arg\max_{\bar{f} \in \mathcal{F}_t} \frac{\bar{f}(x_i)}{|\mathcal{F}_t(x_i) \cup \mathcal{F}((x_i, \bar{f}(x_i))|} & i \notin \{I_1, \ldots, I_{t-1}\} \\ f^*(x_i) & \text{o/w} \end{cases}$$

Let $\tilde{I} = \{i_1, \ldots, i_l\} \subset [n]$ attain the optimum in the following optimization problem:

$$\min_{l \in [n], i_1, \ldots, i_l \in [n]} \sum_{j=1}^{l-1} g(x_{i_j}) \tag{7}$$

$$\text{s.t. } \exists j \in [n] \text{ s.t. } \{f \in \mathcal{F}_t : f(x_i) = g(x_i) \forall i \in \{i_1, \ldots, i_l\}\} \tag{8}$$

$$\subset \{f \in \mathcal{F}_t : j \in \mathrm{argmin}_{i \in [n]} f(x_i)\} \tag{9}$$

Define $\tilde{\mathcal{F}} = \{f \in \mathcal{F}_t : f(x_i) = g(x_i) \forall i \in \tilde{I}\}$. If $\tilde{F}$ is empty, set $I = \tilde{I}$. If $\tilde{F}$ is nonempty, then by definition of the above optimization problem, there exists $j_0 \in [n]$ such that for all $f \in \tilde{F}$, $j_0 \in \mathrm{argmin}_{i \in [n]} f(x_i)$. Define $I = \tilde{I} \cup \{j_0\}$. **Step 1: Case 1.** Suppose that $|\mathcal{F}_t(x_{i_l}) \cup \mathcal{F}((x_i, g(x_{i_l}))| \leq \frac{|\mathcal{F}_t|}{2}$. Note that

$$\mathcal{F}_t \subset \{f \in \mathcal{F}_t : \exists i \in I \text{ s.t. } f(x_i) \neq g(x_i) \text{ or } i \in \mathrm{argmin}_{j \in [n]} f(x_j)\}.$$

Then,

$$|\mathcal{F}_t| \leq |\{f \in \mathcal{F}_t : \exists i \in I \text{ s.t. } f(x_i) \neq g(x_i) \text{ or } i \in \operatorname{argmin}_{j \in [n]} f(x_j)\}|$$

$$\leq \sum_{i \in I} |\{f \in \mathcal{F}_t : f(x_i) \neq g(x_i) \text{ or } i \in \operatorname{argmin}_{j \in [n]} f(x_j)\}|$$

$$\leq \sum_{i \in I \setminus \{i_l\}} |\{f \in \mathcal{F}_t : f(x_i) \neq g(x_i) \text{ or } i \in \operatorname{argmin}_{j \in [n]} f(x_j)\}| + \frac{|\mathcal{F}_t|}{2}$$

which implies that

$$\frac{|\mathcal{F}_t|}{2} \leq \sum_{i \in I \setminus \{i_l\}} |\{f \in \mathcal{F}_t : f(x_i) \neq g(x_i) \text{ or } i \in \operatorname{argmin}_{j \in [n]} f(x_j)\}|. \tag{10}$$

Then, we have that

$$\frac{f^*(x_{I_t})}{|\mathcal{F}_t(x_{I_t}) \cup \mathcal{F}((x_i, f^*(x_{I_t})))|}$$

$$\leq \frac{g(x_{I_t})}{|\mathcal{F}_t(x_{I_t}) \cup \mathcal{F}((x_i, g(x_{I_t})))|} \tag{11}$$

$$\leq \frac{\sum_{i \in I \setminus \{i_l\}} g(x_i)}{\sum_{i \in I \setminus \{i_l\}} |\{f \in \mathcal{F}_t : f(x_i) \neq g(x_i) \text{ or } i \in \operatorname{argmin}_{j \in [n]} f(x_j)\}|} \tag{12}$$

$$\leq 2 \frac{\sum_{i \in I \setminus \{i_l\}} g(x_i)}{|\mathcal{F}_t|} \tag{13}$$

$$\leq \frac{2(v_{loss}^* + \max_{i \in [n], f \in \mathcal{F}_t} f(x))}{|\mathcal{F}_t|} \tag{14}$$

where (11) follows by the definition of $g$, (12) follows by definition of $I_t$ and the mathematical fact that given positive numbers $a_1, \ldots, a_k$ and $b_1, \ldots, b_k$

$$\min_{i=1,\ldots,k} \frac{a_i}{b_i} \leq \frac{\sum_{i=1}^k a_i}{\sum_{i=1}^k b_i},$$

(13) follows by (10), and (14) follows by

$$\sum_{i \in \bar{I} \setminus \{l_l\}} g(x_i) \leq v_{loss}^*$$

by the definition of the optimization problem in (7) and we used $g(x_{j_0}) \leq \max_{i \in [n], f \in \mathcal{F}_t} f(x_i))$ by definition of $g$. Rearranging, we have that

$$f^*(x_{I_t}) \leq \frac{2|\mathcal{F}_t(x_{I_t}) \cup \mathcal{F}((x_i, f^*(x_{I_t}))(v_{loss}^* + \max_{i \in [n], f \in \mathcal{F}_t} f(x_i))}{|\mathcal{F}_t|}$$

$$= 2 \frac{(|\mathcal{F}_t| - |\mathcal{F}_{t+1}|)(v_{loss}^* + \max_{i \in [n], f \in \mathcal{F}_t} f(x_i))}{|\mathcal{F}_t|}.$$

**Step 2: Case 2.** On the other hand, suppose that $|\mathcal{F}_t(x_{i_l}) \cup \mathcal{F}((x_i, g(x_{i_l})))| \geq \frac{|\mathcal{F}_t|}{2}$. Then,

$$\frac{f^*(x_{I_t})}{|\mathcal{F}_t(x_{I_t}) \cup \mathcal{F}((x_i, f^*(x_{I_t})))|} \leq \frac{g(x_{I_t})}{|\mathcal{F}_t(x_{I_t}) \cup \mathcal{F}((x_i, g(x_{I_t})))|}$$

$$\leq \frac{g(x_{i_l})}{|\mathcal{F}_t(x_{i_l}) \cup \mathcal{F}((x_i, g(x_{i_l})))|} \tag{15}$$

$$\leq \frac{2 \max_{i \in [n], f \in \mathcal{F}_t} f(x_i)}{|\mathcal{F}_t|} \tag{16}$$

where (15) follows by the definition of $I_t$ and (16) follows by the definition of $g$. Then, we have rearranging once again

$$f^*(x_{I_t}) \leq 2 \frac{2|\mathcal{F}_t(x_{I_t}) \cup \mathcal{F}((x_i, f^*(x_{I_t}))|\max_{i \in [n], f \in \mathcal{F}_t} f(x)}{|\mathcal{F}_t|}$$

**Step 3: Putting it together.** Thus, we have that

$$\sum_{t \geq 1} f^*(x_{I_t}) \leq 2[v_{loss}^* + \max_{i \in [n], f \in \mathcal{F}_t} f(x)] \sum_{t \geq 1} \frac{(|\mathcal{F}_t| - |\mathcal{F}_{t+1}|)}{|\mathcal{F}_t|}$$

$$\leq 2[v_{loss}^* + \max_{i \in [n], f \in \mathcal{F}_t} f(x)] \sum_{t \geq 1} \frac{1}{|\mathcal{F}_t|} + \frac{1}{|\mathcal{F}_t| - 1} + \ldots + + \frac{1}{|\mathcal{F}_{t+1}| + 1}$$

$$\leq c2[v_{loss}^* + \max_{i \in [n], f \in \mathcal{F}_t} f(x)] \ln(|\mathcal{F}|)$$

where we used that $\frac{1}{|\mathcal{F}_t|} \leq \frac{1}{i}$ for $i \leq |\mathcal{F}_t|$ and that $\sum_{l=1}^{k} \frac{1}{l} = \Theta(\ln(k))$.

$\square$

*Proof of Theorem 4.* For the sake of abbreviation, let $\mathbb{P}(\cdot) := \mathbb{P}_{r \sim \pi}(\cdot)$ and $\mathbb{P}_k(\cdot) := \mathbb{P}_{r \sim P_k}(\cdot)$.

**Step 1: Bounding the loss in each phase**

Consider round $t$ in phase $k$. Define the function $g : \{x_1, \ldots, x_n\} \mapsto \mathcal{Y}$ in the following way:

$$g(x_i) = \begin{cases} \mathcal{D}[\tilde{r}(x_i)] \text{ s.t. } \tilde{r} = \arg\max_{\bar{r} \in \mathcal{R}_t} \frac{\bar{r}(x_i)}{\mathbb{P}_k(\mathcal{R}_t(x_i) \cup \mathcal{R}((x_i, \bar{f}(x_i))))} & i \notin \{I_1, \ldots, I_{t-1}\} \\ f^*(x_i) & \text{o/w} \end{cases}$$

Let $\tilde{I} = \{i_1, \ldots, i_l\} \subset [n]$ attain the optimum in the following optimization problem:

$$\min_{l \in [n], i_1, \ldots, i_l \in [n]} \sum_{j=1}^{l-1} g(x_{i_j})$$

$$\text{s.t. } \exists j \in [n] : \{r \in \mathcal{R}_t : \mathcal{D}[r](x_i) = g(x_i) \, \forall i \in \{i_1, \ldots, i_l\}\}$$
$$\subset \{r \in \mathbb{R}_t : j \in \arg\min_{i \in [n]} \mathcal{D}[r(x_i)]\}$$

Define $\tilde{\mathcal{R}} = \{r \in \mathcal{R}_t : \mathcal{D}[r(x_i)] = g(x_i) \forall i \in \tilde{I}\}$. If $\tilde{\mathcal{R}}$ is empty, set $I = \tilde{I}$. If $\tilde{\mathcal{R}}$ is nonempty, then by definition of the above optimization problem, there exists $j_0 \in [n]$ such that for all $r \in \tilde{\mathcal{R}}$, $j_0 \in \arg\min_{i \in [n]} \mathcal{D}[r(x_i)]$. Define $I = \tilde{I} \cup \{j_0\}$.

**Step 2.1: Case 1.** Suppose that $\mathbb{P}_k(\mathcal{R}_t(x_{i_l}) \cup \mathcal{R}((x_i, g(x_{i_l})))) \leq \frac{\mathbb{P}_k(\mathcal{R}_t)}{2}$. Note that

$$\mathcal{R}_t \subset \{r \in \mathcal{R}_t : \exists i \in I \text{ s.t. } \mathcal{D}[r(x_i)] \neq g(x_i) \text{ or } i \in \arg\min_{j \in [n]} \mathcal{D}[r(x_j)]\}.$$

Then,

$$\mathbb{P}_k(\mathcal{R}_t) \leq \mathbb{P}_k(\{r \in \mathcal{R}_t : \exists i \in I \text{ s.t. } \mathcal{D}[r(x_i)] \neq g(x_i) \text{ or } i \in \arg\min_{j \in [n]} \mathcal{D}[r(x_j)]\})$$

$$\leq \sum_{i \in I} \mathbb{P}_k(\{r \in \mathcal{R}_t : \exists i \in I \text{ s.t. } \mathcal{D}[r(x_i)] \neq g(x_i) \text{ or } i \in \arg\min_{j \in [n]} \mathcal{D}[r(x_j)]\})$$

$$\leq \sum_{i \in I \setminus \{i_l\}} \mathbb{P}_k(\{r \in \mathcal{R}_t : \exists i \in I \text{ s.t. } \mathcal{D}[r(x_i)] \neq g(x_i) \text{ or } i \in \arg\min_{j \in [n]} \mathcal{D}[r(x_j)]\})$$

$$+ \frac{\mathbb{P}_k(\mathcal{R}_t)}{2}$$

which implies that

$$\frac{\mathbb{P}_k(\mathcal{R}_t)}{2} \leq \sum_{i \in I \setminus \{i_l\}} \mathbb{P}_k(\{r \in \mathcal{R}_t : \exists i \in I \text{ s.t. } \mathcal{D}[r(x_i)] \neq g(x_i) \text{ or } i \in \arg\min_{j \in [n]} \mathcal{D}[r(x_j)]\}).$$

Then, we have that

$$\frac{f^*(x_{I_t})}{\mathbb{P}_k(\mathcal{R}_t(x_{I_t}) \cup \mathcal{R}((x_i, f^*(x_{I_t}))))} \tag{17}$$

$$\leq \frac{g(x_{I_t})}{\mathbb{P}_k(\mathcal{R}_t(x_{I_t}) \cup \mathcal{R}((x_i, g(x_{I_t}))))} \tag{18}$$

$$\leq \frac{\sum_{i \in I \setminus \{i_l\}} g(x_i)}{\sum_{i \in I \setminus \{i_l\}} \mathbb{P}_k(\{r \in \mathcal{R}_t : \exists i \in l \text{ s.t. } \mathcal{D}[r(x_i)] \neq g(x_i) \text{ or } i \in \operatorname{argmin}_{j \in [n]} \mathcal{D}[r(x_j)]\})}$$

$$\leq 2 \frac{\sum_{i \in I \setminus \{i_l\}} g(x_i)}{\mathbb{P}_k(\mathcal{R}_t)}$$

$$= 2 \frac{\sum_{i \in \tilde{I} \setminus \{i_l\} \cup \{j_0\}} g(x_i)}{\mathbb{P}_k(\mathcal{R}_t)}$$

$$\leq \frac{2(v^*_{loss} + \max_{i \in [n], r \in \mathcal{R}_t} \mathcal{D}[r(x_i)])}{\mathbb{P}_k(\mathcal{R}_t)}$$

where the above series of inequalities follows by the same arguments used to show the inequalities (11) to (14) Rearranging, we have that

$$f^*(x_{I_t}) \leq \frac{2\mathbb{P}_k(\mathcal{R}_t(x_{I_t}) \cup \mathcal{R}((x_i, f^*(x_{I_t})))[v^*_{loss} + \max_{i \in [n], r \in \mathcal{R}_t} \mathcal{D}[r(x_i)]]}{\mathbb{P}_k(\mathcal{R}_t)}$$

$$= 2 \frac{(\mathbb{P}_k(\mathcal{R}_t) - \mathbb{P}_k(\mathcal{R}_{t+1}))[v^*_{loss} + \max_{i \in [n], r \in \mathcal{R}_t} \mathcal{D}[r(x_i)]]}{\mathbb{P}_k(\mathcal{R}_t)}.$$

**Step 2.2: Case 2.** On the other hand, suppose that $\mathbb{P}_k(\mathcal{R}_t(x_{i_l}) \cup \mathcal{R}((x_i, g(x_{i_l})))) \geq \frac{\mathbb{P}_k(\mathcal{R}_t)}{2}$. Then,

$$\frac{f^*(x_{I_t})}{\mathbb{P}_k(\mathcal{R}_t(x_{I_t}) \cup \mathcal{R}((x_i, f^*(x_{I_t}))))} \leq \frac{g(x_{I_t})}{\mathbb{P}_k(\mathcal{R}_t(x_{I_t}) \cup \mathcal{R}((x_i, g(x_{I_t}))))} \tag{19}$$

$$\leq \frac{g(x_{i_l})}{\mathbb{P}_k(\mathcal{R}_t(x_{i_l}) \cup \mathcal{R}((x_i, g(x_{i_l}))))}$$

$$\leq \frac{2\max_{i \in [n], r \in \mathcal{R}_t} \mathcal{D}[r(x_i)]}{\mathbb{P}_k(\mathcal{R}_t)}$$

where the above series of inequalities follows by the same arguments used to show (15) to (11). These imply

$$f^*(x_{I_t}) \leq 2 \frac{\mathbb{P}_k(\mathcal{R}_t(x_{I_t}) \cup \mathcal{R}((x_i, f^*(x_{I_t})))\max_{i \in [n], r \in \mathcal{R}_t} \mathcal{D}[r(x_i)]}{\mathbb{P}_k(\mathcal{R}_t)}$$

Thus, we have that

$$\sum_{t \geq t_k : \mathbb{P}_k(\mathcal{R}_t) \geq 1/2n} f^*(x_{I_t})$$

$$\leq 2[v^*_{loss} + \max_{i \in [n], r \in \mathcal{R}_t} \mathcal{D}[r(x_i)]] \sum_{t \geq t_k : \mathbb{P}_k(\mathcal{R}_t) \geq 1/2n} \frac{\mathbb{P}_k(\mathcal{R}_t) - \mathbb{P}_k(\mathcal{R}_{t+1})}{\mathbb{P}_k(\mathcal{R}_t)}$$

$$= 2[v^*_{loss} + \max_{i \in [n], r \in \mathcal{R}_t} \mathcal{D}[r(x_i)]] \sum_{t \geq t_k : \mathbb{P}_k(\mathcal{R}_t) \geq 1/2n} \frac{2n(\mathbb{P}_k(\mathcal{R}_t) - \mathbb{P}_k(\mathcal{R}_{t+1}))}{2n\mathbb{P}_k(\mathcal{R}_t)}$$

$$\leq 2[v^*_{loss} + \max_{i \in [n], r \in \mathcal{R}_t} \mathcal{D}[r(x_i)]]$$

$$\cdot [1 + \sum_{t \geq t_k : \mathbb{P}_k(\mathcal{R}_{t+1}) \geq 1/2n} \frac{1}{2n\mathbb{P}_k(\mathcal{R}_t)} + \frac{1}{2n\mathbb{P}_k(\mathcal{R}_t) - 1} + \ldots + \frac{1}{2n\mathbb{P}_k(\mathcal{R}_{t+1}) + 1}]$$

$$\leq c \ln(n)[v^*_{loss} + \max_{i \in [n], r \in \mathcal{R}_t} \mathcal{D}[r(x_i)]]$$

where we used that $\frac{1}{2n\mathbb{P}_k(\mathcal{R}_t)} \leq \frac{1}{i}$ for $i \leq 2n\mathbb{P}_k(\mathcal{R}_t)$ and that $\sum_{l=1}^{k} \frac{1}{l} = \Theta(\ln(k))$. This bounds the cumulative loss incurred in round $k$.

**Step 2: Finish the proof.** We have that

$$\frac{\mathbb{P}(C_{l_0}(O_{t_k}))}{2} \geq \mathbb{P}(C_{l_0}(O_{t_{k+1}})) \tag{20}$$

by the same argument used in the proof of Theorem 2. Thus, the loss incurred by the algorithm is at most

$$\ln(n)\ln(\frac{1}{\mathbb{P}(S_{\min})})[v_{loss}^* + \max_{i \in [n], r \in \mathcal{R}_t} \mathcal{D}[r(x_i)]].$$

$\square$

## D.1 Other Cumulative Loss Minimization Results

*Proof of Proposition 7.* Fix $f^* \in \mathcal{F}$. Let $T_0 = n$. There exists a round $\bar{T} \leq T_0$ at which Algorithm 3 identifies $i_* \in \min_{j \in [n]} f^*(x_j)$, at which point it only plays $i_*$. Note that since the algorithm is deterministic and there is no noise $\bar{T}$ is deterministic. We have that

$$\sum_{t=1}^{T_0}(f^*(x_{I_t}) - \min_{j \in [n]} f^*(x_j)) = \sum_{t=1}^{\bar{T}} f^*(x_{I_t}) - \bar{T}\min_{j \in [n]} f^*(x_j)$$

since no regret is incurred for $t > \bar{T}$.

**Case 1:** Suppose that $\bar{T} \geq 4v_{best}^* \ln(|\mathcal{F}|)$. First, we note that by Theorem 5, we have that

$$\sum_{t=1}^{\bar{T}} f^*(x_{I_t}) \leq 2(v_{loss}^* + \max_{i \in [n], f \in \mathcal{F}} f^*(x_i))\ln(|\mathcal{F}|) \tag{21}$$

$$\leq 2(v_{best}^* \max_{y \in \mathcal{Y}} y + \max_{i \in [n], f \in \mathcal{F}} f^*(x_i))\ln(|\mathcal{F}|) \tag{22}$$

$$\leq 4v_{best}^* \ln(|\mathcal{F}|)\max_{y \in \mathcal{Y}} y \tag{23}$$

where (22) follows since $v_{loss}^* \leq v_{best}^* \max_{y \in \mathcal{Y}} y$ by Proposition 6, and (23) follows since $v_{best}^* \geq 1$ by assumption on $\mathcal{F}_\mathcal{R}$. Since $\bar{T} \geq 4v_{best}^* \ln(|\mathcal{F}_\mathcal{R}|)$ and $\min_{y \in \mathcal{Y}} y \geq 0$, we have that

$$\sum_{t=1}^{\bar{T}} f^*(x_{I_t}) - \bar{T}\min_{j \in [n]} f^*(x_j) \leq 4v_{best}^* \ln(|\mathcal{F}_\mathcal{R}|)\max_{y \in \mathcal{Y}} y - \bar{T}\min_{j \in [n]} f^*(x_j)$$

$$\leq 4v_{best}^* \ln(|\mathcal{F}|)\max_{y \in \mathcal{Y}} y - 4v_{best}^* \ln(|\mathcal{F}|)\min_{y \in \mathcal{Y}} y$$

$$= 4v_{best}^* \ln(|\mathcal{F}_\mathcal{R}|)\Delta_{\max}.$$

**Case 2:** Now, suppose that $\bar{T} \leq 4v_{best}^* \ln(|\mathcal{F}_\mathcal{R}|)$. Then,

$$\sum_{t=1}^{\bar{T}} f^*(x_{I_t}) - \bar{T}\min_{j \in [n]} f^*(x_j) \leq \bar{T}(\max_{y \in \mathcal{Y}} y - \min_{j \in [n]} f^*(x_j))$$

$$\leq \bar{T}(\max_{y \in \mathcal{Y}} y - \min_{y \in \mathcal{Y}} y)$$

$$\leq 4v_{best}^* \ln(|\mathcal{F}|)\Delta_{\max}$$

$\square$

*Proof of Proposition 8.* Let $\bar{y} > \tilde{y} > y_*$. We have $\Delta_{\min} = \tilde{y} - y_*$ and $\Delta_{\max} = \bar{y} - y_*$. Let $m$ be even and $n = m + m/2$. Let $\mathcal{F} = \{f_1, \ldots, f_m\}$ where

$$f_j(x_i) = \begin{cases} \bar{y} & i \in [m/2] \setminus \{2j-1, 2j\} \\ y_* & i \in \{2j-1, 2j\} \\ \tilde{y} & i \in ([m+m/2] \setminus [m/2]) \setminus \{m/2+j\} \\ y_* & i = m/2+j \end{cases}$$

The greedy algorithm in the regret minimization algorithm from [11] finds the best arm using a greedy best arm identification algorithm that greedily removes functions from the version space. Once if has found the best arm, it pulls the best arm for the remainder of the game. Each query in $[m/2]$ removes 2 functions while each query in $[m + m/2] \setminus [m/2]$ removes 1 function. Thus, the regret minimization algorithm from [11] would use the queries in $[m/2]$ to remove functions, incurring a regret of $\Delta_{\max}$ for each query. In the worst case, it would incur a regret of $\Omega(\Delta_{\max} m) = \Omega(\Delta_{\max} n)$. On the other hand, Algorithm 3 would query only in $[m + m/2] \setminus [m/2]$ if $\Delta_{\max} >> \Delta_{\min}$, incurring a regret of $O(m\Delta_{\min}) = O(n\Delta_{\min})$. $\square$

*Proof of Proposition 9.*

$$\max_{f \in \mathcal{F}} \text{loss}(\bar{\mathcal{A}}; f; T_0) = \max_{f \in \mathcal{F}} \sum_{t=1}^{T_0} f(x_{I_t})$$

$$= \max_{f \in \mathcal{F}} \sum_{t=1}^{T_0} (f(x_{I_t}) - \min_{j \in [n]} f(x_j)) + T_0 \min_{j \in [n]} f(x_j)$$

$$= \max_{f \in \mathcal{F}} \text{regret}(\mathcal{A}; f; T_0) + T_0 \min_{j \in [n]} f(x_j)$$

$$\leq \max_{f \in \mathcal{F}} \text{regret}(\mathcal{A}; f; T_0) + \max_{f \in \mathcal{F}} T_0 \min_{j \in [n]} f(x_j)$$

$$\leq c \min_{\mathcal{A}} \max_{f \in \mathcal{F}} \text{regret}(\mathcal{A}; f; T_0) + \max_{f \in \mathcal{F}} T_0 \min_{j \in [n]} f(x_j)$$

$$\leq (c + 1) \min_{\mathcal{A}} \max_{f \in \mathcal{F}} \text{loss}(\mathcal{A}; f; T_0)$$

where in the last line we used that $\text{regret}(\mathcal{A}; f; T_0) \leq \text{loss}(\mathcal{A}; f; T_0)$ since $\min_{y \in \mathcal{Y}} y \geq 0$ by assumption, which implies that $\min_{\mathcal{A}} \max_{f \in \mathcal{F}} \text{regret}(\mathcal{A}; f; T_0) \leq \min_{\mathcal{A}} \max_{f \in \mathcal{F}} \text{loss}(\mathcal{A}; f; T_0)$, and $\max_{f \in \mathcal{F}} T_0 \min_{j \in [n]} f(x_j) \leq \min_{\mathcal{A}} \max_{f \in \mathcal{F}} \text{loss}(\mathcal{A}; f; T_0)$. $\square$

*Proof of Proposition 6.* Fix $g : \mathcal{X} \mapsto \mathcal{Y}$. By definition of $v_{best}^*$, there exists $\bar{I} \subset [n]$ such that $|\bar{I}| \leq v_{best}^*$ and there exists $j_0 \in [n]$ such that

$$\{f \in \mathcal{F} : f(x_i) = g(x_i) \forall i \in \bar{I}\} \subset \{f \in \mathcal{F} : j_0 \in \text{argmin}_{k \in [n]} f(x_k)\}$$

This immediately implies that

$$\min_{l \in [n], i_1, \ldots, i_l} \sum_{j=1}^{l-1} g(x_{i_j})$$
$$\text{s.t. } \exists j \in [n] \text{ s.t. } \{f \in \mathcal{F} : f(x_i) = g(x_i) \forall i \in \{i_1, \ldots, i_l\}$$
$$\subset \{f \in \mathcal{F} : j \in \text{argmin}_{i \in [n]} f(x_i)\}$$
$$\leq \sum_{i \in \bar{I}} g(x_i)$$
$$\leq |\bar{I}| \max_{y \in \mathcal{Y}} y$$
$$\leq v_{best}^* \max_{y \in \mathcal{Y}} y$$

$\square$

*Proof of Proposition 10.* Suppose that $n = 2$ and $\mathcal{F} = \{f_1, f_2\}$ such that $f_1(x_1) = \Delta$, $f_1(x_2) = \Delta - 1$, $f_2(x_1) = 1$, and $f_2(x_2) = \Delta/2 + 1$. Suppose $f_1$ is true. If one queries $x_1$, then one obtains a regret of 1 and a loss of $\Delta$. If one queries $x_2$, one obtains a regret of 0 and a loss of $\Delta - 1$. Now, suppose $f_2$ is true. If one queries $x_1$, then one obtains a regret of 0 and a loss of 1. If one queries $x_2$, one obtains a regret of $\Delta/2$ and a loss of $\Delta/2 + 1$. Thus, to minimize worst-case regret, one must query $x_1$, obtaining a worst case regret of 1. To minimize worst-case loss, one must query $x_2$, obtaining a worst-case loss of $\Delta - 1$. But, querying $x_2$ has a regret of $\Delta/2$, completing the proof. $\square$

```
k ⟵ 1, t_1 = 1, O_1 ⟵ ∅;
for t = 1, 2, … do
    Let I_t ∈ arg max_{i∈[n]\O_t} min_{j∈[k]} ℙ_{f∼P_k}(f ∈ 𝓕_t(x_i) ∪ 𝓕_t((x_i, L_j)));
    Query x_{I_t} and observe y_{I_t} and set O_{t+1} ⟵ O_t ∪ {I_t} ;
    Let 𝓕_{t+1} ⟵ 𝓕_t \ (𝓕_t(x_{I_t}) ∪ 𝓕_t((x_{I_t}, L(y_{I_t}))));
    if ℙ_{f∼P_k}(f ∈ 𝓕_{t+1}) ≤ 1/(2n) then
        P_{k+1} ⟵ 1/(n−|O_{t_k}|) Σ_{l∈[n]\O_{t_k}} π_{C_l(O_{t_k})};
        k ⟵ k + 1;
        t_k ⟵ t + 1;
```

**Algorithm 4:** `GRAILS` for Continuous Output Space

# E   Extension to Continuous Output Space

Now, we turn to the continuous output setting where we assume that the scores are continuous, that is, $y_1, \ldots, y_n \in [-1, 1]$. Let $\mathcal{F} \subset [-1, 1]^{\mathcal{X}}$ and suppose that there exists $f^* \in \mathcal{F}$ such that $f^*(x_i) = y_i$ for all $i = 1, \ldots, n$. Let $\epsilon > 0$. The goal is to identify $j \in [n]$ such that $f^*(x_j) \leq \min_{i \in [n]} f^*(x_i) + \epsilon$.

Let $\delta > 0$ and $L_i = [-1 + (i-1)\delta, -1 + i\delta)$ for $i = 1, 2, \ldots, \frac{2}{\delta} =: p$. $L_1, \ldots, L_p$ is a grid on the output space $[-1, 1]$. For $\tilde{y} \in [-1, 1]$, let $L(\tilde{y})$ denote the interval $L_j$ such that $\tilde{y} \in L_j$. Furthermore, define $\bar{L}_j := \frac{1}{2}(-1 + (j-1)\delta + (-1 + j\delta))$, the midpoint of the $L_j$ interval. Define

$$\mathcal{F}((x_i, L_j)) = \{f \in \mathcal{F} : f(x_i) \notin L_j\}$$
$$\mathcal{F}(x_i) = \{f \in \mathcal{F} : f(x_i) \leq \min_{j \in [n]} f(x_j) + \epsilon\}.$$

We overload notation, defining

$$C_l(O_t) := \{f \in \mathcal{F} : f(x_i) \in L(y_i) \, \forall i \in O_t\} \cap \{f(x_i) \leq f(x_l) + \epsilon \, \forall i \in O_t\}.$$

Define the following extended teaching dimension notion for the continuous output setting:

$$\upsilon_{\epsilon,\delta} := \max_{g:\mathcal{X} \mapsto \mathbb{R}} \min_{I \subset [n]} |I|$$
$$\text{s.t. } \exists j \in [n] : \{f \in \mathcal{F} : |f(x_i) - g(x_i)| \leq \delta \forall i \in I\} \subset \{f(x_j) \leq \min_{l \in [n]} f(x_l) + \epsilon\}.$$

Define for every $f \in \mathcal{F}$ the set

$$S_f = \{f' \in \mathcal{F} : L(f'(x_i)) = L(f(x_i)) \forall i \in [n]\}.$$

The sets $S_f$ induce a partition of $\mathcal{F}$.

**Theorem 7.** *Let $f^* \in \mathcal{F}$ such that $f^*(x_i) = y_i$ for all $i \in [n]$. After $c \ln(\frac{1}{\mathbb{P}_\pi(S_{f^*})})\upsilon^*_{\epsilon,\delta}$ queries, Algorithm 4 has queried some $i \in [n]$ satisfying $f^*(x_i) \leq \min_{j \in [n]} f^*(x_j) + \epsilon$*

We note that our algorithm for loss minimization can be extended to the continuous output space with a similar guarantee.

*Proof of Theorem 7.* For the sake of abbreviation, let $\mathbb{P}(\cdot) := \mathbb{P}_{f \sim \pi}(\cdot)$ and $\mathbb{P}_k(\cdot) := \mathbb{P}_{f \sim P_k}(\cdot)$.

**Step 1:** The Algorithm 1 breaks time into phases $[t_1, t_2], [t_2, t_3], \cdots$. In each phase $k$, the algorithm samples functions from a distinct distribution $P_k$. Consider round $k$, let $t \geq t_k$. Define $\theta \in \{w_1, \ldots, w_L\}^n$ by

$$\theta_i = \begin{cases} y_i & i \in O_t \\ \text{argmin}_{\tilde{y} \in \{\bar{L}_1, \ldots, \bar{L}_p\}} \mathbb{P}_k(f \in \mathcal{F}_t(x_i) \cup \mathcal{F}_t((x_i, L(\tilde{y}))) & \text{o/w} \end{cases}.$$

By definition of $\upsilon^*_{\epsilon,\delta}$, there exists $\tilde{I} = \{i_1, \ldots, i_{\upsilon^*}\}$ such that there exists $j_0 \in [n]$ such that $\forall f \in \mathcal{F}_t$ that satisfy $|f(x_i) - \theta_i| \leq \delta$ for all $i \in \tilde{I}$, it holds that $f(x_{j_0}) - \epsilon \leq \min_{l \in [n]} f(x_l)$. Define

$I = \{i_1, \ldots, i_{v^*}, j_0\}$. Then, we have that

$$\mathcal{F}_t = \{f \in \mathcal{F}_t : \exists i \in I \text{ s.t. } f(x_i) \leq \min_{l \in [n]} f(x_l) + \epsilon \text{ or } |f(x_i) - \theta_i| > \delta\}$$

$$= \{f \in \mathcal{F}_t : \exists i \in I \text{ s.t. } f(x_i) \leq \min_{l \in [n]} f(x_l) + \epsilon \text{ or } f(x_i) \notin L(\theta_i)\}$$

where the last equality follows since by definition of $L(\cdot)$, $|f(x_i) - \theta_i| > \delta$ implies that $f(x_i) \notin L(\theta_i)$. From this, it follows that

$$\mathbb{P}_k(\mathcal{F}_t) = \mathbb{P}_k(\{f \in \mathcal{F}_t : \exists i \in I \text{ s.t. } f(x_i) \leq \min_{l \in [n]} f(x_l) + \epsilon \text{ or } f(x_i) \notin L(\theta_i)\})$$

$$\leq \sum_{i \in I} \mathbb{P}_k(\{f \in \mathcal{F}_t : f(x_i) \leq \min_{l \in [n]} f(x_l) + \epsilon \text{ or } f(x_i) \notin L(\theta_i)\})$$

$$\leq (v^*_{\epsilon,\delta} + 1)\max_{i \in I} \mathbb{P}_k(\{f \in \mathcal{F}_t : f(x_i) \leq \min_{l \in [n]} f(x_l) + \epsilon \text{ or } f(x_i) \notin L(\theta_i)\})$$

$$= (v^*_{\epsilon,\delta} + 1)\mathbb{P}_k(\{f \in \mathcal{F}_t : f(x_{I_t}) \leq \min_{l \in [n]} f(x_l) + \epsilon \text{ or } f(x_{I_t}) \notin L(\theta_{I_t})\}) \qquad (24)$$

$$\leq (v^*_{\epsilon,\delta} + 1)\mathbb{P}_k(\{f \in \mathcal{F}_t : f(x_{I_t}) \leq \min_{l \in [n]} f(x_l) + \epsilon \text{ or } f(x_{I_t}) \notin L(y_{I_t})\}) \qquad (25)$$

$$= (v^*_{\epsilon,\delta} + 1)\mathbb{P}_k(\mathcal{F}_t(x_{I_t}) \cup \mathcal{F}_t((x_{I_t}, L(y_{I_t})))$$

where line (24) comes from the definition of $I_t$ and the line (25) comes from the definition of $\theta$. Noticing that

$$\mathcal{F}_t = (\mathcal{F}_t(x_{I_t}) \cup \mathcal{F}_t((x_{I_t}, L(y_{I_t}))) \cup \mathcal{F}_t \setminus (\mathcal{F}_t(x_{I_t}) \cup \mathcal{F}_t((x_{I_t}, L(y_{I_t})))$$

$$= (\mathcal{F}_t(x_{I_t}) \cup \mathcal{F}_t((x_{I_t}, L(y_{I_t}))) \cup \mathcal{F}_{t+1},$$

we have that

$$\mathbb{P}_k(\mathcal{F}_{t+1}) = \mathbb{P}_k(\mathcal{F}_t) - \mathbb{P}_k(\mathcal{F}_t(x_{I_t}) \cup \mathcal{F}_t((x_{I_t}, L(y_{I_t}))) \leq (1 - \frac{1}{v^*_{\epsilon,\delta} + 1})\mathbb{P}_k(\mathcal{F}_t).$$

Unraveling this recursive statement, we have that

$$\mathbb{P}_k(\mathcal{F}_{t+1}) \leq (1 - \frac{1}{v^* + 1})^{t - t_k}.$$

Thus, we see that in phase $k$ after $s_k := O(\ln(n)v^*_{\epsilon,\delta})$ rounds , we have that

$$\mathbb{P}_k(\mathcal{F}_{t_k + s_k}) \leq \frac{1}{2n}. \qquad (26)$$

The rest of the proof is similar to the proof of Theorem 2.

$\square$

# F  Active Classification and Extensions to other Objectives

In this Section, we apply our framework to other problems such as active classification and a problem for identifying a sufficiently good arm, defined shortly. We give a result for active classification and give brief sketches for the other objectives since their treatment is similar to the problems already studied in this paper.

**Active Classification.** We recall the active classification setting. Let $\mathcal{X}$ denote the input space and $\mathcal{Y} = \{-1, 1\}$ the output space. Let $x_1, \ldots, x_n \in \mathcal{X}$ be a fixed pool with associated scores $y_1, \ldots, y_n \in \mathcal{Y}$. At each round $t$, the learner selects (or queries) $I_t \in [n]$ and observes $y_{I_t}$. The goal is to identify $\text{argmin}_{f \in \mathcal{R}_\mathcal{R}} \sum_{i=1}^n \mathbb{1}\{y_i \neq f(x_i)\}$ using as few queries as possible. Recall that under realizability there exists $f^* \in \mathcal{F}_\mathcal{R}$ such that $f^*(x_i) = y_i$ for all $i \in [n]$.

We note that for active classification, the version space is convex, and so we may use a simpler version space. We also note that STOP($\mathcal{F}_\mathcal{R}, O_t$) can check whether there are $f, f' \in \mathcal{F}_t$ such that $f \neq f'$ by solving the following optimization problem for $j \in [n] \setminus O_t$ and $y \in \mathcal{Y}$

$$\exists f \text{ s.t. } f(x_i) = y_i \ \forall i \in O_t \wedge f(x_j) = y. \qquad (27)$$

```
k ⟵ 1, t₁ = 1, O₁ ⟵ ∅;
for t = 1, 2, . . . do
    Let Iₜ ∈ arg max_{i∈[n]\Oₜ} min_{y∈𝒴} ℙ_{r∼π_{ℛₜ}}(ℛₜ((xᵢ, y)));
    Query x_{Iₜ} and observe y_{Iₜ} and set O_{t+1} ⟵ Oₜ ∪ {Iₜ} ;
    Let ℛ_{t+1} ⟵ ℛₜ \ ℛₜ((x_{Iₜ}, y_{Iₜ}));
    if STOP(ℱ_ℛ, Oₜ) then
        return argmin_{i∈Oₜ} yᵢ
```

**Algorithm 5:** GRAILS for Active Classification

If there exists $j$ such that there exists $f, f' \in \mathcal{F}_t$ and $f(x_j) \neq f(x_j)$, then the Algorithm does not terminate. If not, then the algorithm terminates. We note that (27) is a convex optimization problem for function classes like linear functions, kernel methods, and convex functions.

We introduce the extended teaching dimension notion [21], which has previously been shown to be a lower bound for active classification:

$$\upsilon^*_{class}(\mathcal{F}) = \max_{g:\mathcal{X}\mapsto\mathcal{Y}} \min_{I\subset[n]} |I|$$

$$\text{s.t. } |\{f \in \mathcal{R} : \mathcal{D}[f(x_i)] = g(x_i) \forall i \in I\}| \leq 1.$$

Recall the definition $S_{\min} = \text{argmin}_{S_f:f\in\mathcal{F}_\mathcal{R},\mathbb{P}_\pi(S_f)>0}\mathbb{P}_\pi(S_f)$.

**Theorem 8.** *After* $c \log(\frac{1}{\mathbb{P}(S_{\min})})\upsilon^*_{class}(\mathcal{F}_\mathcal{R})$ *queries, the version space only contains* $S_{f^*}$.

*Proof of Theorem 8.* For the sake of abbreviation, let $\mathbb{P}(\cdot) := \mathbb{P}_{f\sim\pi}(\cdot)$ and $\mathbb{P}_t(\cdot) := \mathbb{P}_{f\sim\pi_t}(\cdot)$ where $\pi_t := \pi_{\mathcal{R}_t}$.

**Step 1:** Consider round $t$. Define

$$\theta_i = \begin{cases} y_i & i \in O_t \\ \text{argmin}_{y\in\mathcal{Y}}\mathbb{P}(r \in \mathcal{R}_t((x_i, y))) & \text{o/w} \end{cases}.$$

By definition of the extended teaching dimension $\upsilon^*_{class}(\mathcal{F}_\mathcal{R})$, there exists $I = \{i_1, \ldots, i_{\upsilon^*_{class}}\}$ such that

$$|\mathcal{G}_t| \leq 1$$

where

$$\mathcal{G}_t := \{\mathcal{D}[r] \in \mathcal{R}_t : \mathcal{D}[r(x_i)] = \theta_i \ \forall i \in I\}. \tag{28}$$

From this, it follows that

$$\mathbb{P}(\mathcal{R}_t \setminus \mathcal{G}_t) = \mathbb{P}(\{r \in \mathcal{R}_t : \exists i \in I \text{ s.t. } \mathcal{D}[r(x_i)] \neq \theta_i\})$$

$$\leq \sum_{i\in I} \mathbb{P}(\{r \in \mathcal{R}_t : \mathcal{D}[r(x_i)] \neq \theta_i\})$$

$$\leq \upsilon^*_{class}\max_{i\in I}\mathbb{P}(\{r \in \mathcal{R}_t : \mathcal{D}[r(x_i)] \neq \theta_i\})$$

$$= \upsilon^*_{class}\mathbb{P}(\{r \in \mathcal{R}_t : I_t \in \text{argmin}_{l\in[n]}\mathcal{D}[r(x_l)] \text{ or } \mathcal{D}[r(x_i)] \neq \theta_{I_t}\}) \tag{29}$$

$$\leq \upsilon^*_{class}\mathbb{P}(\{r \in \mathcal{R}_t : \mathcal{D}[r(x_i)] \neq y_{I_t}\}) \tag{30}$$

$$= \upsilon^*_{class}\mathbb{P}(\mathcal{R}_t((x_{I_t}, y_{I_t})))$$

where line (29) comes from the definition of $I_t$, the definition of $\theta$, and the fact that

$$\text{argmin}_i\mathbb{P}(r \in \mathcal{R}_t((x_i, \theta_i))) = \text{argmin}_i\frac{\mathbb{P}(r \in \mathcal{R}_t((x_i, \theta_i)))}{\mathbb{P}(\mathcal{R}_t)} = \text{argmin}_i\mathbb{P}_t(r \in \mathcal{R}_t((x_i, \theta_i)))$$

and the line (30) comes from the definition of $\theta$. Noticing that

$$\mathcal{R}_t = \mathcal{R}_t((x_{I_t}, y_{I_t}) \cup \mathcal{R}_t \setminus \mathcal{R}_t((x_{I_t}, y_{I_t}))$$

$$= \mathcal{R}_t((x_{I_t}, y_{I_t}) \cup \mathcal{R}_{t+1},$$

we have that

$$\mathbb{P}(\mathcal{R}_{t+1}) = \mathbb{P}(\mathcal{R}_t) - \mathbb{P}(\mathcal{R}_t((x_{I_t}, y_{I_t}))) \leq \mathbb{P}(\mathcal{R}_t) - \frac{1}{v^*_{class}} \mathbb{P}(\mathcal{R}_t \setminus \mathcal{G}_t). \tag{31}$$

We have that

$$\sum_{t=1}^{\infty} \mathbb{1}\{\mathcal{R}_t \neq S_{f^*}\} = \sum_{t=1}^{\infty} \mathbb{1}\{\mathcal{R}_t \neq S_{f^*} \wedge \mathbb{P}(\mathcal{G}_t) \leq \frac{1}{2}\mathbb{P}(\mathcal{R}_t)\}$$

$$+ \mathbb{1}\{\mathcal{R}_t \neq S_{f^*} \wedge \mathbb{P}(\mathcal{G}_t) > \frac{1}{2}\mathbb{P}(\mathcal{R}_t) \wedge \mathcal{G}_t \neq S_{f^*}\}$$

$$+ \mathbb{1}\{\mathcal{R}_t \neq S_{f^*} \wedge \mathbb{P}(\mathcal{G}_t) > \frac{1}{2}\mathbb{P}(\mathcal{R}_t) \wedge \mathcal{G}_t = S_{f^*}\}.$$

**Step 2: Bounding the first term.** We begin by bounding the first sum. Suppose $\mathbb{P}(\mathcal{G}_t) \leq \frac{1}{2}\mathbb{P}(\mathcal{R}_t)$. Then, (31) implies that we have that

$$\mathbb{P}(\mathcal{R}_{t+1}) \leq \mathbb{P}(\mathcal{R}_t) - \frac{1}{v^*_{class}} \mathbb{P}(\mathcal{R}_t \setminus \mathcal{G}_t) = (1 - \frac{1}{v^*_{class}})\mathbb{P}(\mathcal{R}_t) + \frac{1}{v^*_{class}}\mathbb{P}(\mathcal{G}_t) \leq (1 - \frac{1}{2v^*_{class}})\mathbb{P}(\mathcal{R}_t).$$

Thus, if for some $t' \geq t$ $\mathbb{1}\{\mathcal{R}_t \neq S_{f^*} \wedge \mathbb{P}(\mathcal{G}_t) \leq \frac{1}{2}\mathbb{P}(\mathcal{R}_t)\}$ occurs $k$ times in the rounds between $t$ and $t'$, we have that

$$\mathbb{P}(\mathcal{R}_{t'}) \leq (1 - \frac{1}{2v^*_{class}})^k \mathbb{P}(\mathcal{R}_t).$$

This implies that if this event occurs $c \log(\frac{1}{\mathbb{P}(S_{f^*})})v^*_{class}$ times, we have that $\mathcal{R}_{t'} = S_{f^*}$. This completes bounding the second term.

**Step 3: Bounding the second term.** Fix round $t$ and suppose that event $\mathbb{1}\{\mathcal{R}_t \neq S_{f^*} \wedge \mathbb{P}(\mathcal{G}_t) > \frac{1}{2}\mathbb{P}(\mathcal{R}_t)\}$ occurs. Since $\mathbb{P}(\mathcal{G}_t) > \frac{1}{2}\mathbb{P}(\mathcal{R}_t)$, there exists $\bar{f} \in \mathcal{F}_{\mathcal{R}}$ such that $\bar{f} \neq f^*$, $\mathcal{G}_t = S_{\bar{f}}$, and $\bar{r} \in \mathcal{R}$ such that $\mathcal{D}[\bar{r}] = \bar{f}$. Let $t' \geq t$ be the round at which $\bar{f}$ is kicked out or is the last remaining function. We claim that for all $s \in [t, t']$, $\mathcal{G}_s = S_{\bar{f}}$. Fix $i \in [n]$ and note that

$$\mathbb{P}_{r \sim \pi_s}(\mathcal{D}[r(x_i)] = \mathcal{D}[\bar{r}(x_i)]) \geq \mathbb{P}_{r \sim \pi_s}(\mathcal{D}[r] = \mathcal{D}[\bar{r}]) \tag{32}$$

$$= \frac{\mathbb{P}(\mathcal{D}[r] = \mathcal{D}[\bar{r}])}{\mathbb{P}(\mathcal{R}_s)} \tag{33}$$

$$> \frac{1}{2} \frac{\mathbb{P}(\mathcal{R}_t)}{\mathbb{P}(\mathcal{R}_s)} \tag{34}$$

$$\geq \frac{1}{2} \tag{35}$$

where (32) follows since if $\mathcal{D}[r] = \mathcal{D}[\bar{r}]$, then $\mathcal{D}[r(x_i)] = \mathcal{D}[\bar{r}(x_i)]$, (33) follows by the definition of conditional probability, (34) follows since $\mathbb{P}(S_{\bar{f}}) = \mathbb{P}(\mathcal{G}_t) > \frac{1}{2}\mathbb{P}(\mathcal{R}_t)$, and (35) follows since $\mathcal{R}_t \supset \mathcal{R}_s$ because $s \geq t$ and the version spaces are monotonically decreasing. Thus, inspection of the definition of $\mathcal{G}_s$ from (28) and the definition of $\theta$ as

$$\theta_i = \operatorname{argmin}_{y \in \mathcal{Y}} \mathbb{P}(r \in \mathcal{R}_s : \mathcal{D}[r(x_i)] \neq y) = \arg\max_{y \in \mathcal{Y}} \mathbb{P}(r \in \mathcal{R}_s : \mathcal{D}[r(x_i)] = y)$$

with $|\mathcal{Y}| = 2$ shows the claim that for all $s \in [t, t']$, $\mathcal{G}_s = S_{\bar{f}}$.

Now, consider round $t + 1$. Then, by (31) and $\mathcal{G}_{t+1} = S_{\bar{f}}$, we have that

$$\mathbb{P}_t(\mathcal{R}_{t+1} \setminus S_{\bar{f}}) + \Pr(S_{\bar{f}}) \leq (1 - \frac{1}{v^*_{class}})\mathbb{P}_t(\mathcal{R}_t \setminus S_{\bar{f}}) + \Pr(S_{\bar{f}}),$$

which implies that

$$\mathbb{P}(\mathcal{R}_{t+1} \setminus S_{\bar{f}}) \leq (1 - \frac{1}{v^*_{class}})\mathbb{P}(\mathcal{R}_t \setminus S_{\bar{f}}).$$

Unraveling this recurrence up to $t'$, we have that

$$\mathbb{P}_t(\mathcal{R}_{t'}) = \mathbb{P}_t(\mathcal{R}_{t'} \setminus S_{\bar{f}}) \le (1 - \frac{1}{v^*_{class}})^{t'-t}\mathbb{P}_t(\mathcal{R}_t \setminus S_{\bar{f}}) \le (1 - \frac{1}{v^*_{class}})^{t'-t}\mathbb{P}(\mathcal{R}_t).$$

where in the first equality we used the fact that by definition of $t'$, $\mathcal{R}_{t'} \cap S_{\bar{f}} = \emptyset$. Thus, we see that the second term is bounded above by at most $c\log(\frac{1}{\mathbb{P}(S_{f^*})})v^*_{class}$.

**Step 4. Bounding the third term.** By the argument in the previous step, we have that $\mathcal{G}_t = S_{f^*}$ for the rest of the game. Thus, we have that from (31)

$$\mathbb{P}(\mathcal{R}_{t+1}) \le \mathbb{P}(\mathcal{R}_t) - \frac{1}{v^*_{class}}\mathbb{P}(\mathcal{R}_t \setminus S_{f^*})$$

$$= (1 - \frac{1}{v^*_{class}})\mathbb{P}(\mathcal{R}_t) + \frac{1}{v^*_{class}}\mathbb{P}(S_{f^*}).$$

Thus, unravelling the recurrence, after $k$ rounds, we have that

$$\mathbb{P}(\mathcal{R}_{t+k}) \le (1 - \frac{1}{v^*_{class}})^k\mathbb{P}(\mathcal{R}_t) + \frac{1}{v^*_{class}}\sum_{l=1}^{k-1}(1 - \frac{1}{v^*_{class}})^l\mathbb{P}(S_{f^*})$$

$$\le (1 - \frac{1}{v^*_{class}})^k\mathbb{P}(\mathcal{R}_t) + \mathbb{P}(S_{f^*})$$

where the last inequality followed by the geometric series. This implies that if this event occurs $c\log(\frac{1}{\mathbb{P}(S_{\min})})v^*_{class}$ times, we have that $\mathcal{R}_{t'} = S_{f^*}$. This completes bounding the third term.

$\square$

**Corollary 2.** *If the most probable classifier is output at each round, then after $c\log(\frac{1}{\mathbb{P}(S_{f^*})})v^*_{class}(\mathcal{F}_{\mathcal{R}})$ queries, only $f^*$ will be output for the rest of the game.*

*Proof.* The only step that is slightly different is step 4. Here, it can be seen that after $c\log(\frac{1}{\mathbb{P}(S_{f^*})})v^*_{class}(\mathcal{F}_{\mathcal{R}})$ queries, $f^*$ is the most probable classifier and it will remain the most probable classifier for the remainder of the game. $\square$

Next, we consider some other settings briefly.

**Threshold/Satisficing Bandits.** Here we are given a threshold $\gamma \in \mathbb{R}$, and the goal is to identify $i \in [n]$ such that $y_i \le \gamma$ (this problem was introduced and studied in the standard multi-armed bandit setting in [34]). Then, the extended-teaching dimension notion is

$$v^*_\gamma = \max_{g:\mathcal{X} \mapsto \mathcal{Y}} \min_{I \subset [n]} |I|$$

$$\text{s.t. } \exists j \in [n] : \{f \in \mathcal{F} : \mathcal{D}[f(x_i)] = g(x_i)\forall i \in I\} \subset \{f \in \mathcal{F} : \mathcal{D}[f(x_j)] \le \gamma\}.$$

The version space is given by

$$\mathcal{F}_t = \{f \in \mathcal{F} : \mathcal{D}[f(x_i)] = y_i\forall i \in O_t\}.$$

The algorithm would query by the following rule

$$I_t \in \arg\max_{i \in [n]\setminus O_t} \min_{y \in \mathcal{Y}} \mathbb{P}_{f \sim P_k}(f \in \mathcal{F}_{t,\gamma}(x_i) \cup \mathcal{F}_t((x_i, y)))$$

where $\mathcal{F}_{t,\gamma}(x_i) = \{f \in \mathcal{F}_t : \mathcal{D}[f(x_i)] \le \gamma\}$. We note that for this problem it is not necessary to sample from the mixture $P_k$ because the version space is convex.

We conjecture that there are extensions to settings where the feedback is multi-dimensional (e.g., [35]) or the goal is constrained best arm identification (e.g., [36]).

**Input:** distribution $P$ ; event $\Sigma$, accuracy $\epsilon$, failure probability $\delta$, ;
$N \longleftarrow 2\epsilon^{-2}\log(1/\delta)$;
Sample $z_1, \ldots, z_N \sim P$ ;
**return** $\frac{\sum_{i=1}^{N} \mathbb{1}\{z_i \in \Sigma\}}{N}$

**Algorithm 6:** EstimateEvent

---

$P_1 \longleftarrow \pi, \mathcal{R}_1 \longleftarrow \mathcal{R}, O_1 \longleftarrow \emptyset$ ;
$k \longleftarrow 1, t_1 = 1, \delta_t = \frac{\delta}{2t^2|\mathcal{Y}|n}$ ;
**for** $t = 1, 2, \ldots$ **do**
    Let $I_t \in \arg\max_{i \in [n] \setminus O_t} \min_{y \in \mathcal{Y}} \text{EstimateEvent}(P_k, \mathcal{R}_t(x_i) \cup \mathcal{R}_t((x_i, y)), \frac{1}{32n^2}, \delta_t)$;
    Query $x_{I_t}$ and observe $y_{I_t}$ and set $O_{t+1} \longleftarrow O_t \cup \{I_t\}$ ;
    Let $\mathcal{R}_{t+1} \longleftarrow \mathcal{R}_t \setminus (\mathcal{R}_t(x_{I_t}) \cup \mathcal{R}_t((x_{I_t}, y_{I_t})))$;
    **if** *EstimateEvent*$(P_k, \mathcal{R}_{t+1}, \frac{1}{8n}, \delta_t) \leq \frac{1}{4n}$ **then**
        $P_{k+1} \longleftarrow \frac{1}{n - |O_{t_k}|} \sum_{l \in [n] \setminus O_{t_k}} \pi_{C_l(O_{t_k})}$;
        $k \longleftarrow k + 1$;
        $t_k \longleftarrow t + 1$;
    **if** *STOP*$(\mathcal{F}_\mathcal{R}, O_t)$ **then**
        **return** $\arg\min_{i \in O_t} y_i$

**Algorithm 7:** `GRAILS` with Estimation

## G  Stopping Condition

Here we briefly give a concrete case for the stopping condition for linear models. Let

$$\mathcal{R} = \{\langle a, \cdot \rangle : a \in \mathbb{R}^d, \|a\| \leq 1\}$$

and $\mathcal{Y} = [K]$ for some $K \in \mathbb{N}$. Suppose that at round $t$ $O_t = \{I_1, \ldots, I_t\}$ and $y_{I_1}, \ldots, y_{I_t}$ have been observed. Suppose $y_{I_1}, \ldots, y_{I_t} \neq 1$ since we are otherwise done and let $\bar{y} = \max_{y < \min_s: y_{I_s}} y$. Fix $l \in [n] \setminus \{I_1, \ldots, I_t\}$. Then, we have that

$$C_l(O_t) = \{a \in \mathbb{R}^d : \|a\| \leq 1, |a^\top x_i - y_i| \leq 1/2 \forall i \in O_t, a^\top x_l \leq \bar{y} + 1/2\}.$$

Determining whether $C_l(O_t)$ is nonempty is a simple convex feasibility problem.

## H  The Algorithms with Approximation

### H.1  Best Arm Identification

Algorithm 7 is the version of Algorithm 1 for best arm identification that estimates the events. It uses the subroutine EstimateEvent, Algorithm 6, to estimate the events. Note that the sampling oracle is called a a number of times that is polynomial in $n$ and polynomial in $|\mathcal{Y}|$.

**Theorem 9.** *With probability at least* $1 - \delta$, *if* $t \geq cv_{best}^* \log(\frac{1}{\mathbb{P}_\pi(S_{f^*})}) \ln(n)$ *where* $c > 0$ *is a universal constant, Algorithm 7 has pulled an arm* $I_s$ *for* $s \leq t$ *such that* $I_s \in \arg\min_{i \in [n]} y_i$.

*Proof of Theorem 9.* The proof is quite similar to the proof of Theorem 2.

**Step 0. Defining the good event.** Define

$$\Sigma_{1,t} = \{\forall y \in \mathcal{Y}, \forall i \in [n] : |\text{EstimateEvent}(P_k, \mathcal{R}_t(x_i) \cup \mathcal{R}_t((x_i, y)), \frac{1}{32n^2}, \delta_t)$$

$$- \mathbb{P}_{f \sim P_k}(r \in \mathcal{R}_t(x_i) \cup \mathcal{R}_t((x_i, y)))| \leq \frac{1}{32n^2}\}$$

$$\Sigma_{2,t} = \{|\text{EstimateEvent}(P_k, \mathcal{R}_{t+1}, \frac{1}{8n}, \delta_t) - \mathbb{P}_{f \sim P_k}(r \in \mathcal{R}_{t+1})| \leq \frac{1}{8n}\}$$

$$\Sigma_t = \Sigma_{1,t} \cap \Sigma_{2,t}$$

$$\Sigma = \cap_{t=1}^\infty \Sigma_t.$$

By the total law of probability, we have that

$$\mathbb{P}(\Sigma^c) \leq \sum_{t=1}^{\infty} \mathbb{P}(\Sigma_t^c | \cap_{s=1}^{t-1} \Sigma_s)$$

$$\leq \sum_{t=1}^{\infty} \mathbb{P}(\Sigma_{t,1}^c | \cap_{s=1}^{t-1} \Sigma_s) + \mathbb{P}(\Sigma_{t,2}^c | \cap_{s=1}^{t-1} \Sigma_s)$$

$$\leq \frac{6}{\pi^2} \sum_{t=1}^{\infty} \frac{\delta}{t^2}$$

$$\leq \delta$$

where the last line follows by the definition of the Algorithm EstimateEvent and standard Hoeffding bounds. For the rest of the proof, we will condition on the event $\Sigma$.

**Step 1.** This step is very similar to step 1 in the proof of Theorem 2. Consider phase $k$, let $t \geq t_k$. Define $\theta \in \mathcal{Y}^n$ by

$$\theta_i = \begin{cases} y_i & i \in O_t \\ \mathrm{argmin}_{y \in \mathcal{Y}} \mathbb{P}_k(r \in \mathcal{R}_t(x_i) \cup \mathcal{R}_t((x_i, y))) & \text{o/w} \end{cases}.$$

By definition of $v_{best}^*$, there exists $\tilde{I} = \{i_1, \ldots, i_{v_{best}^*}\}$ such that there exists $j_0 \in [n] \setminus O_t$ such that

$$\{r \in \mathcal{R}_t : \mathcal{D}[r(x_i)] = \theta_i \; \forall i \in \tilde{I}\} \subset \{r \in \mathcal{R}_t : j_0 \in \mathrm{argmin}_{l \in [n]} \mathcal{D}[r(x_l)]\}$$

Define $I = \{i_1, \ldots, i_{v_{best}^*}, j_0\}$. Then, we have that

$$\mathcal{R}_t = \{r \in \mathcal{R}_t : \exists i \in I \text{ s.t. } i \in \mathrm{argmin}_{l \in [n]} \mathcal{D}[r(x_l)] \text{ or } \mathcal{D}[r(x_i)] \neq \theta_i\}.$$

Define

$$J_t := \arg\max_{i \in I} \mathbb{P}_k(\{r \in \mathcal{R}_t : i \in \mathrm{argmin}_{l \in [n]} \mathcal{D}[r(x_l)] \text{ or } \mathcal{D}[r(x_i)] \neq \theta_i\})$$

$$\bar{y} := \mathrm{argmin}_{y \in \mathcal{Y}} \text{EstimateEvent}(P_k, \mathcal{R}_t(x_{I_t}) \cup \mathcal{R}_t((x_{I_t}, \bar{y}), \frac{1}{32n^2}, \delta_t)$$

Note $J_t$ would be chosen if there were no noise from estimation.

By the event $\Sigma_{1,t}$,

$$\mathbb{P}_k(\{r \in \mathcal{R}_t : J_t \in \mathrm{argmin}_{l \in [n]} \mathcal{D}[r(x_l)] \text{ or } \mathcal{D}[r(x_{J_t})] \neq \theta_{J_t}\})$$

$$\leq \text{EstimateEvent}(P_k, \mathcal{R}_t(x_{J_t}) \cup \mathcal{R}_t((x_{J_t}, \theta_{J_t})), \frac{1}{32n^2}, \delta_t) + \frac{1}{32n^2} \quad (36)$$

$$\leq \text{EstimateEvent}(P_k, \mathcal{R}_t(x_{I_t}) \cup \mathcal{R}_t((x_{I_t}, \bar{y})), \frac{1}{32n^2}, \delta_t) + \frac{1}{32n^2} \quad (37)$$

$$\leq \mathbb{P}_k(\{r \in \mathcal{R}_t : I_t \in \mathrm{argmin}_{l \in [n]} \mathcal{D}[r(x_l)] \text{ or } \mathcal{D}[r(x_{I_t})] \neq \bar{y}\}) + \frac{1}{16n^2} \quad (38)$$

$$\leq \mathbb{P}_k(\{r \in \mathcal{R}_t : I_t \in \mathrm{argmin}_{l \in [n]} \mathcal{D}[r(x_l)] \text{ or } \mathcal{D}[r(x_{I_t})] \neq \theta_{I_t}\}) + \frac{1}{16n^2} \quad (39)$$

where lines (36) and (38) follow from event $\Sigma_{1,t}$, line (37) follows from the definition of $I_t$ and $\bar{y}$, and line (39) used the definition of $\theta_{I_t}$.

From this, it follows that

$$
\begin{aligned}
\mathbb{P}_k(\mathcal{R}_t) &= \mathbb{P}_k(\{r \in \mathcal{R}_t : \exists i \in I \text{ s.t. } i \in \mathrm{argmin}_{l \in [n]} \mathcal{D}[r(x_l)] \text{ or } \mathcal{D}[r(x_i)] \neq \theta_i\}) \\
&\leq \sum_{i \in I} \mathbb{P}_k(\{r \in \mathcal{R}_t : i \in \mathrm{argmin}_{l \in [n]} \mathcal{D}[r(x_l)] \text{ or } \mathcal{D}[r(x_i)] \neq \theta_i\}) \\
&\leq (v_{best}^* + 1)\max_{i \in I} \mathbb{P}_k(\{r \in \mathcal{R}_t : i \in \mathrm{argmin}_{l \in [n]} \mathcal{D}[r(x_l)] \text{ or } \mathcal{D}[r(x_i)] \neq \theta_i\}) \\
&= (v_{best}^* + 1)\mathbb{P}_k(\{r \in \mathcal{R}_t : J_t \in \mathrm{argmin}_{l \in [n]} \mathcal{D}[r(x_l)] \text{ or } \mathcal{D}[r(x_i)] \neq \theta_{I_t}\}) \\
&= (v_{best}^* + 1)[\mathbb{P}_k(\{r \in \mathcal{R}_t : I_t \in \mathrm{argmin}_{l \in [n]} \mathcal{D}[r(x_l)] \text{ or } \mathcal{D}[r(x_i)] \neq \theta_{I_t}\}) + \frac{1}{16n^2}] \quad (40) \\
&\leq (v_{best}^* + 1)[\mathbb{P}_k(\{r \in \mathcal{R}_t : I_t \in \mathrm{argmin}_{l \in [n]} \mathcal{D}[r(x_l)] \text{ or } \mathcal{D}[r(x_i)] \neq y_{I_t}\}) + \frac{1}{16n^2}] \quad (41) \\
&= (v_{best}^* + 1)[\mathbb{P}_k(\mathcal{R}_t(x_{I_t}) \cup \mathcal{R}_t((x_{I_t}, y_{I_t})) + \frac{1}{16n^2}]
\end{aligned}
$$

where line (40) uses (39), and the line (41) comes from the definition of $\theta$. Noticing that

$$
\begin{aligned}
\mathcal{R}_t &= (\mathcal{R}_t(x_{I_t}) \cup \mathcal{R}_t((x_{I_t}, y_{I_t})) \cup \mathcal{R}_t \setminus (\mathcal{R}_t(x_{I_t}) \cup \mathcal{R}_t((x_{I_t}, y_{I_t})) \\
&= (\mathcal{R}_t(x_{I_t}) \cup \mathcal{R}_t((x_{I_t}, y_{I_t})) \cup \mathcal{R}_{t+1},
\end{aligned}
$$

we have that

$$
\mathbb{P}_k(\mathcal{R}_{t+1}) = \mathbb{P}_k(\mathcal{R}_t) - \mathbb{P}_k(\mathcal{R}_t(x_{I_t}) \cup \mathcal{R}_t((x_{I_t}, y_{I_t})) \leq (1 - \frac{1}{v_{best}^* + 1})\mathbb{P}_k(\mathcal{R}_t) + \frac{1}{16n^2}.
$$

Unraveling this recursive statement, we have that

$$
\begin{aligned}
\mathbb{P}_k(\mathcal{R}_{t+1}) &\leq (1 - \frac{1}{v_{best}^* + 1})^{t - t_k} + (t - t_k)\frac{1}{16n^2} \\
&\leq (1 - \frac{1}{v_{best}^* + 1})^{t - t_k} + n\frac{1}{16n^2} \\
&\leq (1 - \frac{1}{v_{best}^* + 1})^{t - t_k} + \frac{1}{16n}.
\end{aligned}
$$

where we used the fact that the game lasts at most $n$ rounds because it is the noiseless setting. Thus, we see that in phase $k$ at $t = t_k + s_k$ with $s_k := O(\ln(n)v_{best}^*)$, we have that

$$
\mathbb{P}_k(\mathcal{R}_{t_k + s_k}) \leq \frac{1}{8n}. \tag{42}
$$

**Step 2.1: Necessary condition for entering the next phase.** Now, we show that if Algorithm 7 enters the phase $k + 1$, then $\mathbb{P}_{f \sim P_k}(r \in \mathcal{R}_{t+1}) < \frac{1}{2n}$. note that if $\frac{1}{4n} \geq$ EstimateEvent$(P_k, \mathcal{R}_{t+1}, \frac{1}{8n}, \delta_t)$, then

$$
\frac{1}{4n} \geq \text{EstimateEvent}(P_k, \mathcal{R}_{t+1}, \frac{1}{8n}, \delta_t) \geq \mathbb{P}_{f \sim P_k}(r \in \mathcal{R}_{t+1}) - \frac{1}{8n}
$$

implying that

$$
\mathbb{P}_{f \sim P_k}(r \in \mathcal{R}_{t+1}) < \frac{1}{2n}.
$$

**Step 2.2: Necessary condition for not entering the next phase.** Now, we show that if $\mathbb{P}_{f \sim P_k}(r \in \mathcal{R}_{t+1}) \geq \frac{1}{2n}$, then Algorithm 7 does not the phase $k + 1$. Suppose $\mathbb{P}_{f \sim P_k}(r \in \mathcal{R}_{t+1}) \geq \frac{1}{2n}$. Then,

$$
\frac{1}{2n} \leq \mathbb{P}_{f \sim P_k}(r \in \mathcal{R}_{t+1}) \leq \text{EstimateEvent}(P_k, \mathcal{R}_{t+1}, \frac{1}{8n}, \delta_t) + \frac{1}{8n},
$$

implying that EstimateEvent$(P_k, \mathcal{R}_{t+1}, \frac{1}{8n}, \delta_t) > \frac{1}{4n}$ and the algorithm does not update the phase.

Given these conditions, the rest of the proof is identical to the proof of Theorem 2.

$\square$

**Input:** distribution $P$ ; event $\Sigma$, multiplicative accuracy $\epsilon \in (0,1)$, failure probability $\delta \in (0,1)$;

$s \longleftarrow 0$;

**while** $\sqrt{\frac{2\log(\frac{s^2\pi^2}{\delta 6})}{s}} \geq \frac{1}{2n^2 max_{y \in \mathcal{Y}} y}$ **do**
    Sample $z_s \sim P$;
    Form $\widehat{\mu}_s = \frac{\sum_{i=1}^{s} \mathbb{1}\{z_i \in \Sigma\}}{s}$;
    **if** $\epsilon \cdot \widehat{\mu}_s \geq \sqrt{2\log(\frac{s^2\pi^2}{\delta 6})/s}$ **then**
        **return** $\widehat{\mu}_s$
    $s \longleftarrow s + 1$;
**return** 0

**Algorithm 8:** EstimateEventMult

---

$P_1 \longleftarrow \pi, \mathcal{R}_1 \longleftarrow \mathcal{R}, O_1 \longleftarrow \emptyset$ ;

$k \longleftarrow 1, t_1 = 1, \delta_t = \frac{\delta}{2t^2|\mathcal{Y}|n}$;

**for** $t = 1, 2, \ldots$ **do**
    Let $I_t \in \text{argmin}_{i \in [n] \setminus O_t} \max_{y \in \mathcal{Y}} \frac{y}{\text{EstimateEventMult}(P_k, \mathcal{R}_t(x_i) \cup \mathcal{R}_t((x_i,y)), \frac{1}{2}, \delta_t)}$;
    Query $x_{I_t}$ and observe $y_{I_t}$ and set $O_{t+1} \longleftarrow O_t \cup \{I_t\}$ ;
    Let $\mathcal{R}_{t+1} \longleftarrow \mathcal{R}_t \setminus (\mathcal{R}_t(x_{I_t}) \cup \mathcal{R}_t((x_{I_t}, y_{I_t})))$;
    **if** $EstimateEvent(P_k, \mathcal{R}_{t+1}, \frac{1}{8n}, \delta_t) \leq \frac{1}{4n}$ **then**
        $P_{k+1} \longleftarrow \frac{1}{n - |O_{t_k}|} \sum_{l \in [n] \setminus O_{t_k}} \pi_{C_l(O_{t_k})}$;
        $k \longleftarrow k + 1$;
        $t_k \longleftarrow t + 1$;
    **if** $STOP(\mathcal{F}_{\mathcal{R}}, O_t)$ **then**
        **return** $\text{argmin}_{i \in O_t} y_i$

**Algorithm 9:** `GRAILS` for Loss Minimization with Estimation

## H.2 Cumulative Loss Minimization

For simplicity, in this section, we assume that $\min_{y \in \mathcal{Y}} y \geq 1$. One can simply add a constant to $\mathcal{R}$ to arrive at this setting. We use the convention that $\frac{1}{0} = \infty$.

**Lemma 1.** *Let $\Sigma$ denote some event that has positive probability under $P$ such that $\epsilon(1-\epsilon)\mathbb{P}(\Sigma) \geq \frac{1}{2n^2 max_{y \in \mathcal{Y}} y}$. With probability at least $1 - \delta$, Algorithm 8 satisfies*

$$(1-\epsilon)EstimateEventMult(P, \Sigma, \epsilon, \delta) \leq P(\Sigma) \leq (1+\epsilon)EstimateEventMult(P, \Sigma, \epsilon, \delta)$$

*Proof of Lemma 1.* Define the event

$$E = \{|P(\Sigma) - \widehat{\mu}_s| \leq \sqrt{\frac{2\log(\frac{s^2\pi^2}{\delta 6})}{s}} \forall s \in \mathbb{N}\}$$

By a union bound and a standard Hoeffding bound, $\mathbb{P}(E) \geq 1 - \delta$. Suppose $E$ holds for the remainder of the proof.

**Step 1.** First, we show that if $\epsilon(1-\epsilon)\mathbb{P}(\Sigma) \geq \frac{1}{n^3 \max_{y \in \mathcal{Y}}}$, then Algorithm 8 does not return 0. Towards a contradiction, suppose the algorithm returns 0 at round $s_0$. Then, we have on the event $E$ that

$$|P(\Sigma) - \widehat{\mu}_{s_0}| \leq \sqrt{\frac{2\log(\frac{s_0^2\pi^2}{\delta 6})}{s_0}} \leq \frac{1}{2n^2 \max_{y \in \mathcal{Y}} y} \leq \epsilon(1-\epsilon)\mathbb{P}(\Sigma).$$

Rearranging we have that

$$P(\Sigma) \leq \frac{1}{1-\epsilon}\widehat{\mu}_{s_0},$$

which implies

$$\sqrt{\frac{2\log(\frac{s^2\pi^2}{\delta 6})}{s}} \leq \frac{1}{2n^2 \max_{y \in \mathcal{Y}}} \leq \epsilon(1-\epsilon)\mathbb{P}(\Sigma) \leq \epsilon\widehat{\mu}_{s_0}.$$

But this implies the algorithm would not have returned 0, yielding a contradiction.

**Step 2.** Suppose that Algorithm 8 does not return 0. Then, there exists a round $s_0 \in \mathbb{N}$ at which

$$\sqrt{\log(\frac{s_0^2 \pi^2}{\delta 6})/s_0} \leq \epsilon \widehat{\mu}_{s_0}.$$

Then, by the event $E$, we have that

$$P(\Sigma) \leq (1 + \epsilon)\widehat{\mu}_{s_0}.$$

The other inequality follows as well, completing the proof. $\qquad\square$

**Theorem 10.** *With probability at least $1 - \delta$, Algorithm 9 incurs a loss of at most $2(v_{loss}^* + max_{i\in[n],f\in\mathcal{F}_\mathcal{R}} f(x_i)) \log(n) \log(\frac{1}{\mathbb{P}(S_{min})})$ to identify $argmin_{i\in[n]} f^*(x_i)$.*

*Proof.* The proof is very similar to the proof of Theorem 4.

**Step 0. Defining the good event.** Define

$$\Sigma_{1,t} = \{\forall y \in \mathcal{Y}, \forall i \in [n] : \mathbb{P}_{f\sim P_k}(f \in \mathcal{R}_t(x_i) \cup \mathcal{R}_t((x_i, y)) \geq \frac{2}{n^2 max_{y\in\mathcal{Y}} y} \implies$$

$$\frac{1}{2}\text{EstimateEventMult}(P_k, \mathcal{R}_t(x_i) \cup \mathcal{R}_t((x_i, y), \frac{1}{2}, \delta_t)$$

$$\leq \mathbb{P}_{f\sim P_k}(f \in \mathcal{R}_t(x_i) \cup \mathcal{R}_t((x_i, y)))$$

$$\leq \frac{3}{2}\text{EstimateEventMult}(P_k, \mathcal{R}_t(x_i) \cup \mathcal{R}_t((x_i, y)), \frac{1}{2}, \delta_t)\}$$

$$\Sigma_{2,t} = \{|\text{EstimateEvent}(P_k, \mathcal{R}_{t+1}, \frac{1}{8n}, \delta_t) - \mathbb{P}_{f\sim P_k}(f \in \mathcal{R}_{t+1})| \leq \frac{1}{8n}\}$$

$$\Sigma_t = \Sigma_{1,t} \cap \Sigma_{2,t}$$

$$\Sigma = \cap_{t=1}^{\infty} \Sigma_t.$$

By the total law of probability, a Standard Hoeffding bound, and Lemma 1, we have that

$$\mathbb{P}(\Sigma^c) \leq \sum_{t=1}^{\infty} \mathbb{P}(\Sigma_t^c | \cap_{s=1}^{t-1} \Sigma_s)$$

$$\leq \sum_{t=1}^{\infty} \mathbb{P}(\Sigma_{t,1}^c | \cap_{s=1}^{t-1} \Sigma_s) + \mathbb{P}(\Sigma_{t,2}^c | \cap_{s=1}^{t-1} \Sigma_s)$$

$$\leq \frac{6}{\pi^2} \sum_{t=1}^{\infty} \frac{\delta}{t^2}$$

$$\leq \delta.$$

For the rest of the proof, we will condition on the event $\Sigma$.

**Step 1:** Define the function $g : \{x_1, \ldots, x_n\} \mapsto \mathcal{Y}$ in the following way:

$$g(x_i) = \begin{cases} \mathcal{D}[\tilde{r}(x_i)] \text{ s.t. } \tilde{r} = \arg\max_{\bar{r}\in\mathcal{R}_t} \frac{\mathcal{D}[\bar{r}(x_i)]}{\mathbb{P}_k(\mathcal{R}_t(x_i)\cup\mathcal{R}((x_i,\bar{f}(x_i)))} & i \notin \{I_1, \ldots, I_{t-1}\} \\ f^*(x_i) & \text{o/w} \end{cases}$$

Define

$$J_t := argmin_{j\in[n]\setminus O_t} \frac{g(x_j)}{\mathbb{P}_k(\mathcal{R}_t(x_j) \cup \mathcal{R}_t((x_i, g(x_j))))},$$

the arm that would be pulled at round $t$ if the probabilities were known exactly. Note that

We have that

$$\frac{g(x_{J_t})}{\mathbb{P}_k(\mathcal{R}_t(x_{J_t}) \cup \mathcal{R}_t((x_i, g(x_{J_t}))))}$$

$$\leq \frac{\sum_{i \in [n] \setminus O_t} g(x_i)}{\sum_{i \in [n] \setminus O_t} \mathbb{P}_k(\{r \in \mathcal{R}_t : \mathcal{D}[r(x_i)] \neq g(x_i) \text{ or } i \in \operatorname{argmin}_{j \in [n]} \mathcal{D}[r(x_j)]\})}$$

$$\leq \frac{\sum_{i \in [n] \setminus O_t} g(x_i)}{\mathbb{P}_k(\mathcal{R}_t)}$$

$$\leq \frac{n \max_{y \in \mathcal{Y}} y}{\mathbb{P}_k(\mathcal{R}_t)}$$

$$\leq n^2 \max_{y \in \mathcal{Y}} y,$$

where the last line uses Step 2.2 in the proof of Theorem 9, which shows that $\mathbb{P}_k(\mathcal{R}_t) \geq \frac{1}{2n}$. This implies that

$$\mathbb{P}_k(\mathcal{R}_t(x_{J_t}) \cup \mathcal{R}((x_i, g(x_{J_t})))) \geq \frac{g(x_{J_t})}{cn^3 \max_{y \in \mathcal{Y}} y} \geq \frac{1}{cn^3 \max_{y \in \mathcal{Y}} y}$$

where we used the fact that $\min_{y \in \mathcal{Y}} y \geq 1$ by assumption.

Thus, by event $\Sigma_{1,t}$, we have that

$$\frac{1}{2}\text{EstimateEventMult}(P_k, \mathcal{R}_t(x_{J_t}) \cup \mathcal{R}_t((x_{J_t}, g(x_{J_t})), \frac{1}{2}, \delta_t)$$

$$\leq \mathbb{P}_{f \sim P_k}(f \in \mathcal{R}_t(x_{J_t}) \cup \mathcal{R}_t((x_{J_t}, g(x_{J_t}))))$$

$$\leq \frac{3}{2}\text{EstimateEventMult}(P_k, \mathcal{R}_t(x_{J_t}) \cup \mathcal{R}_t((x_{J_t}, g(x_{J_t})), \frac{1}{2}, \delta_t)$$

Then, by standard arguments, it can be shown that Algorithm 9 chooses an arm $I_t$ that satisfies

$$\frac{g(x_{I_t})}{\mathbb{P}_{f \sim P_k}(f \in \mathcal{R}_t(x_{I_t}) \cup \mathcal{R}_t((x_{I_t}, g(x_{I_t}))))} \leq c \frac{g(x_{J_t})}{\mathbb{P}_{f \sim P_k}(f \in \mathcal{R}_t(x_{J_t}) \cup \mathcal{R}_t((x_{J_t}, g(x_{J_t}))))}$$

where $c$ is a universal constant.

**Step 2:** The only difference in the proof with Theorem 4 comes in lines (18) and (19). In these lines, we lose a mulitiplicative constant factor from using EstimateEventMult$(P_k, \mathcal{R}_t(x_i) \cup \mathcal{R}_t((x_i, y), \frac{1}{2}, \delta_t)$ instead of the exact probability, as shown in the previous step. The argument for changing phases is identical to the argument in the proof of Theorem 9. The rest of the proof for bounding the cumulative loss incurred is identical.

**Step 3:** The total number of samples taken is a polynomial function of $n$, $\max_{y \in \mathcal{Y}} y$, and $\log(1/\delta)$ by definition of Algorithm 8.

$\square$

# I   Discussion of Function Classes

*Proof of Proposition 5.* $\mathcal{F}_{\mathcal{R}}$ consists of functions $f_1, \ldots, f_n$ of the following form

$$f_j(x_i) \begin{cases} 10 & i = j+1 \\ 1 & i \geq j \\ 0 & i < j \end{cases}.$$

Suppose that $g : \mathcal{X} \mapsto \mathcal{Y}$ is such that $g \notin \mathcal{F}_{\mathcal{R}}$. Then, a simple computation shows that at most 3 measurements are required to eliminate all functions from the version space. On the other hand, if $g = f_j$ for some $f_j \in \in \mathcal{F}_{\mathcal{R}}$, then it suffices to query $x_j$ and $x_{j-1}$. Therefore, a simple computation shows that $v_{best}^*(\mathcal{F}_{\mathcal{R}}) \leq 3$.

Since $\pi$ is uniform over $[0,1]$, we have that $\mathbb{P}_{\pi}(S_{f^*}) \geq \frac{c}{n}$ for some positive universal constant. Thus, the result follows.

$\square$

**Linear Separators $\epsilon$-good arm identification:** We give an intuitive sufficient condition for lower bounding $\mathbb{P}_\pi(S_f)$.

**Assumption 1.** *Fix $y_1, \ldots, y_n \in \mathcal{Y}$. Define the* minimum margin*:*

$$\gamma^* := max_{r^* \in \mathcal{R} : \mathcal{D}[r^*(x_i)] = y_i \forall i \in [n]} \min_{i \in [n]} \min_{y \neq y_i} |r^*(x_i) - y| - |r^*(x_i) - y_i|.$$

The following Proposition lower bounds $\mathbb{P}_\pi(S_{f^*})$ for linear functions.

**Proposition 13.** *Let $\mathcal{R} = \{\langle a, \cdot \rangle : a \in \mathbb{R}^d, \|a\| \leq R_1\}$. Let $x_1, \ldots, x_n \in \mathbb{R}^d$ such that $max_{i \in [n]} \|x_i\|_* \leq R_2$ where $\|\cdot\|_*$ denotes the dual norm of $\|\cdot\|$. Let $y_1, \ldots, y_n \in \mathcal{Y}$. Suppose that $\gamma^* > 0$. Let $f^* = \mathcal{D}[\langle a_*, \cdot \rangle]$ for some $a_*$ with $\|a_*\| \leq R_1$. Let $\pi$ be the uniform distribution over $\{a : \|a\| \leq R_1\}$. Then,*

$$\mathbb{P}_\pi(S_{f^*}) \geq \frac{volume(\{z \in \mathbb{R}^d : \|z\| \leq \frac{\gamma^*}{R_2}\})}{volume(\{z \in \mathbb{R}^d : \|z\| \leq R_1\})}.$$

*In particular, if $\|\cdot\| = \|\cdot\|_2$, then $\mathbb{P}_\pi(S_{f^*}) \geq (\frac{\gamma^*}{R_1 R_2})^d$.*

The above Proposition shows that for linear separators with a Euclidean constraint and a margin $\gamma^*$, $\log(\frac{1}{\mathbb{P}_\pi(S_{f^*})}) \lesssim d \log(\gamma^*)$.

*Proof of Proposition 13.* By assumption $\gamma^* > 0$. Define $\delta_0 = \frac{\gamma^*}{6R_2}$. Let $f^*(\cdot) = \langle a_*, \cdot \rangle$ where $a_* \in \mathbb{R}^d$ and $\|a_*\| \leq R_1$. Define

$$\bar{B}_\alpha = B_{\delta_0}(\alpha a_*)$$

where $B_\gamma(v) := \{v \in \mathbb{R}^d : \|v\| \leq \gamma\}$.

**Step 1. $\bar{B}_\alpha$ is feasible.** We claim that for all $\alpha \in [0, 1 - \delta_0/R_1]$, $\bar{B}_\alpha \subset \{a : \|a\| \leq R_1\}$. Let $\alpha \in [0, 1 - \delta_0/R_1]$. Let $v$ such that $\|v\| \leq \delta_0$. Then,

$$\|\alpha a_* + v\| \leq \alpha \|a_*\| + \|v\| \leq (1 - \delta_0/R_1) \|a_*\| + \|v\| \leq R_1 - \delta_0 + \delta_0 = R_1,$$

showing the claim.

**Step 2. $a_* \in \bar{B}_{\alpha_0}$ for some $\alpha_0$** Let $\alpha_0 = \max(1 - \delta_0/\|a_*\|, 0) \leq 1 - \delta_0/R_1$. We claim that $a_* \in \bar{B}_{\alpha_0}$. If $1 - \delta_0/\|a_*\| \geq 0$, then

$$\|(\alpha_0 - 1)a_*\| = \frac{\delta_0}{\|a_*\|} \|a_*\| = \delta_0$$

so that $a_* \in \bar{B}_{\alpha_0}$. On the other hand, if $1 - \delta_0/\|a_*\| < 0$, then $\|a_*\| < \delta_0$, so that $a_0 \in \bar{B}_{\alpha_0}$ since $\alpha_0 = 0$.

**Step 3. Putting it together.** Now, let $a \in \bar{B}_{\alpha_0}$. Then,

$$\|a_* - a\| \leq 2\delta_0.$$

Then, using Holder's inequality,

$$|a_*^\top x_i - (a^\top x_i)| \leq \|a_* - a\| \max_i \|x_i\|_* \leq 2\delta_0 R_2 \leq \frac{\gamma^*}{3}$$

by definition of $\delta_0$.

Let $y \neq y_i$. Then,

$$|\langle a, x_i \rangle - y| - |\langle a, x_i \rangle - y_i| > |\langle a_*, x_i \rangle - y_i| - |\langle a_*, x_i \rangle - y_i| - \frac{2}{3}\gamma^*$$

$$> 0$$

where the last line uses the definition of $\gamma^*$. Thus, we have that if $f = \langle a, \cdot, \rangle \in S_{f^*}$. Thus,

$$\mathbb{P}_\pi(S_{a_0}) \geq \frac{volume(\{z \in \mathbb{R}^d : \|z\| \leq \delta_0\})}{volume(\{z \in \mathbb{R}^d : \|z\| \leq R_1\})}.$$

The result for the case $\|\cdot\| = \|\cdot\|_2$ follows from standard formula for the volume of an $n$-dimensional ball.

$\square$

Next, we give a simple bound on $v_{best,\epsilon}^*$ for linear functions. For simplicity, we let $|\mathcal{Y}|$ be countably infinite, but it could be extended to the $|\mathcal{Y}| < \infty$ case.

**Proposition 14.** *Suppose $\mathcal{Y} = \{\ldots, -2\Delta, -\Delta, 0, \Delta, 2\Delta, \ldots\}$ where $\Delta > 0$. Let $\mathcal{R} = \{r : r(\cdot) = \langle a, \cdot \rangle, a \in \mathbb{R}^d\}$. Let $x_1, \ldots, x_n \in \mathbb{R}^d$ such that $max_{i \in [n]} \|x_i\|_i \leq 1$. Define*

$$\phi := \min_{i_1, \ldots, i_d \in [n]} \sigma_d \left( \begin{pmatrix} x_{i_1}^\top \\ \vdots \\ x_{i_d}^\top \end{pmatrix} \right)$$

*where $\sigma_d(A)$ denotes the dth singular value of a matrix $A$. Then if $\Delta \leq \min(\frac{\phi\epsilon}{4\sqrt{d}}, \frac{\epsilon}{2})$, $v_{best,\epsilon}^* \leq d$.*

This result, combined with Proposition 13, suggests that `GRAILS` passes the sanity check of not sampling too many linearly dependent arms for the linear functions case.

*Proof of Proposition 14.* **Step 1:** Let $g : \mathbb{R}^d \mapsto \mathcal{Y}$ be an arbitrary mappying. By assumption, there exists $i_1, \ldots, i_d \in [n]$ such that defining $\bar{X} := \begin{pmatrix} x_{i_1}^\top \\ \vdots \\ x_{i_d}^\top \end{pmatrix}$, we have that $\sigma_d(\bar{X}) = \phi$. Suppose that the algorithm queries $i_1, \ldots, i_d$. If there is no $a_* \in \mathbb{R}^d$ such that $g(x_{i_j}) = \mathcal{D}[a_*^\top x_{i_j}]$ for all $j \in [d]$, then we are done. So, assume that there exists $a_* \in \mathbb{R}^d$ such that $g(x_{i_j}) = \mathcal{D}[a_*^\top x_{i_j}]$ for all $j \in [d]$. Suppose wlog that $x_1 \in argmin_{i \in [n]} a_*^\top x_i$. Now, we will show that all models consistent with $g(x_{i_j})$ for all $j \in [d]$ put $x_1$ as an $\epsilon$-good arm. Let $a \in \mathbb{R}^d$ be another vector such that

$$\mathcal{D}[a^\top x_{i_j}] = g(x_{i_j})$$

for all $j \in [d]$. Then, we have that

$$\Delta\sqrt{d} \geq \left\| \bar{X}(a_* - a) \right\| \geq \sigma_d(\bar{X}) \|a - a_*\|.$$

Let $x_i \neq x_1$. Note that by definition of the rounding operator and $\mathcal{Y} = \{\ldots, -2\Delta, -\Delta, 0, \Delta, 2\Delta, \ldots\}$,

$$a_*^\top x - \frac{\Delta}{2} \leq \mathcal{D}[a_*^\top x] \leq a_*^\top x + \frac{\Delta}{2}$$

for all $x$.

We have that

$$\mathcal{D}[a^\top x_1] - \mathcal{D}[a^\top x_i] - \Delta \leq a^\top(x_i - x_1) \leq (a - a_*)^\top(x_1 - x_i) + a_*^\top(x_1 - x_i)$$
$$\leq \|a - a_*\| \|x_i - x_1\| + a_*^\top(x_1 - x_i)$$
$$\leq \frac{\Delta\sqrt{d}}{\phi} 2 + a_*^\top(x_1 - x_i)$$
$$\leq \frac{\epsilon}{2}$$

where we used $a_*^\top x_1 - a_*^\top x_i \leq 0$ by definition of $x_1$ and the setting of $\Delta$. Thus, rearranging, we have that

$$\mathcal{D}[a^\top x_1] - \mathcal{D}[a^\top x_i] \leq \Delta + \frac{\epsilon}{2} \leq \epsilon$$

implying that $x_1$ is an $\epsilon$-good arm for any $a$ such that

$$\mathcal{D}[a^\top x_{i_j}] = g(x_{i_j}).$$

This completes the proof.

$\square$

**Active Classification with Linear Separators:** Next, we lower bound the minimum probability for linear separators under a certain geometric that can be interpreted as a quantitative version of the concept of general position. Define

$$\Gamma(\mathcal{X}) = \{(y_1, \ldots, y_n)^\top \in \{-1, 1\}^n : \exists w \in \mathbb{R}^d, \, b \in \mathbb{R} \text{ s.t. } \text{sign}(w^\top x_i + b) = y_i \, \forall i \in [n]\}.$$

**Proposition 15.** *Let $\gamma > 0$. Suppose $\mathcal{X} = \{x_1, \ldots, x_n\}$ satisfy $\forall S \subset \mathcal{X}$ such that $|S| \leq d - 1$,*

$$\min_{x \in \mathcal{X} \setminus S} \text{dist}(x, \text{aff}(S)) \geq \gamma.$$

*Suppose $\max_{i \in [n]} \|x_i\| \leq R$. Let $\mathcal{F}_R = \{\langle w, \cdot \rangle + b : (w^\top, b)^\top \in W\}$ and $W = \{(w^\top, b)^\top : \|w\|_2^2 + b^2 \leq R^2\}$. where $R$ is large enough such that for every $(y_1, \ldots, y_n)^\top \in \Gamma(\mathcal{X})$, the max-margin separator achieving $(y_1, \ldots, y_n)^\top$ belongs to $\mathcal{F}_R$. Let $\pi$ denote the uniform probability distribution over $W$. Then,*

$$\min_{(y_1, \ldots, y_n)^\top \in \Gamma(\mathcal{X})} \mathbb{P}_{(w^\top, b)^\top \sim \pi}(\text{sign}(w^\top x_i + b) = y_i \, \forall i \in [n]) \geq \left(\frac{\gamma}{R}\right)^d$$

Thus, under the condition in the Proposition 15, the sample complexity in Theorem 8 scales as $cd \log(\gamma) v_{class}^*(\mathcal{F}_R)$. Often $\log(|\mathcal{F}_R|) = \Omega(d)$, in which case Theorem 8 is larger than the active classification upper bound in [21] by a factor of $\log(\gamma)$. On the other hand, Algorithm 5 is computationally efficient.

*Proof of Proposition 15.* Fix $(\bar{y}_1, \ldots, \bar{y}_n)^\top \in \Gamma(\mathcal{X})$. Let $f = \langle \bar{w}, \cdot \rangle + \bar{b} \in \mathcal{F}_B$ attain the max-margin separator achieving $(\bar{y}_1, \ldots, \bar{y}_n)^\top$, which exists by assumption. Let $v_1, \ldots, v_l$ denote the support vectors with positive labels and $u_1, \ldots, u_k$ the support vectors with negative labels. By definition of support vectors, we have that

$$\text{aff}(v_1, \ldots, v_l) = \text{aff}(u_1, \ldots, u_k) + z$$

for some $z \in \mathbb{R}^d$. We claim that $\|z\|_2 \geq \gamma$. Towards a contradiction, suppose not. Then,

$$\text{dist}(v_1, \text{aff}(u_1, \ldots, u_k)) = \|z\|_2 < \gamma$$

which contradicts our assumption. This implies that the margin is at least $\gamma/2$ and that

$$|\bar{w}^\top x - \bar{b}| \geq \frac{\gamma}{2}$$

for all $x \in \mathcal{X}$. Then, for any $w, b$ such that $\|w\|_2^2 + b^2 \leq [\gamma/4]^2$, we have that

$$|\bar{w}^\top x - \bar{b} - [(\bar{w} + w)^\top x - \bar{b} - b]| = |w^\top x - b| \leq 2 \left\|(w^\top, b)^\top\right\|_2 \leq \frac{\gamma}{2}$$

The result follows by Proposition 13. $\qquad\square$

**Strongly Convex Functions:** Finally, we consider strongly convex functions. The following Proposition considers a setting of strongly convex functions where $v_{best}^* \leq O(1)$.

**Proposition 16.** *Let $K \in \mathbb{N}$ and let $\mathcal{X} = [K]$. Let $\mathcal{R} = \{r : r \text{ is } \alpha\text{-strongly convex}\}$. Let $\Delta = \min_{y \neq y' \in \mathcal{Y}} |y - y'|$. If $\alpha \geq 2\Delta$ Then, $v_{best}^*(\mathcal{F}_\mathcal{R}) \leq 3$.*

*Proof.* **Step 1:** Suppose that $g : \mathcal{X} \mapsto \mathcal{Y}$ such that there exists $y \in \mathcal{Y}$ such that $g(x) = y$ for all $x \in \mathcal{X}$. Now, let $x_0 \in \{2, \ldots, K - 1\}$. Let $r$ be any $\alpha$-strongly convex function defined over $[1, K]$. Then, $x_0 \in \text{intdom}(r)$. Then, there exists a subgradient at $x, g$. Let $z_1 = x_0 - 1$ and $z_2 = x_0 + 1$. Then, either $g \cdot (z_1 - x) \geq 0$ or $g \cdot (z_2 - x_0) \geq 0$. Suppose wlog that $g \cdot (z_1 - x_0) \geq 0$. Then, using $\alpha$-strong convexity we have that

$$r(z_1) \geq r(x_0) + g \cdot (z_1 - x_0) + \frac{\alpha}{2}|x_0 - z_1| \geq r(x_0) + 2\Delta$$

where we used $\alpha \geq 2\Delta$. This shows that for any $\alpha$-strongly convex function defined over $[1, K]^d$, either $\mathcal{D}[r(z_1)] \neq \mathcal{D}[r(x_0)]$ or $\mathcal{D}[r(z_1)] \neq \mathcal{D}[r(x_0)]$. Thus, it suffices to query $x_0, z_1, z_2$ to finish this case.

**Step 2:** Now, suppose that $g : \mathcal{X} \mapsto \mathcal{Y}$ is such that there exists $x, x' \in \mathcal{X}$ such that $g(x) \neq g(x')$. Then, let $I \in \text{argmin}_{i \in [n]} g(x_i)$. It suffices to query $x_{I-1}, x_I$, and $x_{I+1}$. By the prior step, we have that $g(x_{I-1}) > g(x_I) < g(x_{I+1})$ for any $\alpha$-strongly convex function such that $\mathcal{D}[r(x)] = g(x)$ for all $x \in \mathcal{X}$. Thus, this completes the case.

$\qquad\square$

## J   `OFUL` **is not minimax-optimal**

Fix $x_1, \ldots, x_{2n}$ and let $\mathcal{F} = \{f_1, \ldots, f_n\}$ where

$$f_i(x_j) = \begin{cases} 1 & j = i \\ .5 & j \in \{n+1, \ldots, n+i\} \\ 0 & \text{o/w} \end{cases}.$$

Consider the instance given by $y_i = f_n(x_i)$ for all $i \in [2n]$. Then, inspection reveals that `OFUL` selects all of the arms in $\{1, \ldots, n\}$ before terminating, obtaining a sample complexity of $n$. On the other hand, it is possible to solve any instance $f \in \mathcal{F}$ in $O(\log(n))$ queries using a binary search procedure..

## K   Additional Experiment and Implementation Details

All experiments were computed on a 36 core cluster machine, and all algorithms were implemented in python. To estimate the probabilities in the algorithm for sample selection, we used 300 independently drawn functions from the version space in the RKHS and constraints experiments and 500 in the case of convex functions. We used 50 iterations of hit and run for the RKHS and constraints experiments and 100 for the convex function experiment. Note that `GRAILS` divides the output space into intervals. In the RKHS experiments, we used 100 evenly spaced intervals in the output space of $f^*$ for computational efficiency. For the constraints experiment, we divided the output space of $f^*$ into 300 spaced for better sample complexity at the cost of computational efficiency. Lastly in the convex experiment, we used 1500 evenly spaced intervals.

Furthermore, note that hit and run must compute a projection onto the boundary of the convex set in order to choose the random step. As the step is along a uniformly chosen direction, we can compute this projection by bisection search. This was performed with tolerance 1e-6 for suitable performance with minimal computational overhead.

Lastly, our general implementation of hit and run requires a membership oracle for each function class. We implemented oracles for RKHS functions and 1d convex functions. In the case of the RKHS, we linearize the problem using the kernel trick by first computing the Grammian matrix and then projecting all data onto the span of the eigenvalues. Furthermore, in order to use hit and run, the MCMC iteration must begin with a point in the convex set. As our set is cut out by hyperplane and elliptical constraints, we achieve this by choosing a random starting point in and using CVXPy with the ECOS solver to compute the projection of that point into the convex set to generate a random stating point for hit and run.

For our implementation, we followed the update of [23] directly in the setting where the noise variance is $0$. For our implementation of OFUL, we track our constraint sets using equality constraints for the observed data and norm constraints for the norm of the unknown parameters. Then we use CVXPy to compute the upper and lower bound on the function value of any point $x_i$.