# OpenReview forum: "Practical, Provably-Correct Interactive Learning in the Realizable Setting: The Power of True Believers"
_NeurIPS.cc/2021/Conference — NeurIPS 2021 Poster_

### Official Review · Reviewer_QoM1 · 2021-07-08

**Rating:** 6
**Confidence:** 4

**Summary:**

This paper presents a general algorithmic framework for interactive learning, which applies to bandit and active classification.

**Limitations And Societal Impact:**

yes

**Main Review:**

The paper positions its main contribution in a general framework for computationally efficient interactive learning, while making a strong assumption that the data are realizable.

It is okay to confine to realizable setting but I am not convinced of the motivating example. The starting point is that for learning of threshold over a bounded interval, agnostic learner is inferior to realizable learner; thus authors decide to stick with the latter. I agree this example is true but it turns out that such motivation is superficial and seems diminish tremendous efforts in agnostic learning in the past decades. Furthermore, the nice query complexity shown in the example seems just a singular case; same result does not hold for more general and practical setup such as halfspace in $R^d$.

I also found the presentation is confusing. It is claimed to develop a general framework for active classification, but neither Algorithm 1 nor 2 is committed to it.

For best arm identification, the dependence on sampling probability of $f^*$ is non-standard and is of little interest since it is unknown.

It is unfair (in fact, misleading) to compare Theorem 5 to [11] because Theorem 5 follows from an inefficient algorithm. What you can really do is to compare Theorem 4 to [11]. However, in this case, the above issue persists so there is no evidence that your algorithm outperforms.

My overall feeling is that the paper is too dense to follow, and the major technical contribution is vague. While it is interesting to see results for general loss functions, it leads an unknown quantity (i.e. sampling probability) in sample complexity so the bound is of little interest.

## updates after author rebuttal

Authors addressed most of my concerns. Authors also promise to state the results (i.e. "minimax optimal") in a more precise way. I am thus happy to raise my score.


**Time Spent Reviewing:**

5

---

> ### Author Response · Authors · 2021-08-09
> **Response**
>
> Thank you for your review. We appreciate your careful reading of our work. In the next version of this paper, we will work to make the paper less dense via more explanatory examples, a clearer statement of its technical contributions, and providing more intuition on mathematical statements. Please see responses at the top for worked out examples of computational and sample complexity bounds and a discussion of the sampling probability of $f^\ast$.
>
> Regarding past work on agnostic active learning: we agree that agnostic active learning is a very important and practically relevant area. The core importance of our Propositions 1-3 is to show however that agnostic learning is not minimax optimal in the realizable setting.
>
>
> Regarding "no free lunch'' for other classes of functions: the gap between realizable and agnostic algorithms can in fact be shown to exist even for homogeneous linear separators since one can embed the simple instance from Proposition 1 into the homogeneous linear separator setting. Consider points forming an $\epsilon$-cover of the union of the Euclidean unit ball $B_1$ centered at $0$ and a second ball, $B_2$, disjoint from $B_1$. There exists a realizability-based algorithm that identifies the optimal homogeneous linear separator in $O(d\log(1/\epsilon))$ queries whereas using our lower bound techniques it can be shown that an agnostic approach necessarily requires $\Omega(d\cdot \text{Poly}(1/\epsilon))$ queries. We also note that the instance in Proposition 1 is not unique to combinatorial settings and, in fact, was inspired by an example constructed for linear separators in $\mathbb{R}^3$ in [17].
>
>
>
>
> Regarding the result of [11]: we wish to stress that the result of [11] is in fact not computationally efficient since it assumes that one can enumerate the function class which is intractable even for linear classes. One of our main contributions is building the first computationally efficient algorithm that is nearly minimax optimal and solves the problem stated in [11].
>
> Regarding a comparison between [11] and Theorem 5: since the algorithm of [11] and our Algorithm 3 from Theorem 5 are both computationally infeasible, this is a fair comparison between algorithms with similar computational costs. Part of the confusion may be due to a typo we found in the statement of Theorem 5. It should be referring to Algorithm 3 as opposed to Algorithm 2. We apologize for the mistake.
>
>
> Regarding results for active classification presented in our paper: please see the comment to all reviewers about the problem settings and contributions of our paper. This paper develops a general technique for efficient interactive learning. Active classification is one example setting we consider alongside best arm identification, loss minimization, $\epsilon$-good identification, and thresholding. Due to space constraints, it was placed in the appendix, but if there is room for it, we agree that it would be good to include in the main body.

---

> > ### Comment · Reviewer_QoM1 · 2021-08-11
> > **reviewer comments**
> >
> > Thank you for the response. It partially addressed my concerns.
> >
> > I am still not convinced of the dependence on $f^*$. The fact that some prior works have similar dependence is not an excuse, because this quantity is not fundamental - I think it only exists in a very tiny fraction of the research papers in the literature.
> >
> > Another concern is that while you developed a unified approach, comparison to state-of-the-art approaches that are specifically designed for each separate problem is vague and seems incomparable due to e.g. nonstandard dependence on $f^*$.
> >
> > From line 240 - 245, it appears that the paper resolves an open question due to Amin et al. 2011 [11]. Is this the first kind of efficient and minimax optimal algorithm for generic function? Is [11] the state of the art prior to this work?

---

> > > ### Author Response · Authors · 2021-08-12
> > > **Follow-up Response**
> > >
> > > Thank you for your follow-up questions and remarks.
> > >
> > > You write: "From line 240 - 245, it appears that the paper resolves an open question due to Amin et al. 2011 [11]. Is this the first kind of efficient and minimax optimal algorithm for generic function? Is [11] the state of the art prior to this work?" Yes, [11] is the prior state-of-the-art. It gave a nearly minimax optimal algorithm for generic function classes that is computationally inefficient. We provide the first nearly minimax optimal algorithm  for generic function classes that is computationally efficient. Therefore, we consider our algorithm state-of-the-art.
> > >
> > > You write: "I am still not convinced of the dependence on $f^*$. The fact that some prior works have similar dependence is not an excuse, because this quantity is not fundamental - I think it only exists in a very tiny fraction of the research papers in the literature." We agree that this dependence exists in a fraction of the literature, but point out that it exists for a large fraction of the literature on algorithms that are both computationally efficient and minimax optimal in a strong sense such as the one defined in our paper, that is, for a fixed function class $\mathcal{F}$ and set of points $x_1,\ldots, x_n$. To the best of our knowledge, all the works that are computationally efficient and nearly minimax optimal in such a strong sense for their respective settings [15,18] have a similar dependence on $f^*$.
> > >
> > > You write: "Another concern is that while you developed a unified approach, comparison to state-of-the-art approaches that are specifically designed for each separate problem is vague and seems incomparable due to e.g. nonstandard dependence on $f^*$". We emphasize that for best arm identification, cumulative loss minimization, and active classification in pool-based setting with generic function classes, there is no nearly minimax optimal algorithm that is computationally efficient, thereby making our algorithms state-of-the-art for these problems. We are only aware of a nearly minimax optimal computationally efficient algorithm for active classification in the streaming setting (unlike the pooled setting in our paper), but it has a similar dependence on the sampling probability of the true function $f^*$. We'd also like to emphasize that we have the state-of-the-art bound for regret minimization with generic function classes (for computationally inefficient algorithms) with no dependence on the sampling probability of $f^*$ where the prior state-of-the-art is [11] (as shown by Propositions 7 and 8, Theorem 5, and our Algorithm 3).

---

> > > > ### Comment · Reviewer_QoM1 · 2021-08-12
> > > > **reviewer comments**
> > > >
> > > > Thank you for the response.
> > > >
> > > > If I understood correctly, you improve [11] in terms of computational efficiency but the algorithm needs to access an unknown (and unrealistic) sampling probability of the groundtruth $f^*$.
> > > >
> > > > My another concern is that your sample complexity is
> > > >
> > > > $HD(F_R) \log(\|{F_R}\|) \cdot \log(1/P(f^*)) \cdot \log n$,
> > > >
> > > > while that of [11] is
> > > >
> > > > $HD(F_R) \log(\|{F_R}\|)$
> > > >
> > > > which is better than yours. Therefore, I was not able to follow why you can claim your result is minimax optimal.
> > > >
> > > > I think to establish minimax optimality of your result, you will have to show something like any computationally *efficient* algorithm must draw the number of samples you gave in the paper.

---

> > > > > ### Author Response · Authors · 2021-08-12
> > > > > **Response**
> > > > >
> > > > > You write: "If I understood correctly, you improve [11] in terms of computational efficiency but the algorithm needs to access an unknown (and unrealistic) sampling probability of the groundtruth $f^*$." This is incorrect. Our algorithm does not need knowledge of the sampling probability of $f^*$. However, our algorithm does improve on [11] in terms of computational efficiency.
> > > > >
> > > > > You write: "I was not able to follow why you can claim your result is minimax optimal. I think to establish minimax optimality of your result, you will have to show something like any computationally efficient algorithm must draw the number of samples you gave in the paper." We establish near minimax optimality. The minimax lower bound is $\text{HD}(\mathcal{F}_R)$. Our algorithm achieves $\text{HD}(\mathcal{F}_R)$ up to logarithmic factors and in this sense is nearly minimax optimal. We will make sure this is clear in the next version of the paper.
> > > > >
> > > > > We agree that minimax lower bounds for computationally efficient algorithms would be interesting, but we are unaware of any such bounds in the literature.

---

> > > > > > ### Comment · Reviewer_QoM1 · 2021-08-12
> > > > > > **reviewer comment**
> > > > > >
> > > > > > Note that Theorem 2 holds only when Algorithm 1 runs for sufficient iterations $t$ which is a function of $f^*$. In other words, if the sampling probability of $f^*$ were unknown, your algorithm would not be able to terminate at right time.
> > > > > >
> > > > > > You bound essentially scales with the inverse of the sampling probability and a factor of $\log(n)$. Without any condition, say some constant lower bound on the sampling probability, the so-called 'minimax optimal bound' can be arbitrarily worse than [11] and way off the lower bound.
> > > > > >
> > > > > > PS. when people say they achieve optimal bound $g$ up to log factors, they often refer to an obtained bound with the form $g \cdot \polylog(g)$.

---

> > > > > > > ### Author Response · Authors · 2021-08-12
> > > > > > > **Response**
> > > > > > >
> > > > > > >
> > > > > > > Thank you for your comments.
> > > > > > >
> > > > > > > You write: "Note that Theorem 2 holds only when Algorithm 1 runs for sufficient iterations  which is a function of $f^*$. In other words, if the sampling probability of $f^*$ were unknown, your algorithm would not be able to terminate at right time." Thank you for clarifying your comment. We agree that our guarantee on Theorem 2 does not say that the algorithm stops after the given number of queries. We can guarantee that the algorithm will terminate after at most $\upsilon^*_{\textit{best}} \log(n)\log(\frac{1}{\text{minimum sampling probability of } f}) $ queries using the stopping condition in the penultimate line of Algorithm 1 $\text{STOP}(\mathcal{F}_{\mathcal{R}}, O_t)$, which we describe in more detail in the appendix. We note that these sample complexity results match similar results in the literature [15,18]. We agree that it is an interesting and important avenue of future research how to improve on these results for the stopping time.
> > > > > > >
> > > > > > > Here we briefly overview the stopping condition: We emphasize that the stopping condition requires no knowledge of the sampling probabilities. The stopping condition checks whether the version space is empty and terminates if it is. The version space is a union of $O(n)$ convex sets and thus it suffices to solve at most $n$ convex feasibiliy problems. In the cases that we focus on in this paper such as linear models, kernel methods, and convex functions, this is a tractable problem.
> > > > > > >
> > > > > > > Thank you for the comment, and we will make this clearer in the next version of the paper.
> > > > > > >
> > > > > > > You write: "You bound essentially scales with the inverse of the sampling probability and a factor of $\log(n)$. Without any condition, say some constant lower bound on the sampling probability, the so-called 'minimax optimal bound' can be arbitrarily worse than [11] and way off the lower bound." As we stated earlier, all results that are computationally efficient and nearly minimax optimal in a strong sense as described in this paper also depend logarithmically on the inverse of the sampling probability. We agree that it is an important open problem to determine whether it is possible to remove such a factor and to do so if it is. We emphasize that prior to our work, there was no algorithm for generic function classes that was computationally efficient and whose query complexity could be related to the minimax lower bound.
> > > > > > >
> > > > > > > You write: "PS. when people say they achieve optimal bound  up to log factors, they often refer to an obtained bound with the form $g \text{polylog}(g)$." Thank you for the comment, and we will use this notation to make the result clearer in the next version of the paper.

---

> > > > > > > > ### Comment · Reviewer_QoM1 · 2021-08-14
> > > > > > > > **reviewer comments**
> > > > > > > >
> > > > > > > > Thank you for the clarification. I am now convinced that the algorithm is practically implementable (and I strongly suggest you highlight this technical detail).
> > > > > > > >
> > > > > > > > I think you should be more precise when arguing "minimax optimal". Virtually you attain the optimal rate/match the rate of an inefficient algorithm [11] only when $n$ is a constant and the sample probability of $f^*$ is $\Omega(1)$ - this needs to be highlighted in abstract/introduction/theorem statements, just to avoid overclaiming the results.
> > > > > > > >
> > > > > > > > I would be happy to increase my score if authors will incorporate them into the revision.

---

> > > > > > > > > ### Author Response · Authors · 2021-08-14
> > > > > > > > > **Response**
> > > > > > > > >
> > > > > > > > > Yes, we will happily incorporate your suggestions. It has been very helpful to hear which points of the work are confusing and how we could better explain and contextualize our results. Thank you for your thoughtful feedback and for improving the paper. We very much appreciate you raising your score.

---

### Official Review · Reviewer_Tyzu · 2021-07-14

**Rating:** 8
**Confidence:** 3

**Summary:**

The authors prove that minimax optimal interactive learning algorithms must be realizability-based. They present 2 efficient algorithms for this setting and prove they are approximately minimax optimal.


**Ethical Concerns:**

None.

**Limitations And Societal Impact:**

The biggest limitations to this work are the computational complexity of the "efficient" algorithms, and the restriction to the noise-free setting. Both of which are noted by the authors.

**Main Review:**

Active learning is the classic approach for reducing label complexity during model learning. Years ago, many empirical experiments showed little improvement over random sampling which led to the suspicion that active learning doesn't work. It has since been demonstrated that active learning algorithms do exist which learn at an exponential rate [see the work of Balcan et al.]. This paper continues that line of work, showing that realizability is a key ingredient in active learning.

Overall the paper is well structured, but notation dense. It would be easier to read if more equations were followed with "in words" or (approximate) conceptual descriptions. Also, previews such as "now we prove such and a such result, the approach is to do this and that" would increase readability. I recognize however, these additions are likely not feasible due to the length constraints.

The empirical experiments are a little light. Again, length constraints likely inhibit further experiments.


**Time Spent Reviewing:**

6

---

> ### Author Response · Authors · 2021-08-09
> **Response**
>
> Thank you for your review and for the suggestions to improve clarity. We agree with your discussion placing this work in the context of the literature, and will include a similar discussion in the next version of the paper. We very much appreciate your concrete suggestions for improving the clarity and readability, and will incorporate them into the next version of the paper.

---

### Official Review · Reviewer_JqDF · 2021-07-16

**Rating:** 7
**Confidence:** 4

**Summary:**

Consider the problem of best arm identification: there are $n$ arms, each with a reward, and the goal is to select the best arm  with the best reward with as few pulls as possible. The query complexity of this specific problem in the realizable setting (where rewards correspond to an unknown function from a known family) is well understood (up to log factors), but there is no matching efficient algorithm. This paper provides nearly optimal algorithms for classes of functions that are convex (up to a discretization). Algorithms for related interactive learning problems are also considered. Finally, the authors present lower bounds for such problems that show that agnostic algorithms (not assuming realizability) can be much worse.

**Ethical Concerns:**

None.

**Limitations And Societal Impact:**

Yes.

**Main Review:**

Interactive learning (in the sense discussed in this paper) consists of gradually querying a set of points $x_1,x_2,\dots,x_n$ so as to obtain information about associated rewards/losses $y_1,\dots,y_n$. Different problems correspond to different goals to be reached with the queries. So-called best-arm identification consists of querying some $x_i$ with smallest possible loss in the least amount of time. A similar problem (apparently introduced here) is to minimize the losses of queried points up to the point where a best arm is found. Finally, in active learning the rewards/losses are binary labels, and the goal is to learn the function $f$ from a given class that best approximates the labels.

This paper consider the realizable setting where the rewards satisfy $y_i=f(x_i)$ for an unknown function $f$ belonging to a known family $\mathcal{F}$ (the case where the $y_i$ may be noisy is considered in the Appendix). For best arm identification, reference [11] gives a nearly tight query complexity bound (without an efficient algorithm). One of the positive results of the present paper is an algorithm that is efficient under certain somewhat awkward assumptions: namely, that $\mathcal{F}$ is a family consisting of discretizations of functions that belong to a convex set $\mathcal{R}$. Cumulative loss minimization is dealt with similarly. Similarly, active learning is relegated to an appendix.

One drawback of the above algorithms is that they require the realizability assumption. The paper also shows that this is in some way unavoidable (this is the content of Propositions 1 through 3).

As far as I can tell, this is a correct and nontrivial contribution with some interest. The main technical idea is replacing the ```````"version space" (where functions compatible with the data are considered) with a convex set. To understand it, first note that a natural version for query complexity problems is to keep a version space containing all functions $f$ compatible with the results that were seen. The gist of the authors approach is to replace this version space with a convex set in function space that consists of an intersection of polynomially many semispaces with the function class $\mathcal{F}$. Using sampling methods for these convex sets, this strategy can be made to work, and one may show that the sets quickly shrink until only one candidate function is left. This works under the assumption that $\mathcal{Y}$ is finite: for infinite sets, a different approach must be used, with less precise bounds.

The results of this paper seem novel and important enough to warrant publication at NeurIPS. In particular, the problems posed here address natural open questions raised in [11] (from 2011), as well as some new problems. The lower bounds are also interesting, and experimental results against Bayesian optimization methods seem promising.

However, the exposition leaves a lot to be desired. I will itemize some issues in what follows. Do note that I have not written down typos, but there are many of them!

* The authors are a bit too cavalier when they imply that sampling in convex bodies is always "efficient". There are subtleties involved, such as the need for membership oracles and the like. Some of those issues should be discussed in the appendix.
* In particular, it would be nice to understand the complexity for at least one of the problems under consideration.
* Similarly, actual query complexity bounds are never fully worked out for the algorithm Grails. It'd be nice to understand the query complexity of the proposed methods in some of the examples considered in the appendix.
* To sum up the above suggestions, it would be nice to see a full result for some simple case of the problems considered in the paper (eg. active classification with linear functions). In this example, all relevant parameters and the complexity upper bound should be worked out.
* In section 2, the authors mention the realizable setting, and then define Active Classification without explaining what it looks like in the realizable setting.
* The assumption that $\mathcal{F}_{\mathcal{R}}$ is obtained from $\mathcal{R}$ by discretization does not seem very natural and is hard to swallow. The one case where it makes a lot of sense is that of active classification, because $\mathcal{Y}=\{1,-1\}$ in that case. Unfortunately, that case is relegated to the appendix. Somewhat awkwardly, a lower bound for Active Classification is given in the main text.
* $\S$ 5.1: this is a minor technicality, but some of the halfspaces considered in the text are open.



**Time Spent Reviewing:**

6h

---

> ### Author Response · Authors · 2021-08-09
> **Response**
>
> Thank you very much for your thorough review of our work. We will work to make the exposition clearer and fix the typos. Note that some of your comments are addressed in the response at the top.
>
> Regarding computational efficiency: we will clarify the subtleties of sampling from a convex body such as the need for a membership oracle and to emphasize this point, we will give an example of a convex body where no such membership is available (e.g., a polyhedron defined by exponentially many constraints). We will move our discussion about efficient implementation to the main body to further clarify these subtleties. Thank you for this comment.
>
> Regarding the discretized setting: while it is often natural to have a discretized output setting (e.g., 1-5 star ratings by humans), we acknowledge that removing discretization is an interesting avenue for future research.
>
>
> Regarding the open halfspaces: Thank you for catching this. We will fix this typo in the final version.
>
>
> Please see the response at the top for a discussion regarding your comments on computational efficiency and sample complexity.

---

### Official Review · Reviewer_97Fq · 2021-07-17

**Rating:** 6
**Confidence:** 3

**Summary:**

The paper considers minimax optimality for some realizable interactive learning settings. A general separation is given between agnostic and realizable learners for the label complexity of active classification. Furthermore efficient algorithms are provided for minimizing the sample complexity of best arm identification and for minimizing the loss incurred while identifying the best arm (which is subsumed by usual regret minimization). Gaps between agnostic and realizable learners in the realizable setting are also shown for these settings. Experiments show that the new algorithms are competitive and can incorporate prior knowledge to obtain gains in sample complexity.

**Limitations And Societal Impact:**

The authors adequately addressed the limitations and potential negative societal impact of their work.

**Main Review:**

The paper presents (in clear, well-organized writing) some new potentially useful insights like the cost of using an agnostic learner in a realizable setting and considers best arm identification for general function classes (in the realizable setting). Computational efficiency is obtained by novel sampling techniques from a non-convex version space of functions, although it is not clear if the algorithms translate into practical implementations as n the number of arms grows. Also it would be nice to see how the sample complexity upper bounds compare with the real sample complexity of their algorithms in the experiments. In particular it is not clear what is the gap due the the -log(probability of sampling f*) term. A clearer description of what function classes the bounds are tight and sampling is efficient for would be helpful.

**Time Spent Reviewing:**

5

---

> ### Author Response · Authors · 2021-08-09
> **Response**
>
> We thank you for your insightful review. Note that some of your comments are addressed in the response at the top.
>
> Regarding the issue of computation, the core computational bottleneck is generating the hit-and-run samples to estimate the probabilities in line 3 of Algorithm 1. While the hit-and-run samples are computationally expensive, one can easily parallelize the computation of each hit-and-run sample, thus enabling GRAILS to scale to dramatically larger instances in a distributed computation setting.
>
> Regarding the question of settings where sampling is efficient, examples are given in the appendix and include linear models, RKHSs, and convex functions. We will move some of this material to the main body in the next version of the paper.

---

### Author Response · Authors · 2021-08-09
**Response**


We thank all the reviewers for their thorough and extremely helpful comments. There is some overlap in the reviewer's comments and therefore we respond to them here. To begin, we summarize some of our key technical contributions:

1. An algorithmic framework for computationally efficient, nearly optimal algorithms for a wide variety of interactive learning problems with generic function classes.
The core technical innovations are a novel technique for sampling from a non-convex version space and new proof ideas in the sample complexity bounds.
2.  Novel, efficient, and state of the art algorithms for best arm identification, active classification, and
regret minimization. In addition to computational efficiency, in the case of regret minimization, we achieve significantly tighter guarantees than prior art, e.g. [11].
3. Several propositions showing the minimax-suboptimality of any agnostic algorithm in the realizable setting. The main technical contribution is a novel technique for deriving instance-dependent lower bounds for agnostic algorithms.

Several reviewers were interested in identifying a worked out example of a sample complexity bound and computational efficiency bound for a single setting.
Regarding sample complexity, via Proposition 5, for linear models $\mathcal{F} = [f : f(\cdot) = \langle a , \cdot \rangle \, \left\lVert a \right\rVert_2 \leq 1]$, we have a sample complexity bound of $\tilde{O}(d \log(n) \upsilon^*_{\textit{best}})$. By contrast, the sample complexity bound of the computationally inefficient algorithm in [11] for linear models is $\tilde{O}(d \upsilon^*_{\textit{best}})$. Finally, we note that one can provide non-trivial bounds on the minimax complexity $\upsilon^*_{\textit{best}}$: for example, if there is a small enough gap between consecutive output labels in $\mathcal{Y}$ and $x_1,\ldots, x_n$ are well-conditioned, then $\upsilon^*_{\textit{best}}$ scales like $d$.

Regarding computational efficiency:
The runtime of GRAILS is setting dependent due to the computational cost of the sampling oracle. In the case of linear models for instance, one can use results of [14] to show a complexity bounds of $\tilde{O}(n * |\mathcal Y| + d^3)$ though the $n$ dependence can be improved via parallelism. We will clarify this in the main body.

Several reviewers commented on the dependence on $\log(\frac{1}{\text{probability of } f^*})$. It is worth noting that while we state this somewhat differently, many other recent works published in similar venues such as ICML and AISTATS [15,18] include similar dependencies
in their theoretical guarantees. We agree that it is an important challenge for interactive learning papers to remove or further explain these dependencies, but as they are endemic to this area, not just this paper, this is beyond the scope of our work. As a first step, we show how to bound this probability for linear models in proposition 5. For example, we show that this leads to a bound of $\tilde{O}(d \log(n) \upsilon^*_{\textit{best}})$, thus removing the dependence on the sampling probability

---

### Decision · Program_Chairs · 2021-09-27

**Decision:**

Accept (Poster)

**Comment:**

The paper presents an algorithm for solving a set of problems in the realizable interactive learning framework via a new sampling based algorithm. All the reviewers liked the theoretical results in the paper. There was considerable and productive discussion among the authors and the reviewers and in the end all the reviewers agreed that the results in the paper warrant publication but the authors must do a better job in the camera ready version of discussing the pros and cons of their bounds and how they related to prior work.